# OXTR^High stroma fibroblasts control the invasion pattern of oral squamous cell carcinoma via ERK5 signaling

Liang Ding [1], Yong Fu[1], Nisha Zhu [1], Mengxiang Zhao[1], Zhuang Ding[1], Xiaoxin Zhang[1], Yuxian Song[1], Yue Jing[1], Qian Zhang[2], Sheng Chen[3], Xiaofeng Huang[3], Lorraine A O'Reilly [4,5], John Silke [4,5], Qingang Hu [1,2] ✉ & Yanhong Ni [1] ✉

The Pattern Of Invasion (POI) of tumor cells into adjacent normal tissues clinically predicts postoperative tumor metastasis/recurrence of early oral squamous cell carcinoma (OSCC), but the mechanisms underlying the development of these subtypes remain unclear. Focusing on the highest score of POIs (Worst POI, WPOI) present within each tumor, we observe a disease progression-driven shift of WPOI towards the high-risk type 4/5, associated with a mesenchymal phenotype in advanced OSCC. WPOI 4-5-derived cancer-associated fibroblasts (CAFs^WPOI4-5), characterized by high oxytocin receptor expression (OXTR^High), contribute to local-regional metastasis. OXTR^High CAFs induce a desmoplastic stroma and CCL26 is required for the invasive phenotype of CCR3^+ tumors. Mechanistically, OXTR activates nuclear ERK5 transcription signaling via Gαq and CDC37 to maintain high levels of OXTR and CCL26. ERK5 ablation reprograms the pro-invasive phenotype of OXTR^High CAFs. Therefore, targeting ERK5 signaling in OXTR^High CAFs is a potential therapeutic strategy for OSCC patients with WPOI 4-5.

Oral squamous cell carcinoma (OSCC) is a characteristic locally aggressive tumor, accounting for more than 90% of all oral cancers, and is a major global public health problem[1]. Early-stage (T1/T2N0M0) OSCC patients are often treated with local surgery alone. However, 20–40% of these patients develop local recurrence and/or regional lymph node metastasis, which is the most significant adverse prognostic factor[2]. The lack of knowledge regarding the individual histological characteristics of tumors is one of the major limitations of the currently used Tumor-Node-Metastasis (TNM) staging system. Defining additional reliable prognostic histologic markers is therefore required to predict biological behavior and better stratify and manage OSCC patients.

Pattern of invasion (POI) refers to the manner in which cancer infiltrates tissues at the tumor/host interface. It is the most important factor in histological grading systems for multiple cancer types, particularly in OSCC[3–7]. Five categories of POI have been identified in OSCC and these are further divided into two subgroups based on both invasive properties and patient survival; low-invasiveness (type 1, broad pushing and cohesive; type 2, broad pushing "fingers" or separate large tumor islands; type 3, invasive tumor islands >15 cells per island) and high-invasiveness (type 4, invasive tumor islands <15 cells per island; type 5, tumor satellites of any size ≥1 mm distant from main tumor or next closest satellite with intervening normal tissue). Intuitively, tumors invading with a discohesive and dispersed growth

[1]Central Laboratory of Stomatology, Nanjing Stomatological Hospital, Medical School of Nanjing University, 30 Zhongyang Road, 210008 Nanjing, Jiangsu, China. [2]Department of Oral and Maxillofacial Surgery, Nanjing Stomatological Hospital, Medical School of Nanjing University, 30 Zhongyang Road, 210008 Nanjing, China. [3]Department of Oral Pathology, Nanjing Stomatological Hospital, Medical School of Nanjing University, 30 Zhongyang Road, 210008 Nanjing, Jiangsu, China. [4]The Walter and Eliza Hall Institute of Medical Research, Parkville, VIC 3052, Australia. [5]Department of Medical Biology, University of Melbourne, Parkville, VIC 3010, Australia. ✉e-mail: qghu@nju.edu.cn; niyanhong12@163.com

pattern are more aggressive and develop more multifocal and recurrent tumors than those growing in a bulky, pushing fashion[8].

Intertumoral heterogeneity results in diverse POI types in different OSCC patients. Intratumoral heterogeneity, on the other hand, may lead to the simultaneous existence of several POI types within a single-tumor microenvironment (TME)[4]. In this scenario, the worst POI (WPOI) is defined as the highest score present in each patient, no matter how focal, as an independent prognostic factor in early-stage OSCC patients. Several studies have shown that cancer cells from WPOI-5 tissues have dysregulated cell-to-cell adherin signals and cellular movement pathways for tumor invasion[9,10]. It is widely acknowledged that all tumorigenic and metastatic stages are not only strictly dependent on genetic and epigenetic alternations, but also controlled by tumor microenvironment components[11,12]. In a revised "seed and soil" hypothesis, the occurrence of multifocal and recurrent epithelial tumors is proposed to result from altered organ homeostasis rather than from dysregulated control of the tumor cell itself and that tumor-stromal interactions influence metastasis[11,13]. Therefore, distinct POI types originate from wider stroma alterations, not necessarily limited to the invasive front, that generate a permissible "soil" for tumor cell metastasis. Pinpointing the stroma components that support POI shift will enable the development of strategies to target the TME of WPOI 4–5 for better tailored OSCC therapies.

In this work, we show that a stromal fibroblast cluster characterized by high expression of oxytocin receptor (OXTR) contributes to sustain the desmoplastic and mesenchymal phenotype of WPOI 4–5 tumor by upregulating nuclear ERK5 signaling. Inhibition of ERK5 effectively impedes OXTR$^{High}$ fibroblastic mediated desmoplastic stroma and tumor metastasis, indicating this approach can be a potential therapeutic avenue for OSCC.

## Results

### WPOI type shift is associated with OSCC progression

We have reported that patients with WPOI 4–5 have shorter recurrence-free survival at early-stage OSCC (T1-2N0M0), correlating with a high depth of invasion (DOI) score[14]. In order to improve the clinical diagnostics and prognostic significance for OSCC, we investigated whether the WPOI type shift as OSCC progress from early stage without metastasis to an advanced stage with a high proliferation index or metastasis capability. Five subtypes of POI were identified in whole OSCC patients and confirmed by Pan-CK staining (Fig. 1a). Although a single patient may harbor several different POIs within the heterogeneous tumor microenvironment, WPOI indicates the highest scores of POI and is a unique characteristic of each tumor (Supplementary Fig. 1a). We divided a cohort of 880 OSCC patients into two groups: early stage or advanced stage, and assessed the proportions of WPOI subtypes (grade 1–5) in each group. Histopathologically, WPOI 4–5 were associated with more severe tumor invasion, whereas the proportion of WPOI 1–3 was notably higher at the early stage (57%) compared to the advanced stage (42%) (Fig. 1b). In addition, advanced OSCC showed a significant increase in WPOI 4–5 (Fig. 1c). Phenotypically, tumor cells in the WPOI 4–5 category showed a significant mesenchymal phenotype when compared with WPOI 1–3 (Fig. 1d).

When compared with WPOI 1–3 patients, WPOI 4–5 patients were more predisposed to postoperative recurrence (36%) in advanced-stage OSCC, compared with 16.6% in early-stage OSCC (Fig. 1e, f). The recurrence-free survival rate in early-stage OSCC patients with WPOI 4–5 was 74%, which was significantly decreased to 50% in advanced-stage OSCC patients with WPOI 4–5. However, regardless of tumor progression, patients with WPOI 1–3 showed relatively high recurrence-free survival (84% in early-stage vs. 74% in advanced stage) (Fig. 1g, h) and overall survival (Supplementary Fig. 1b, c). Multivariable Cox regression analysis adjusted for established clinical risk factors, such as age at diagnosis, TNM stage, and differentiation grade, demonstrated that WPOI served as a significant independent prognostic factor for poor survival in OSCC (Fig. 1i).

The dynamic tumor stroma, includes many components, such as cancer-associated fibroblasts (CAFs), immune cells, blood vessels, extracellular matrix (ECM) components, and elevated cytokines[15], which may influence tumorigenesis. Therefore, we speculated that POI heterogeneity could be attributed to such stroma complexity and heterogeneity, resulting in distinct postoperative outcomes for OSCC patients (Fig. 1j).

### Stromal fibroblasts participate in WPOI type shift

To delineate the discrepant stroma characteristics between WPOI 1–3 and WPOI 4–5, we evaluated the staining intensity for several stromal components, including CD4$^+$ T cells, CD8$^+$ T cells, Foxp3$^+$ Tregs, CD68$^+$ tumor-associated macrophages (TAM), CD31$^+$ endothelial cells, CD146$^+$ pericytes, and Collagen I$^+$ stromal cells (Fig. 2a). The only component found to be significantly altered between WPOI 1–3 and WPOI 4–5 types were collagen-I expressing stromal cells (Fig. 2b). Moreover, we retrospectively analyzed the ratio and absolute number of lymphocyte subsets in peripheral blood of OSCC patients ($n = 328$) (Supplementary Fig. 1d), no significant difference was found in different WPOI subgroups (Supplementary Fig. 1e, f). The expressions of five classical inflammatory mediators (IL-1β, IL-6, IL-8, SDF-1, TNF-α) were also simultaneously detected in situ in OSCC tissues. In addition to SDF-1, fibroblast-derived IL-1β/IL-6, tumor cell-derived-IL-8, and tumor-infiltrated lymphocyte-derived-TNF-α were upregulated in OSCC patients with high TNM stage (Supplementary Fig. 1g–k). However, only fibroblast-derived IL-1β was enriched in WPOI 4–5 tissues (Supplementary Fig. 1g). Thus major pro-inflammatory cytokines and chemokines are important contributors to tumor progression but may not be the primary instigating factors inducing different WPOI subtype.

Since stromal fibroblasts are the key cell type responsible for the production of collagen in the tumor microenvironment, we analyzed α-SMA in the tumor center (bed) and at the invasion front in 150 independent OSCC patients (Fig. 2c, d). Notably, both tumor center and invasion front harbored higher α-SMA$^+$ stromal fibroblasts in WPOI 4–5 compared to the WPOI 1–3 tumors, which was confirmed by fibronectin and PDGFRβ immunoreactivity (Supplementary Fig. 1l, m). Thus, we isolated primary CAFs for in vitro analysis. After screening postoperative pathological characteristics, we successfully acquired WPOI 1–3 type-derived CAF cell lines (CAF$^{WPOI\,1-3}$, $n = 3$) and WPOI 4–5 type-derived CAF cell lines (CAF$^{WPOI\,4-5}$, $n = 9$) for further studies (Fig. 2e). Conventional fibroblast markers, including PDGFRβ, FAP, and CD29, failed to distinguish between WPOI 1–3 and WPOI 4–5 types. However, using the ECM-remodeling assay, the matrix remodeling capacity of CAF$^{WPOI\,4-5}$ was much stronger than CAF$^{WPOI\,1-3}$ (Fig. 2f, g). In three-dimensional (3D) organotypic invasion and growth models (Fig. 2h), tumor cells were placed on top of Matrigel embedded with CAFs. CAF$^{WPOI\,4-5}$ significantly increased the invasive capacity of OSCC epithelial cell HSC3 into the CAF-remodeled matrices (Fig. 2i). Additionally, CAF$^{WPOI\,4-5}$ also promoted the proliferation of organoids in co-culture with increased intensity of pan-CK staining (Fig. 2j).

To verify the above findings in vivo, we subcutaneously co-injected human CAFs and HSC3 into nude mice. Tumors arising from the HSC3 OSCC cell line co-injected with CAF$^{WPOI\,4-5}$ showed significantly accelerated growth, displaying a pattern of marked and widespread cellular dissociation (in small cellular groups, $n < 15$), resembling the WPOI 4–5 type with more micrometastatic foci than those of mice co-injected with CAF$^{WPOI\,1-3}$ (Fig. 2k, l and Supplementary Fig. 1n). The more severe phenotype was accompanied by more abundant stroma, as determined by Masson Trichrome and Sirius red staining (Supplementary Fig. 1o). We also established an orthotopic model of OSCC in NOD$^{prkdc-/-IL-2Rg-/-}$ mice by injecting human OSCC cell line HN6/hCAFs into the tongue. As in the previous xenograft model,

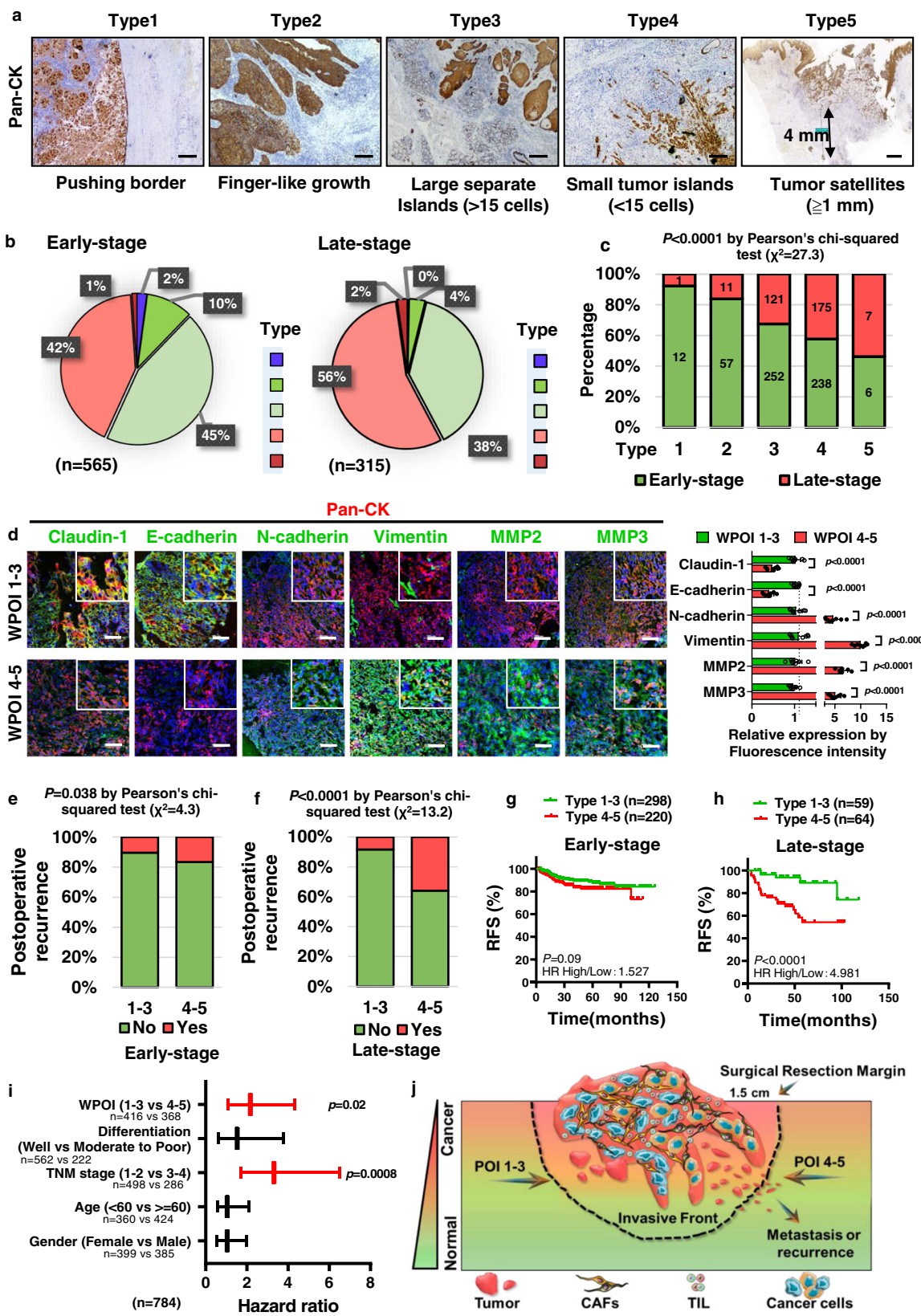

CAF$^{\text{WPOI 4-5}}$ dramatically enhanced tongue tumorigenesis (Fig. 2m, n), lymph node metastasis (LNM) (Fig. 2o and Supplementary Fig. 1p), and micrometastatic foci (Fig. 2p) in the tongue. Collectively, these data suggested that CAF$^{\text{WPOI 4-5}}$ engaged in the tumor invasion and metastasis phenotype of WPOI 4–5 type tumor.

## The CAF$^{\text{WPOI 4-5}}$ is characterized by high OXTR expression

To better understand the molecular differences between CAF subtypes, we compared the transcriptomes of our previously isolated CAF$^{\text{WPOI 1-3}}$ and CAF$^{\text{WPOI 4-5}}$ types. Conventional fibroblast markers failed to distinguish these CAF subtypes (Supplementary Fig. 2a), but KEGG

**Fig. 1 | Switch to WPOI 4–5 type is associated with OSCC progression.**
**a** Representative images in OSCC patients of the five types of POI stained with pan-cytokeratin (CK). Scale bar: 100 µm for type 1–4, 1000 µm for type 5. Representative of $n = 200$ total. **b** Pie chart showing the distribution of WPOI types in early- or late-stage OSCC patients. **c** Bar graph showing the gradual increasing ratio of WPOI 4–5 in late-stage OSCC. $P$ = Pearson's correlation test. **d** Immunofluorescence staining of EMT-related markers from OSCC tumor samples of WPOI 1–3 and 4–5 types ($n = 8$/WPOI type). $P$ = multiple $t$ test. Scale bar: 100 µm. **e, f** Graphical depiction of the distribution ratio of postoperative recurrence in early stage (WPOI 1–3: $n = 298$ and WPOI 4–5: $n = 220$) or advanced stage (WPOI 1–3: $n = 59$ and WPOI 4–5: $n = 64$) of OSCC patients, classified by WPOI type. $P$ = Pearson's correlation test. **g, h** Graphical depiction of relapse-free survival from early-stage (WPOI 1–3: $n = 298$ and WPOI 4–5: $n = 220$) and late-stage-stage (WPOI 1–3: $n = 59$ and WPOI 4–5: $n = 64$) OSCC patients classified as WPOI types 1–3 or 4–5. $P$ = Log-rank (Mantel–Cox) test. **i** Graphical depiction of the independent prognostic significance of WPOI as determined by multivariable analysis (Cox proportional hazards model) ($n = 784$). **j** Schematic representation of the predicted heterogeneous tumor microenvironments within OSCC tissues for varying POI types. CAF cancer-associated fibroblasts, TIL tumor-infiltrated lymphocytes. Results are shown as mean and standard deviation (SD). Source data are provided as a Source data file.

pathway annotation and enrichment analysis for differentially expressed genes revealed that several pathways were significantly enriched: neuroactive ligand–receptor interaction, calcium signaling pathway, oxytocin signaling, cytokine-cytokine receptor interaction, and cell adhesion molecules (Supplementary Fig. 2b). From the differential genes involved in these pathways, we identified several gene clusters uniquely upregulated in the CAF$^{WPOI\ 4-5}$ type; neuronal development and function-related (*RELN, MYH10, CACNB4, OXTR, CAV3, CHRM2*) and inflammation and cancer-related (*RELN, IL21R, EDNRB, CAV3, OXTR*) (Fig. 3a). Among these differentially dysregulated pathways, oxytocin signaling is significantly related with multiple pathways involved in tumorigenesis[16–19]. In addition, *OXTR* showed the highest gene expression abundance in this study when compared with other genes (Supplementary Fig. 2c), therefore *OXTR* was chosen for further investigation. *OXTR* encodes the human oxytocin (OXT) receptor, and it is expressed predominantly in the brain. OXTR signaling can function as a stress-managing molecule, an anti-inflammatory, and an antioxidant, with protective effects particularly in the face of distress or trauma[20]. Hormones could link the tumor development with mental and physiological stress[21], but the function of peripheral OXT/OXTR signaling during carcinogenesis of OSCC requires further clarification[22–24]. To explore this further, we first sought to confirm our initial finding by expanding the OSCC sample size, and we observed a similarly increased *OXTR* mRNA expression in CAF$^{WPOI\ 4-5}$ compared to the CAF$^{WPOI\ 1-3}$ type (Fig. 3b). Western blots from OSCC-derived CAFs showed that endogenous OXTR migrated at ~75 kDa, similar to previous reports[25–28]. Other studies have indicated that OXTR protein expression was detected at ~45 (predicted), 55, or 66 kDa; however, this discrepancy could be attributed to the different tissue origins and cell types used and the putative glycosylation sites at the N-terminus[29–31]. Although *OXTR* DNA-methylation levels have been associated with dysregulated *OXTR* expression in many neurodevelopmental disorders[32], no significant relationship between *OXTR* methylation and expression was found in the different WPOI-derived CAFs (Supplementary Fig. 2d).

To further characterize the cell subset expressing OXTR in human OSCC, we first analyzed the correlation between *OXTR* and the genes differentially represented in a variety of tumor-associated cell populations in head and neck squamous cell carcinoma (HNSCC, $n = 522$, TCGA dataset, Firehose Legacy) (Fig. 3c). Strikingly, several markers of activated fibroblasts were positively associated with *OXTR* level. Analysis of data from cell sorting indicated that the primary source of *OXTR* mRNA was also observed in PDGFR-β⁺ and FAP⁺ stroma fibroblast population (Fig. 3e). Furthermore, we observed a similar increase in OXTR expression in CAFs isolated from breast and pancreas cancers (Fig. 3d). The localization of OXTR⁺ CAFs in OSCC in situ was verified by FISH (Fig. 3f), where densities of α-SMA⁺OXTR⁺ CAFs in WPOI 4–5 type were two- to threefold higher than that WPOI 1–3 type and low in non-cancerous epithelial regions (Fig. 3j, h). Correspondingly, flow cytometry analysis showed that OXTR$^{High}$ CAFs were also enriched in CAF$^{WPOI\ 4-5}$ (Supplementary Fig. 2e).

Three broadly defined CAF subpopulations were first described in pancreatic ductal adenocarcinoma (PDAC) and referred to as myofibroblastic CAFs (myCAFs, high α-SMA), inflammatory CAFs (iCAFs, low α-SMA, and high inflammatory cytokines) and antigen-presenting CAFs (apCAFs, high MHC II–related genes)[33]. The levels of *OXTR* mRNA were also found to positively correlate with genes characteristic of myCAFs, rather than iCAFs or apCAFs (Supplementary Fig. 2f). Gene Set Variation Analysis (GSVA) was performed on scRNA-seq datasets from human HNSCC (Fig. 3i and Supplementary Fig. 2g), PDAC biopsies (Supplementary Fig. 2h, i) and mouse PDAC model (Supplementary Fig. 2j). We found that activated OXTR signaling was markedly enriched in myCAFs. Therefore, α-SMA⁺ OXTR⁺ stroma fibroblasts can be categorized as a myCAFs subpopulation. Further, in formalin-fixed, paraffin-embedded OSCC tissues, OXTR immunoreactivity was found to be enriched in stromal fibroblasts from CAF$^{WPOI\ 4-5}$ compared to CAF$^{WPOI1-3}$, but not in tumor cells (Fig. 3j). More than 80% patients with WPOI 4–5 had moderate to high expression of stromal OXTR, compared to only 30% in WPOI 1–3 stoma (Fig. 3k). Interestingly, OSCC patients with moderate to high OXTR⁺ CAF accounted for remarkably high risk for poor prognosis (Fig. 3l).

## OXTR deficiency impairs ECM reconstruction and tumor invasion

To investigate a putative role for OXTR expression in CAF$^{WPOI4-5}$-dependent OSCC tumor invasion, we performed a transcriptomic profiling analysis of different OSCC patients-derived CAFs with high or low OXTR levels. KEGG enrichment analysis revealed that high OXTR was associated with a signature encompassing cell adhesion and ECM remodeling (Supplementary Fig. 3a). GSEA analysis of the HNSCC dataset showed a similar finding (Fig. 4a). Next, we isolated OXTR$^{High}$ and OXTR$^{Low}$ CAFs from CAF$^{WPOI4-5}$ cell line by flow cytometry sorting (Fig. 4b and Supplementary Fig. 3b) and measured the expression of eight fibroblast markers (*PDGFRB, FAP, VIM, CDH2, CAV-1, ACTA2, FSP-1, CD29*). We found that OXTR$^{High}$ CAFs were associated with high expression of *PDGFRB* and *CDH2* (Fig. 4c).

To further investigate the control of stromal cell plasticity, we co-cultured the HN6 OSCC cell line with OXTR$^{High}$ or OXTR$^{Low}$ CAFs in an ultra-low adhesive environment (Fig. 4d). We found that organoid proliferation and the ability to form heterotypic spheroids was dependent on the presence of OXTR$^{High}$ expressing CAFs. When the OXTR$^{High}$ CAFs and/or HN6 organoids were cultured in Matrigel mixed with collagen I(heterotypic spheroids), the maximal invasive distance from the spheroid border (Lmax) was increased, regardless of whether they were homo- or heterotypic spheroids (Fig. 4e and Supplementary Fig. 3c). Human OXTR is activated and binds to its natural ligand OXT with high affinity or to arginine vasopressin (AVP) with low affinity[20]. To investigate the role of OXT/AVP ligand signaling in facilitating the invasive capacity of OXTR$^{High}$ CAFs, the OXT/AVP concentration in the organoid culture system was assessed. However, we failed to detect either OXT or AVP in these in vitro culture systems (Supplementary Fig. 3e, f). Carbetocin, a long-acting oxytocin analog, also showed no effects on OSCC progression in orthotopic model of OSCC in NCG mice (Supplementary Fig. 3g). These findings suggest OXTR$^{High}$ CAFs promote tumor cell invasion in a ligand-independent manner.

To provide more direct evidence of OXTR function, we used lentiviral CRISPR/Cas9 editing to stably knock down *OXTR* in OXTR$^{High}$ CAFs (Supplementary Fig. 3h). T7E1 assay (Supplementary Fig. 3i) and

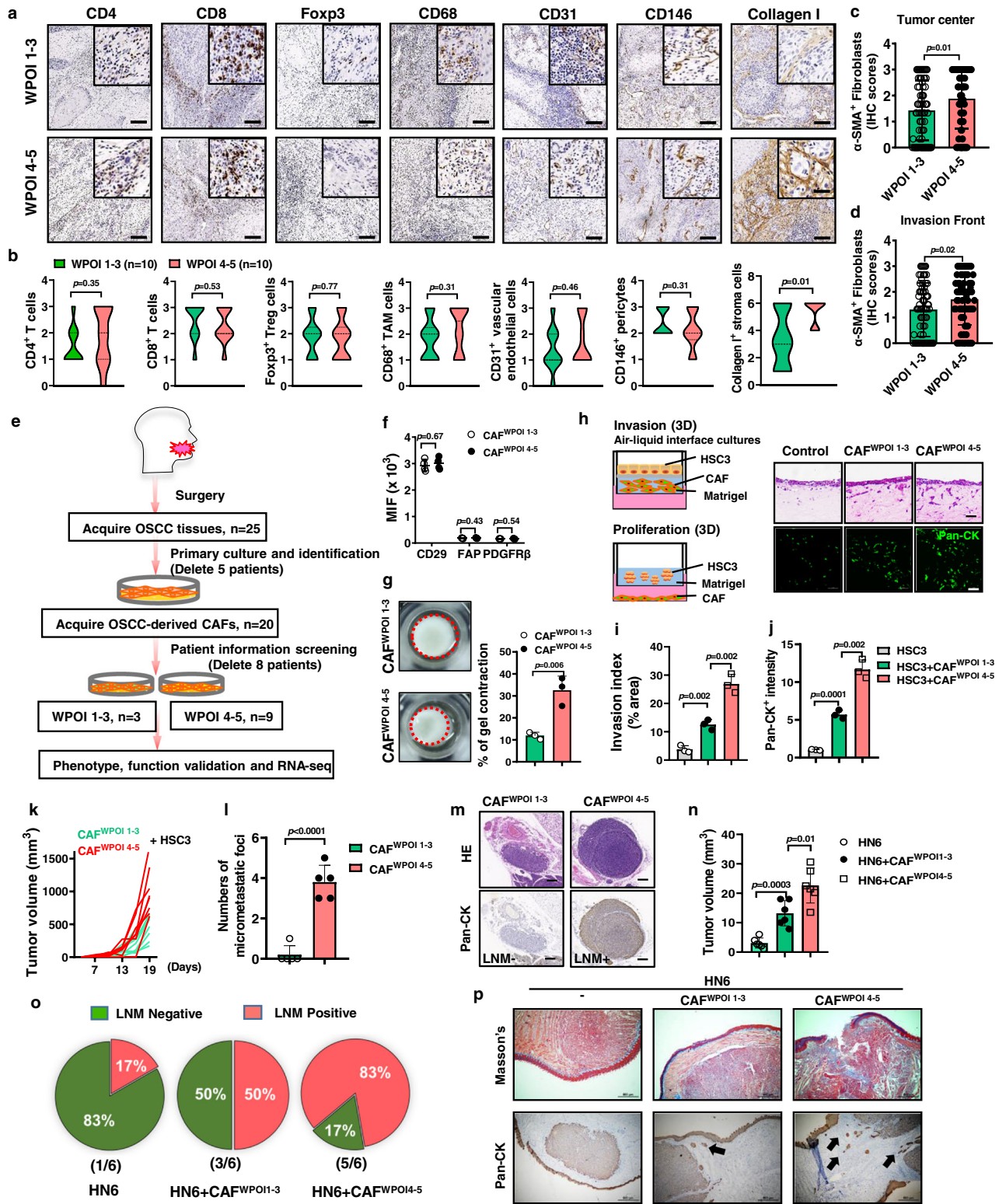

western blot (Fig. 4f) confirmed ablation of OXTR in edited cells, which effectively reduced expression of PDGFR-β and N-cadherin in OXTR^High CAFs. Considering that the members of the OXTR/AVPR family are conserved and share a high degree of sequence similarity[34], we determined AVPR1 (V1aR), AVPR2 (V2R) and AVPR3 (V1bR) levels in response to OXTR knockdown, however, the expression levels of three AVPR family member remained unchanged (Supplementary Fig. 3j). OXTR^High CAFs-induced contraction of collagen gels and pro-invasion of HN6 in vitro could be significantly impaired by OXTR knockdown

(Supplementary Fig. 3k). In vivo, we found that mice co-injected with OXTR^High CAFs and HN6 tumor cells had a more significant tumor burden and increased metastasis to lymph nodes than OXTR^High CAFs deficient in OXTR (Fig. 4h, i). Histological analysis revealed that desmoplastic stroma and impaired invasive phenotypes were also reduced by OXTR knockout (Supplementary Fig. 3l). In order to specifically investigate the role of fibroblast-derived OXTR in OSCC, we created a double mutant *s100a4-cre; Oxtr^fl/fl* mouse model. Double IHC staining for OXTR and FSP-1 indicated that fibroblast OXTR deletion

**Fig. 2 | CAF^WPOI4-5 enhances the invasiveness of OSCC.** Stromal components were estimated in WPOI 1–3 ($n = 10$) and 4–5 ($n = 10$) type OSCC samples by (**a**) IHC analysis and (**b**) graphically summarized. Scale bars, 200 μm and 50 μm. $P =$ two-tailed $t$ test. Graphical summary from IHC analysis of α-SMA+ stained stromal fibroblasts from **c** tumor centers or (**d**) at the invasion front of OSCC (WPOI 1–3: $n = 80$ and 4–5: $n = 70$). $P =$ two-tailed $t$ test. **e** Schematic of the protocol used to isolate, culture, and establish primary human CAF^WPOI1-3 and CAF^WPOI4-5 types, based on patient pathology. **f, g** CAF markers and the ability of ECM remodeling were analyzed by flow cytometry ($n = 4$) and Matrix gel contraction assay ($n = 3$). $P =$ multiple $t$ tests and two-tailed $t$ test. **h** Setup of 3D Organotypic cultures of CAFs/tumor cells and (**i**) graphical summaries of estimates of the ability of tumor invasion and (**j**) proliferation. Tumor HSC3 cells were stained with a green fluorescent dye. $n = 3$/group. Scale bar: 50 μm, $P =$ two-tailed $t$ test. **k** In all, $1 \times 10^6$ HSC3 tumor cells were mixed with $2 \times 10^6$ CAF^WPOI1-3 or CAF^WPOI4-5 and injected subcutaneously into male NCG mice, with the graphical representation of tumor volume, measured using slide calipers and (**l**) frequency of micrometastatic foci. $n = 5$/group, $P =$ two-tailed $t$ test. **m** In total, $2 \times 10^5$ HN6 OSCC tumor cells were mixed with $1 \times 10^6$ CAF^WPOI1-3 or CAF^WPOI4-5 and injected into the anterior portion of the tongue of NCG mice, HN6/CAF^WPOI4-5-derived tumor induce LNM (LNM +) by CK staining with (**n**) tumor volume and (**o**) LNM percentage graphically represented. $n = 6$. Scale bar: 200 μm, $P =$ two-tailed $t$ test. **p** Representative photomicrographic images showing increased desmoplastic stroma components and micrometastatic foci (arrows) in mice injected with CAF^WPOI4-5 and HN6 cells, compared to CAF^WPOI1-3. Scale bars, 500 μm. Results are shown as mean and standard deviation (SD). Source data are provided as a Source data file.

was efficient (Fig. 4j). The murine OSCC cell line SCC7 was used to induce orthotopic OSCC carcinoma in the *s100a4-cre; Oxtr^fl/fl* double mutant mice. While the tumor volume of *s100a4-cre; Oxtr^fl/fl* tumors were reduced, this was not statistically significant (Fig. 4k), however the incidence of LNM was remarkably decreased, visualized as highly inhibited desmoplastic properties (Fig. 4l, m).

### OXTR^High CAFs-derived chemokine CCL26 induces a mesenchymal phenotype in CCR3+ tumor cells

Many studies have highlighted the paracrine and autocrine effects of myCAFs[35]. A comparison of conditioned media (CM) from OXTR^Low and OXTR^High CAFs using a Human Cytokine Array revealed that seven cytokine/chemokines were significantly upregulated in OXTR^High CAFs, including CXCL6, CCL26, CCL18, IL-1β, CCL7, CCL5, and TNFSF14/LIGHT (Fig. 5a, b), IL-1β expression in situ in WPOI 4–5 tissues was also upregulated as showed above. Loss of OXTR inhibited all the cytokines/chemokines tested, with the exception of CXCL6 (Fig. 5c). Post co-culture with OXTR^Low CAFs and recombinant cytokines/chemokines, we quantified the Lmax of tumor cell invasion in conditional 3D collagen matrix (Fig. 5d, f). The addition of recombinant human CXCL6, IL-1β, CCL7, CCL5, or LIGHT into the 3D co-culture system, resulted in a modest increase in Lmax, in accordance with other reports[36,37]. On the other hand, the addition of CCL18 or CCL26, caused a doubling in Lmax. In order to confirm the specific impacts of CAFs in this co-culture model, we also knocked down seven cytokine/chemokine genes in OXTR^High CAFs. In this system, only CCL26 knockdown in OXTR^High CAFs significantly attenuated the invasion length of tumor cells (Fig. 5e, g). Thus, knockdown of IL-1β and CCL7 etc. were insufficient to inhibit tumor invasion. Similar assays also found that the invasive ability of OXTR^Low CAFs itself was significantly promoted by CCL18 and CCL26 (Supplementary Fig. 4a–d). In order to specifically mimic the histopathology of tumor invasion in vivo, we adapted another organotypic invasion assay by depositing tumor cells on top of Matrigel-collagen-embedded CAFs ± cytokine/chemokines to induce tumor cell invasion (Fig. 5h, i). Similar to our previous results, CCL18, CCL26, and LIGHT had the most dramatic effects in promoting OXTR^Low CAFs-induced HN6 invasion (Fig. 5j), but only knockdown of CCL26, and with a lesser extent of CCL18, impaired OXTR^High CAFs-dependent tumor invasion (Fig. 5k). Thus, CCL26 in OXTR^High CAFs had a relatively superior ability to promote tumor invasion.

Since elevated CCL26 appears to be a generalized finding in a broad range of cancer stroma (Supplementary Fig. 4e), we explored its role further. To investigate whether CCL26 receptor CCR3 expression was also required for CCL26-induced tumor invasion, sgRNA-CCR3 was used to stably knockdown *CCR3* in HN6 OSCC cells (Supplementary Fig. 4f). As before, CCL26 promoted matrix invasion by tumor cells in the OXTR^Low CAF/Tumor co-culture model but this was abrogated by CCR3 knockout (Fig. 5l). Moreover, when HN6 was treated with OXTR^Low CAFs-derived culture medium, added extra CCL26 caused upregulation of snail, twist and N-cadherin but down-regulation of E-cadherin expression. This process was impaired in HN6 cells with CCR3 knockdown (Supplementary Fig. 4f). Consistently, CCR3 was upregulated in WPOI 4–5 tumor cells (Fig. 5m) and showed a marked dysregulated elevation in EMT pathway-related genes in HNSCC patient samples (Fig. 5n), which correlated with a high incidence of recurrence of various human cancers (Fig. 5o and Supplementary Fig. 4g–i). These findings further support that OXTR^High CAF-derived CCL26 is a recurrence-related risk factor for human CCR3+ tumors.

### Activated ERK5 are indispensable for the phenotype and function maintenance of OXTR^High CAFs

We have shown that sustained OXTR and CCL26 expression are critical for the pro-invasive function of OXTR^High CAFs. However, unraveling the control of its transcriptional regulation is necessary for the development of potential intervention strategies. Small activating RNAs (saRNAs) are short double-stranded oligonucleotides that can endogenously and selectively enhance gene transcription through RNA interference machinery and have been developed for clinical use[38,39]. To explore saRNAs as a means to control endogenous *OXTR* gene expression, we designed four candidate saRNAs targeting the human *OXTR* gene promoter (Fig. 6a) to overexpress OXTR in OXTR^Low CAFs to mimic OXTR^High CAFs. Among these saRNAs, saRNA-OXTR-3 and -4 were shown to significantly upregulate OXTR protein expression levels in CAFs, accompanied by an upregulation in the activated CAF signatures for PDGFR-β, N-cadherin, FAP, and CAV-1. sa-OXTR-4 was also confirmed to promote transcription of the *OXTR* gene in CAFs using a dual-luciferase reporter assay (Fig. 6b). We next performed an ATAC−seq assay to assess genome-wide chromatin assembly by measuring transposase-accessible chromatin in CAFs overexpressing OXTR. Since saRNA-mediated transcriptional activation is dependent on the histone remodeling enzymes-open chromatin structure and RNA polymerase II-mediated transcription initiation[40], more accessible regions in *OXTR* were found in CAFs overexpressing sa-OXTR, and in addition we also discovered that *GNAQ, GNAI, PRKC, COLs, MMPs,* and *CCL26* genes were more accessible in the sa-OXTR group than in the sa-negative control (NC) group (Fig. 6c). In order to identify the key TFs orchestrating the OXTR and CCL26 expression in OXTR^High CAFs, we performed a Genome Transcription Regulation Database (GTRD) database analysis (http://gtrd.biouml.org/). Several predicted TF-binding sites in the *OXTR* and *CCL26* promoters were found and, in particular, the mitogen-activated protein kinase (MAPK) pathways harbored the most TFs targeting the *OXTR* and *CCL26* promoters (Supplementary Fig. 5a). MAPK signaling is composed of four classical pathways including JNK, ERK1/2, p38, and ERK5[41]. However, more activated nuclear ERK5, rather than JNK, ERK1/2, or p38, was found in sa-OXTR group (Fig. 6d, e). Western blot analysis of cytoplasmic and nuclear fractions also showed elevated total ERK5 levels and nuclear phospho-ERK5 (p-ERK5) levels in OXTR^High CAFs (Fig. 6f).

To confirm a specific role for ERK5 activation in the maintenance of OXTR and CCL26 expression, we knocked down JNK, ERK1/2, p38, or ERK5 in sa-OXTR CAFs. Consistent with our earlier findings, reduction in ERK5 levels significantly impaired several activated CAF signatures

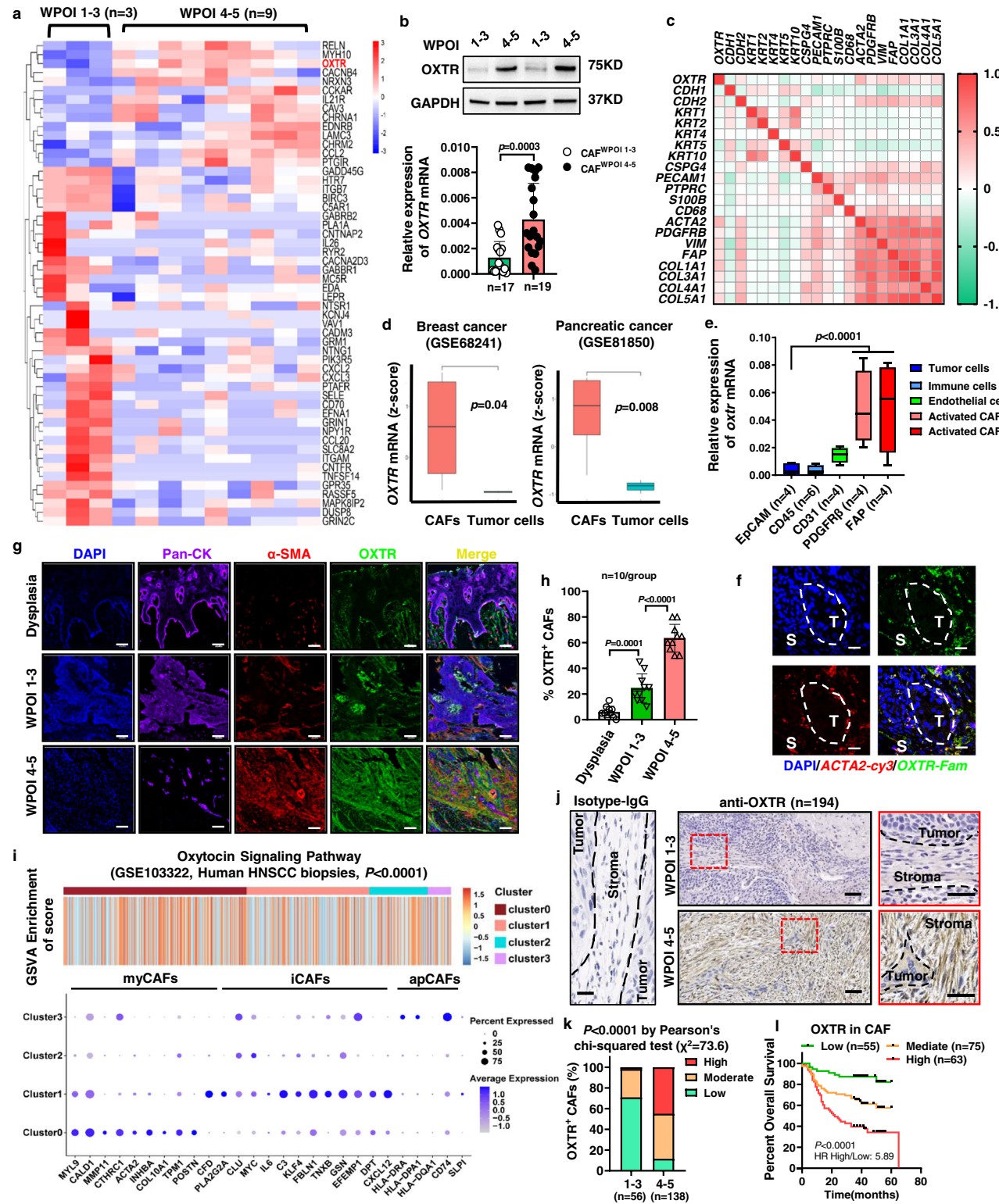

and CCL26 and OXTR transcription and translation in OXTR^High CAFs (Fig. 6g and Supplementary Fig. 5b). When the gene knockdown was replaced with small molecular inhibitor, the ERK5 inhibitor XMD8-92 could also efficiently suppress the CCL26 and OXTR protein expression and transcriptional activity (Supplementary Fig. 5c, d). ERK5 has been reported to recruit many TFs to promote transcription, including c-fos, c-jun, c-myc, and MEF2C[42]. Similar to previous findings, we also observed such interactions between ERK5 and these TFs in sa-OXTR CAFs by co-immunoprecipitation (Fig. 6h). Elevated c-fos and c-myc

were found to bind to activated ERK5 in sa-OXTR CAFs, and translocate into the nucleus (Supplementary Fig. 5e, f), indicating that c-fos and c-myc levels might be recruited by ERK5 for gene transcription. Moreover, knockdown of c-fos and c-myc, but not c-jun or MEF2C, was able to abrogate OXTR/ERK5 signaling-induced OXTR and CCL26 expression (Supplementary Fig. 5g). The JASPAR database (http://jaspar.genereg.net/) predicts several potential binding sites for c-fos and c-myc within the OXTR (Fig. 6i) and CCL26 promoters (Fig. 6k), and mutations within these predicted promoter binding sites strongly

**Fig. 3 | CAF^WPOI 4-5 type is characterized by high OXTR expression. a** Differential gene analysis on enriched KEGG pathway from RNA sequencing of CAF^WPOI 1-3 (*n* = 3 independent CAFs) and CAF^WPOI 4-5 (*n* = 9 independent CAFs). Individual gene expression is shown as in a heatmap. **b** OXTR protein and mRNA expression in primary CAFs. *P* = Student's *t* test. The experiments were performed three times with similar results. **c** Correlative analysis between OXTR gene expression and other stromal cell markers in HNSCC, *n* = 530 (TCGA dataset, Firehose Legacy). **d** Graphical representation of OXTR mRNA expression in CAFs or tumor cells from breast cancer (CAF: *n* = 3, tumor cell: *n* = 4) and PDAC (CAFs: *n* = 3, tumor cells: *n* = 2), *P* = two-tailed *t* test. **e** Graphical representation of OXTR mRNA expression from flow cytometry-sorted EpCAM^+ tumor cells, CD45^+ leukocytes, CD31^+ endothelial cells, PDGFR-β^+ or FAP^+ stroma fibroblasts. *P* = two-tailed *t* test. **f** Representative fluorescence in situ hybridization in *ACTA2*-cy3 and *OXTR*-fam-labeled stroma fibroblasts (scale bar: 20 μm) and (**g**) IF staining of OSCC tissues with WPOI 1–3 or 4–5 type indicated the localization of OXTR^+ CAFs (scale bar: 100 μm) and (**h**) relative quantitation (*n* = 10 independent samples for each group), *P* = two-tailed *t* test. **i** Representation of GSVA gene set analysis of the oxytocin signaling pathway performed on human HNSCC scRNA-seq datasets (GSE103322). The similarities between different fibroblasts subtypes and myCAFs, iCAFs and apCAFs were analyzed in the lower bubble plots. **j** Representative IHC analysis of staining for OXTR^+ CAFs from OSCC patients with WPOI 1–3 (*n* = 56) or 4–5 (*n* = 138) and (**k**) their graphical quantification, *n* = 194. Scale bar: 20 μm (left), 100 μm (middle), 50 μm (right). **l** Graph representing the correlation between OXTR^+ CAFs and OSCC patient postoperative recurrence and survival. *P* = Pearson's correlation test and Log-rank (Mantel–Cox). *n* = 193, scale bars, 50 μm. Results are shown as mean and standard deviation (SD). Boxes indicate the first and third quartiles, bands indicate medians, and whiskers indicate ±1.5 interquartile range. Source data are provided as a Source data file.

suppressed the transcriptional activity of *OXTR* (Fig. 6j) and *CCL26* promoters (Fig. 6l).

Next, we aimed to determine the mechanism of ERK5 activated in OXTR^High CAFs. ERK5 in addition to its ability to activate downstream targets by phosphorylation in classic MAPK fashion, is unique among MAPKs in having a transcriptional activation domain. Thus nuclear translocation is a particularly important mechanism to regulate its activity by HSP90/CDC37-induced SUMOylation and MEK5-induced phosphorylation[43–46]. Indeed, we observed an upregulation of MEK5 and nuclear ERK5 in sa-OXTR CAFs (Fig. 6f and Supplementary Fig. 5h). We also observed an increase in the levels of endogenous CDC37-ERK5 complex, elevated HSP90 dissociation, and extensive ERK5 SUMOylation/phosphorylation (Fig. 6m, n). Knockdown of CDC37 could impair ERK5 activation and nuclear translocation in sa-OXTR CAFs (Supplementary Fig. 5i, j). Moreover, as a member of the GPCR family, OXTR can couple to a trimeric complex of G proteins (Gα and Gβ/γ) for signal transduction. Recent studies have also reported that GPCR activation can promote direct interaction between Gαq and protein kinase C (PKC)ζ to activate the MEK5/ERK5 pathway, but β-arrestin recruitment is not required for this process[47,48]. We, therefore, determined the role of Gαq and β-arrestin in OXTR-induced ERK5 activation by endogenous co-immunoprecipitation (IP) assays (Fig. 6o). Our analysis showed elevated ERK5, PKCζ, and MEK5 expression in the Gαq-IP complex (Fig. 6p). However, PKCζ and MEK5 showed weak binding affinity to β-arrestin, demonstrating that β-arrestin might not serves as a scaffold for MEK5/ERK5 activation in CAFs. Moreover, when we knocked down the expressions of Gαq and β-arrestin 1/2 by siRNAs (Fig. 6q), Gαq inhibition, but not β-arrestin 1/2, significantly reduced MEK5/ERK5 activation and OXTR expression (Fig. 6r and Supplementary Fig. 5k), which was consistent with previous co-IP results. Thus, OXTR/ERK5 activation also could be mediated by Gαq in CAFs. Similar to the sa-OXTR CAFs, there was an increase in nuclear p-ERK5 in sorted OXTR^High CAF compared with OXTR^Low CAFs which correlated with elevated MEK5 and CDC37 levels (Fig. 6s, t). The key findings were repeatable in primary CAFs which also showed that knockdown of OXTR significantly reduced pro-invasion ability and down-regulated ERK5 signaling (Supplementary Fig. 5i–o). In summary, ERK5/c-fos/c-myc transcriptional signaling in OXTR^High CAFs can be attributed to elevated OXTR and CCL26 expression, which, in turn, further activate CDC37/Gαq/ERK5 signaling to form a positive feedback loop, which is the key mechanism for maintaining the high OXTR/CCL26 phenotype in OXTR^High CAFs (Fig. 6u).

### Suppression of ERK5 attenuates the pro-invasion effects of OXTR^High CAF

To investigate the therapeutic potential of ERK5 inhibition in WPOI type 4−5 tumors, we knocked down ERK5 in sa-OXTR CAFs (Fig. 7a). Our analysis indicated that the reduction of ERK5 contributed to reduced tumor proliferation and invasion in the 3D collagen matrix model (Fig. 7b and Supplementary Fig. 6a, b). This was also associated with reduced expression levels of the ECM components fibronectin and MMP3 (Fig. 7c). The function of ERK5 inhibition in vivo was also determined by mixing OSCC cells with sa-OXTR CAFs, implanted into immunocompromised NCG mice with suppression of ERK5 using the inhibitor XMD8-92 or ERK5 knockdown (Fig. 7d). Interestingly, both XMD8-92 and ERK5 knockdown significantly induced tumor growth, but only ERK5 knockdown had the additional ability to overcome desmoplastic stroma by reducing the incidence of LNM and the mesenchymal phenotype of the tumor (Fig. 7e). These findings might attribute to the XMD8-92-mediated paradoxical activation of ERK5 transcriptional activity or the off-target effects on the bromodomain-containing protein[49]. We also knocked down CCL26 in OXTR^High CAFs and established an orthotopic model of OSCC, and demonstrated that CCL26 knockdown could impair OXTR^High CAFs-supported tumor growth and LNM (Supplementary Fig. 6c), which is consistent with the findings in Fig. 7d.

To provide additional clinical evidence to support an ERK5 targeting therapy approach, we assessed paired nuclear ERK5 scores, CCL26 expression and their correlation with OXTR in individual OSCC patients (*n* = 106, Fig. 7f). IHC analysis on serial sections showed that patients with WPOI 4–5 type had significantly elevated nuclear ERK5 expression and CCL26 in stroma fibroblasts (Fig. 7g). In addition, we consistently demonstrated that OXTR expression in stromal fibroblasts significantly correlated with ERK5 activation and CCL26 expression (Fig. 7h, i) and which predicted poor survival for OSCC patients (Fig. 7j, k). These results, therefore, suggest that targeting the transcription activating activity of ERK5 promises to be an effective therapeutic strategy for tumors with WPOI 4–5 type.

### Oxytocin levels in human OSCC show no impact on POI shift

Endogenous oxytocin and stimulation of the oxytocin receptor have the capacity to act as a "natural medicine" protecting against stress and illness[50]. While OXTR^High CAFs can exert their functionality in a ligand-independent manner in vitro, we wished to explore whether there might be a role for the OXTR ligands, OXT and AVP, in patient samples. In order to assess this, two OSCC cohorts were collected for study. In the test group (*n* = 180), the concentration of serum AVP in peripheral blood was the same in healthy people and patients with OSCC (median: 20.33 pg/ml in OSCC patients). On the other hand, serum OXT levels were significantly elevated in OSCC patients when compared with either healthy people or patients with leucoplakia and increased significantly as the tumor progressed (median: 81.21 pg/ml in OSCC patient vs 16.5 pg/ml in healthy people) (Fig. 8a, b). OXT and AVP levels were comparable in WPOI 1–3 and WPOI 4−5 type serum (Fig. 8c, d), which was verified in the validation group (*n* = 96) (Fig. 8e, f). Consistently, OXT and AVP levels showed no relationship to clinical TNM stage, lymph node metastasis, and postoperative metastasis in the two cohorts (Fig. 8g, h and Supplementary Fig. 6d, e). Interestingly, patients with low serum AVP level tended to have shorter survival times. However, OXT levels had no

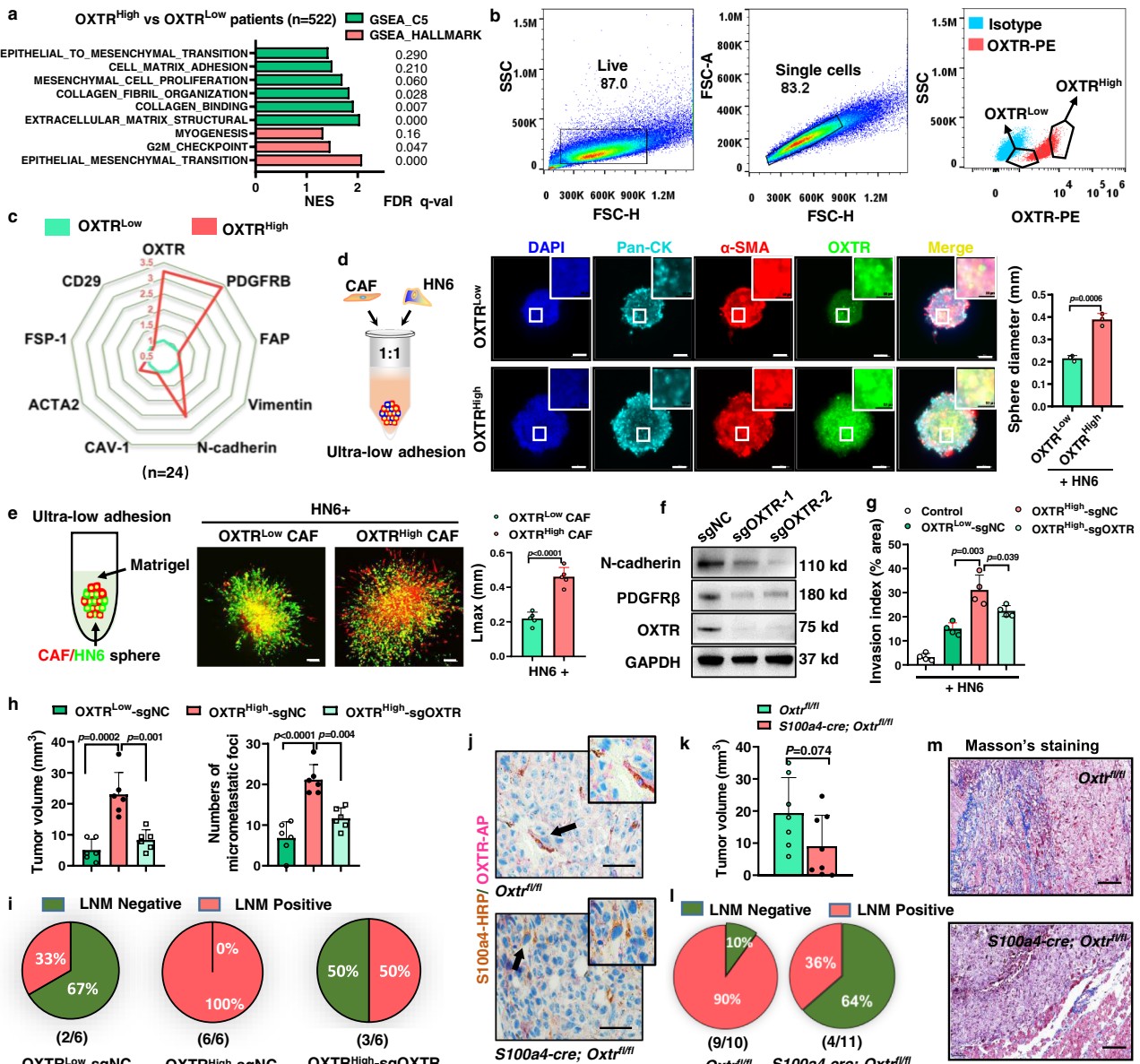

**Fig. 4 | Loss of OXTR in CAFs attenuates pro-invasion function in a ligand-independent manner. a** Top ECM-remodeling pathways hits from a GSEA analysis of OXTR^High versus OXTR^Low HNSCC patient samples, $n = 522$, using compilations H and C5 (MSigDB). **b** Flow cytometry gating used to sort OXTR^High and OXTR^Low CAFs from CAF^WPOI4-5 cell lines, and **c** estimated qPCR expression patterns of eight fibroblast markers. OXTR^Low CAF was used as normalized control (Green wire, value = 1). The red wire is OXTR^High CAF, scale of the coordinate axis is also labeled in red. **d** Setup of co-culture of HN6 with OXTR^low or OXTR^high CAFs to form heterotypic spheroids and graphical representation of organoid diameter. Pan-CK-blue−tumor cells, α-SMA-red−CAFs, green−OXTR. $n = 3$/group, $P =$ two-tailed $t$ test. Scale bars, 50 μm or 100 μm. **e** Setup of heterotypic organoid formation from CAFs (red) or mixed with HN6 (green) cultured in Matrigel gels mixed with collagen I. The matrix invasion of CAF/HN6 heterotypic organoid was determined estimating the maximal distance of invasion from the spheroid border (Lmax). $n = 5$/group, $P =$ two-tailed $t$ test. Scale bars, 100 μm. **f** CRISPR-Cas9 editing human *OXTR*, PDGFR-β, N-cadherin, and OXTR expression were validated by Weston blot. The experiments were performed three times with similar results. **g** OXTR knockdown by lentivirus-sgRNA in OXTR^High CAFs and established heterotypic organoid. Graphical representation of the Lmax of tumor cell invasion into matrix gel. $P =$ two-tailed $t$ test. **h** Graphical representation of the tumor volume, micrometastatic foci, and (**i**) pie chart representation of LNM percentage in an orthotopic model. Two-tailed $t$ test and Pearson's correlation test ($n = 6$/group). **j** Representative images of OXTR knockdown in fibroblasts validated by double-staining IHC analysis in wild-type or CKO (*Oxtr*^fl/fl *S100a4*^cre) mice. **k** The tumor volume, **l** LNM percentage and (**m**) desmoplastic stroma components were measured and summarized graphically. $P =$ two-tailed $t$ test. Scale bars, 50 μm (**j**) and 200 μm (**l**). Results are shown as mean and standard deviation (SD). Source data are provided as a Source data file.

impact on patient survival, even in patients of the WPOI 4−5 type, with high OXTR expression (Fig. 8i and Supplementary Fig. 6f, g). Furthermore, OXTR expression and nuclear ERK5 activity did not correlate with OXT concentrations in several OSCC cohorts (Fig. 8j). Although OXT/AVP could activate ERK1/2 or AKT as previously reported[51] (Supplementary Fig. 6h), neither OXT nor AVP affected ECM-remodeling ability (Supplementary Fig. 6i) and the pro-invasion of CAFs in organotypic invasion assays (Fig. 8k, l). These findings

indicate that the development of WPOI type in OSCC patients is independent of peripheral OXT/AVP levels.

## Discussion

Currently, the underlying cellular and molecular events behind WPOI type and how these are responsible for the poor clinical outcomes of OSCC patients are unknown. Studies investigating the invasive nature of tumor cells in WPOI-5 type have found that apolipoprotein E (APOE)

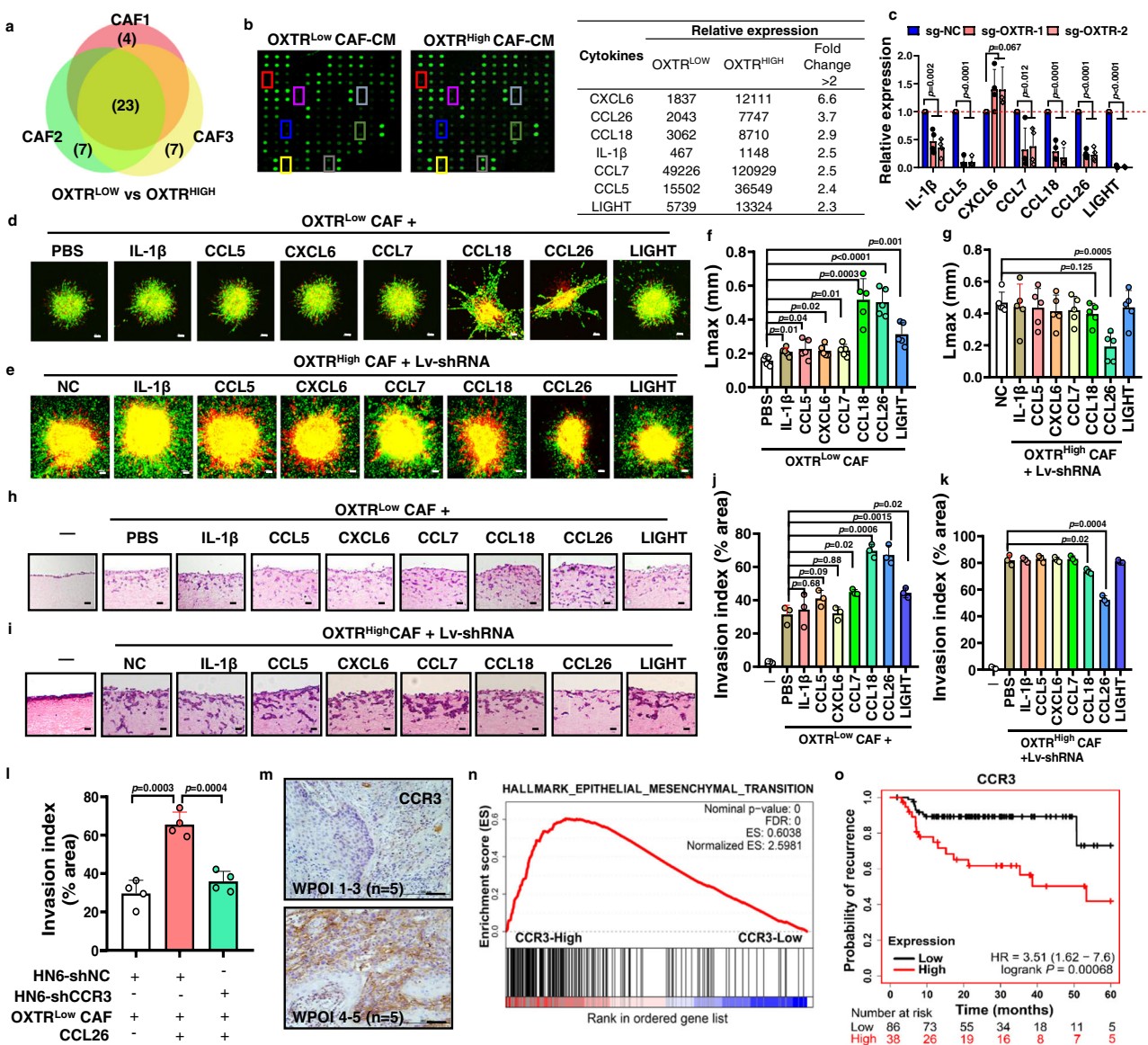

**Fig. 5 | CCL26 is required for OXTR^High CAFs function. a** Cytokines secreted from OXTR^Low and OXTR^High CAFs sorted from three individual CAF cell lines (CAF1-3), displayed as a venn diagram, with the numbers of overlapping cytokines indicated (left image). The numbers of shared cytokines elevated in all OXTR^High CAFs are indicated (n = 23). **b** The scan map of Human Cytokine array (middle) (n = 3 independent samples for each group). Table (right) displays seven cytokines/chemokines which were significantly differently upregulated (fold change >2) in the media of OXTR^High CAFs. **c** Relative expression (RT-PCR) of seven cytokines/chemokines after OXTR knockdown in OXTR^High CAFs. P = multiple t test. n = 4 for each group. **d, e** Representative images of DiI-labeled conditional CAFs (red) co-cultured with CMFDA-labeled HN6 (green) to form heterotypic spheroids for organoid invasion studies. Cultured cells were supplemented with recombinant human CXCL6, CCL26, CCL18, IL-1β, CCL7, CCL5, or LIGHT or stable knockdown of each of seven cytokines/chemokines. Scale bar: 100 μm. **f, g** Graphical representation of maximal distance of spheroid invasion from the spheroid border (Lmax) in (d, e) and

quantification using ImageJ software, n = 5/group. P = two-tailed t test. **h, i** Representative H&E images of 3D organotypic cultures of CAFs/tumor cells under the conditions indicated and used to estimate the ability of tumor invasion, displayed graphically. Scale bar: 100 μm. **j, k** The invasive index was calculated using ImageJ software. n = 3/group, P = two-tailed t test. **l** Graphical presentation of the invasion index of HN6 OSCC cells with CCR3 knockdown co-cultured in the 3D Organotypic system used in (h) with OXTR^Low CAFs and CCL26, n = 4/group. P = two-tailed t test. **m** IHC analysis for CCR3 expression in OSCC tumor microenvironment. Scale bars, 200 μm (n = 10). Scale bar: 100 μm. **n** Graphical representation of the correlation between EMT pathways from a GSEA analysis of CCR3^High versus CCR3^Low HNSCC patients, n = 522 (TCGA, Firehose Legacy). **o** The correlation between CCR3 expression and recurrence-free survival in HNSCC patients (n = 124) using Kaplan–Meier plotter (http://kmplot.com/analysis/). Results are shown as mean and standard deviation (SD) of three independent experiments. Source data are provided as a Source data file.

is overexpressed in tumor cells and that *APOE* knockdown inhibited both matrix degradation and the number of mature invadopodia in vitro[9]. The protein NEDD9 has also been shown to be more highly expressed in WPOI-5 tissues compared with WPOI-3 tumors, although the significance of this finding was not explored further[10]. In this study, we focused on the stromal microenvironment of the highest risk OSCC with WPOI 4−5 type. We report the discovery of a phenotypically and functionally stromal fibroblast subpopulation accompanied by

activated OXTR/ERK5 signaling in tumors with WPOI 4−5 type. However, the role of OXT/OXTR signaling in different cancers seems to be inconsistent. OXTR signaling was found to inhibit the development of metastatic colorectal cancer[52] and ovarian cancer[53]. Hypothalamic OXT-producing neurons was also found to suppress colitis-associated colorectal cancer[22]. However, OXT/OXTR signaling had been demonstrated to be pro-tumorigenic in lung cancer[23], melanoma[24], prostate cancer[54], and pancreatic cancer[55]. Our findings provide additional

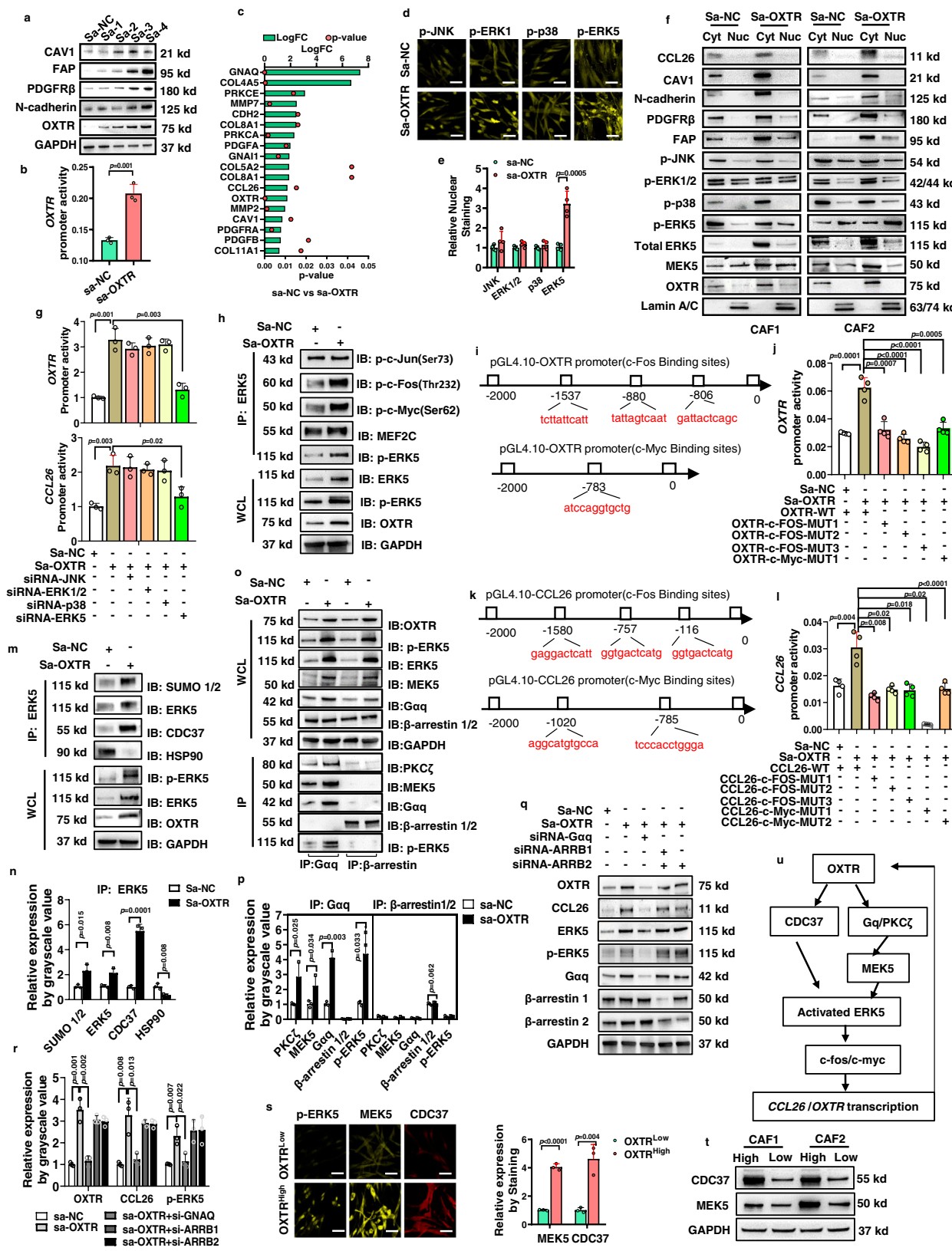

mechanistic insights into the pro-metastasis role of OXTR in OSCC development.

OXT is synthesized by nerve cells and is released into the blood to regulate digestion, blood pressure, heart rate, pain, etc., which are often dysregulated during carcinogenesis[56,57]. Moreover, cancer patients have elevated psychological stress[58], which can activate OXT

neuronal activity to induce OXT secretion in blood. Indeed, we found higher serum OXT level in the OSCC cohorts, which may due to tumor stress, a condition associated with all OSCC patients. Unexpectedly, the OXTR ligands OXT or AVP were absent in cell culture medium in vitro and additionally dispensable for the pro-invasion function of OXTR[High] CAF. Most pertinently, even in patients with WPOI 4–5 group,

**Fig. 6 | OXTR/ERK5 signaling maintains OXTR and CCL26 expression levels in OXTR$^{High}$ CAF. a** saRNAs targeting the human *OXTR* gene promoter activate CAF markers and OXTR levels, and (**b**) graphical representation of *OXTR* promoter transcriptional activity. *P* = two-tailed *t* test. *n* = 3. **c** Results of ATAC−seq assay to assess chromatin accessibility of CAF activation and OXTR signaling-related genes in sa-NC and sa-OXTR groups of CAF1 for three independent experiments. **d** Representative IF staining and (**e**) quantification for four p-MAPK kinases in situ in Sa-NC or Sa-OXTR CAFs. Scale bar: 50 μm. *P* = multiple t test. *n* = 4. **f** Representative western blots of cytoplasmic and nuclear fractions of Sa-OXTR CAFs. *n* = 3 biologically independent experiments. **g** Graphical analyses of the promoter activity of *OXTR/CCL26* of CAFs with sa-NC or sa-OXTR. *P* = two-tailed *t* test. *n* = 3. **h** Representative western blots of CAFs with sa-NC or sa-OXTR for endogenous ERK5 in in whole-cell lysates (WCL) or after immunoprecipitation with anti-ERK5 antibody for the proteins indicated. Four TFs in whole-cell lysates and IP were analyzed. **i**−**l** Mutations of potential binding sites of c-fos, c-myc on *OXTR* and *CCL26* promoter and its transcription activity were analyzed by a dual-luciferase reporter assay system. *P* = multiple *t* test. *n* = 3. **m** Representative western blots and (**n**) quantification for the indicated proteins from CAFs lysates with or without sa-NC or sa-OXTR in WC) or immunoprecipitation with ERK5 for the CDC37/HSP90/ERK5 complex. *P* = multiple *t* test. *n* = 3. **o** Representative western blots and (**p**) quantification for the PKC/MEK5 signal complex for endogenous Gαq or β-arrestin in WCL or immunoprecipitation with Gαq or β-arrestin antibodies. *n* = 3. **q** Representative western blots and **r** quantification for knockdown of Gαq or β-arrestin by siRNA and determined the p-ERK5/OXTR/CCL26 levels. *P* = multiple *t* test. *n* = 3. **s** Representative (*n* = 3) IF images (scale bar: 50 μm) and (**t**) quantification of nuclear ERK5, MEK5 or CDC37 in sorted OXTR$^{High}$ or OXTR$^{Low}$ CAFs. *P* = two-tailed *t* test. **u** The OXTR signaling pathway diagram. Results are shown as mean and standard deviation (SD). All immunoblotting results are representative of three independent experiments. Source data are provided as a Source data file.

which harbored high OXTR expression, OXT level still did not impact the prognosis of this patient population. This suggests that OXTR and its ligand might not co-express or signal together. Similarly, in the absence of ligands, high membrane concentrations of estrogen receptor, androgen receptor or epidermal growth factor receptor (EGFR) have also been identified to induce autonomous receptor signaling activation in several cancer types[59,60]. Thus, targeting the downstream signals of OXTR instead of blocking OXT/OXTR interaction is feasible and may be a more promising mechanism to block its function for beneficial therapeutic outcomes.

We here knocked down Gαq or ERK5 by shRNA to identified the role of Gαq/PKCζ/ERK5 pathway in the invasive tumor phenotype of OXTR$^{High}$ CAFs. Similarly, in cardiac fibroblasts, stimulation of Gαq-coupled GPCR by angiotensin II also found to promote ERK5 activation via PKCζ, and activation of MEK5 signaling was inhibited in PKCζ-deficient mice[61]. Further work in Gαq-deficient mice are warranted to clarify the role of Gαq/PKCζ/MEK5 complexes in OXTR/ERK5 activation. Aberrant nuclear ERK5 expression is associated with tumor progression, inhibition of ERK5 has been shown to have therapeutic potential in both cancer and inflammatory diseases, prompting the development of ERK5 kinase inhibitors (ERK5is)[62,63]. However, there are many safety concerns still to be addressed. In addition to off-target effects (either on BRD4 or other kinases), ERK5i including XMD8-92, bind to the ERK5 kinase domain and paradoxically promote nuclear localization signal (NLS) and transcriptional activation domain (TAD) exposure. This in turn promotes nuclear localization and paradoxical activation of ERK5 transcriptional activity, resulting in the failure to control tumor progression[49,64]. Our findings also cemented this paradoxical activation by ERK5i and we found treatment with XMD8-92 was not able to effectively reduce tumor metastasis but ERK5 knockdown itself proved to be an efficient strategy to mitigate OSCC metastasis.

In summary, this study highlights the pro-invasion nature of the previously unrecognized OXTR$^{High}$ stroma fibroblast in WPOI 4−5 type tumor (Fig. 8m). OXTR$^{High}$ CAFs are rich in the WPOI 4−5 type stroma and promote ECM remodeling through ERK5 signaling, which, in turn maintains OXTR and CCL26 expression, correlating with an invasive tumor phenotype. Development of efficient strategies to knock down ERK5 promises to overcome ERK5 activation-induced tumor progression in WPOI 4−5 type OSCC.

## Methods
Our research complies with all relevant ethical regulations; Ethical approval for this study, including tumor biopsy and serum collection, and patients informed consent was obtained in accordance with the guidelines and policies of the Research Ethics Committee of Nanjing Stomatology Hospital (No. 2019NL-009(KS)). The study was conducted in accordance with the criteria set by the Declaration of Helsinki. All animal experiments were performed in accordance with

Jiangsu Association for Laboratory Animal Science (Authorization Number: 220195073) and were subject to review by the animal welfare and ethical review board of Nanjing university.

### Reagents and cell lines
Detailed information relating to the antibodies and dilutions, reagents, oligonucleotides, and cell lines used in this study is provided in Supplementary Tables 1–3 and Reporting Summary. Primer sequences are provided in Supplementary Table 2. SiRNA sequences are provided in Supplementary Table 3. The OSCC cell lines were professionally verified and free from contamination. No presence of mycoplasma was found. Short tandem-repeat analysis for DNA fingerprinting was also used to verify the cell lines.

### Human subjects
All patients in this study (No. 2019NL-009(KS)) diagnosed with primary OSCC were confirmed by hematoxylin and eosin staining of tumor biopsy and scoring by experienced pathologists from the Department of Pathology at Nanjing Stomatology Hospital. To determine the expression of OXTR, ERK5, CCL26, and EMT phenotype, OSCC tissues were evaluated according to the WHO classification and International Cancer Control (UICC) tumor−node−metastasis (TNM) staging system. To determine serum OXT and AVP levels in healthy people, OSCC patients and leukoplakia patients, peripheral blood obtained and was used to isolate serum, which was stored at −80 °C until required. Death records were complete until March 1, 2020, the censor date. Patients diagnosed with autoimmune or other malignant diseases as well as pregnant or lactating individuals were excluded from this study. No patient included in this study had undergone preoperative chemotherapy and/or radiotherapy interventions.

### Mouse models
NCG triple immunodeficient mice, lacking T, B, and NK cells (NOD/ShiLtJGpt-Prkdc$^{em26Cd52}$Il2rg$^{em26Cd22}$/Gpt, from GemPharmatech Co. Ltd., Nanjing, China) were used (both genders) for the orthotopic xenograft at 4–6 weeks of age and housed in ultraclean barrier facilities. *Fsp1-Cre* mice have been previously documented[65]. The *Oxtr$^{flox/flox}$* mouse strain was established in the Genetically Engineered Mouse Facility at GemPharmatech Co. Ltd. (Nanjing, China). The *Oxtr$^{fl/fl}$ S100a4$^{cre}$* mouse strain was generated by crossing the *Fsp1-Cre* (B6/JGpt-Tg(S100a4-Cre-PolyA)3/Gpt) and *Oxtr$^{flox/flox}$* mouse strains (B6/JGpt-Oxtr$^{em1Cflox}$/Gpt) with established genotypic and phenotypic characterization. The *Oxtr* gene has two transcripts, and according to the structure of *Oxtr* gene, exon1 of *Oxtr-201* (ENSMUST00000053306.7) transcript containing a start codon ATG was recommended as the knockout region for CRISPR/Cas9 gene editing technology resulting in the disruption of protein function. Loss of function of the *Oxtr* target gene in specific tissues and cell types was observed after mating with *Fsp1-Cre* mice expressing Cre recombinase.

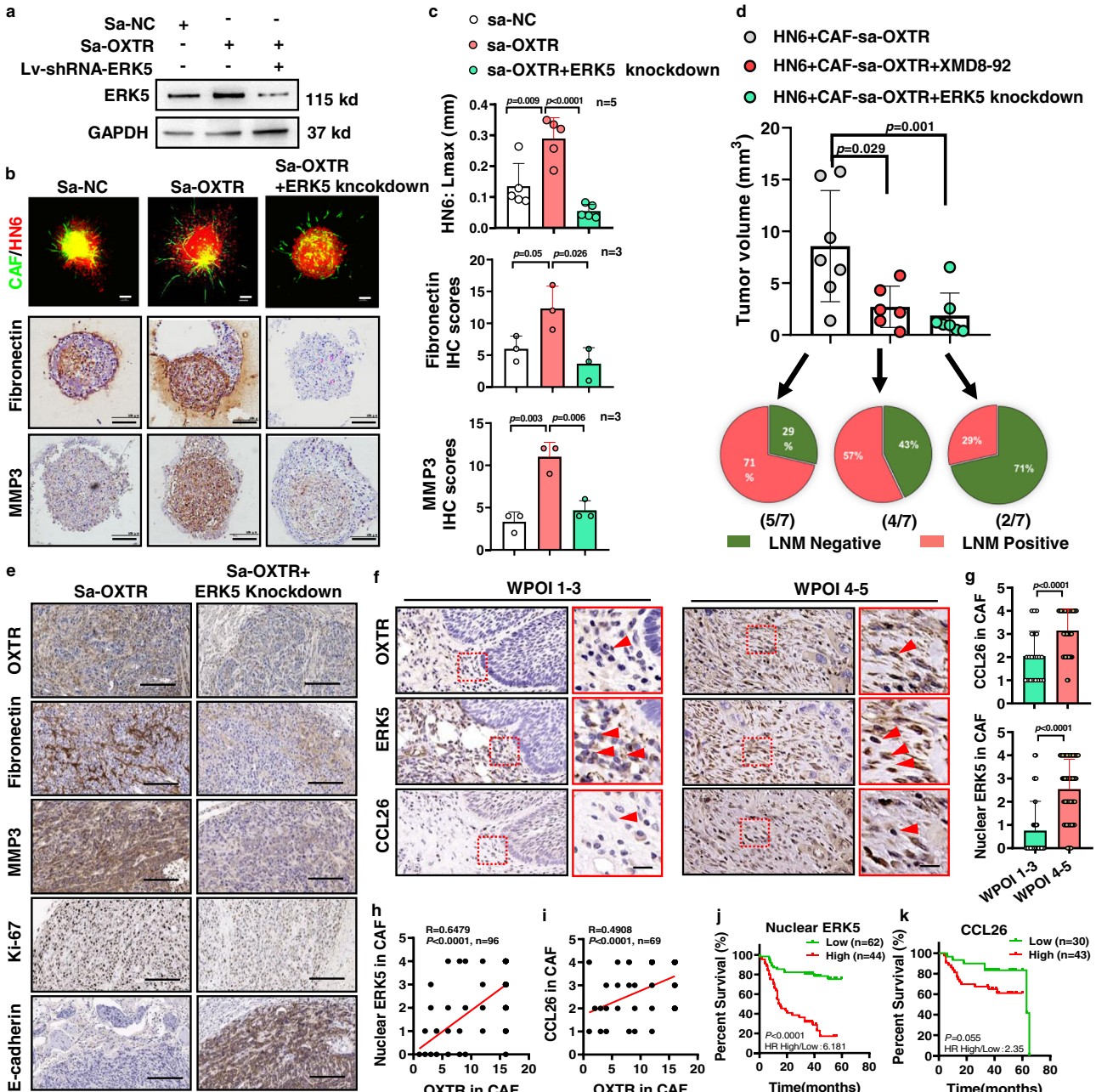

**Fig. 7 | ERK5 inhibition in CAFs is capable of suppressing OSCC tumor progression. a** Stable knockdown of ERK5 in CAFs overexpressing OXTR (representative of *n* = 3 biologically independent experiments) and (**b**, **c**) its impacts on tumor invasion, fibronectin, and MMP3 levels. Representative images of conditional CAFs (green) co-cultured with HN6 (red) to form heterotypic spheroids embedded in conditional 3D collagen matrix for organoid invasion studies (*n* = 5 biologically independent samples). Spheroids were then embedded in paraffin for histochemistry analyses by two-tailed *t* test (*n* = 3 biologically independent samples). Scale bar: 100 μm. **d** Graphical analysis of tongue tumor volume or LMN percentage after 2 × 10⁵ HN6 OSCC cells were mixed with 1 × 10⁶ sa-OXTR CAFs and injected into the tongue of 4- to 5-week-old NCG mice with either XMD8-92 treatment (50 mg/kg, twice a week) or CAFs with ERK5 knockdown. *n* = 7/group, *P* = two-tailed *t* test or Pearson's correlation test. **e** Representative IHC images from OSCC tumor tissues of CAFs with sa-OXTR or sa-OXTR with ERK5 knockdown stained with antibodies for OXTR, fibronectin, MMP3, Ki-67, or E-cadherin in tumor tissues. *n* = 3. Scale bars, 200 μm. **f** Representative IHC images showing nuclear ERK5 (*n* = 107) or CCL26 (*n* = 74) expression in OSCC patient tissue and **g** their graphical quantification. ERK5 (WPOI 1–3: *n* = 32 and WPOI 4–5: *n* = 71) and CCL26 (WPOI 1–3: *n* = 21 and WPOI 4–5: *n* = 48). *P* = two-tailed *t* test. Tumors were categorized according to WPOI stage. *P* = Student's *t* test. Scale bars, 20 μm. **h**, **i** Graphical representation of correlative indexes between nuclear ERK5 (*n* = 96) or CCL26 (*n* = 69) and OXTR analyzed by spearman rank correlation test. **j**, **k** Graphs of OSCC patient postoperative survival and their correlation with nuclear ERK5 (*n* = 106) or CCL26 (*n* = 73). *P* = Log-rank (Mantel–Cox) test. Results are shown as mean and standard deviation (SD). Source data are provided as a Source data file.

Mice were sacrificed by cervical dislocation at the indicated time points or when the largest tumor exceeded 1500 mm³ in subcutaneous models or 40 mm³ in an orthotopic model. In some cases, this limit has been exceeded the last day of measurement, and the mice were immediately euthanized. All animal experiments were carried out following animal protocols approved by the Laboratory Animal Welfare and Ethics Committee of Nanjing university. The housing conditions for the mice were as follows: 12 light/12 dark cycle; the temperature was 72 degrees Fahrenheit; and humidity was 40–50%.

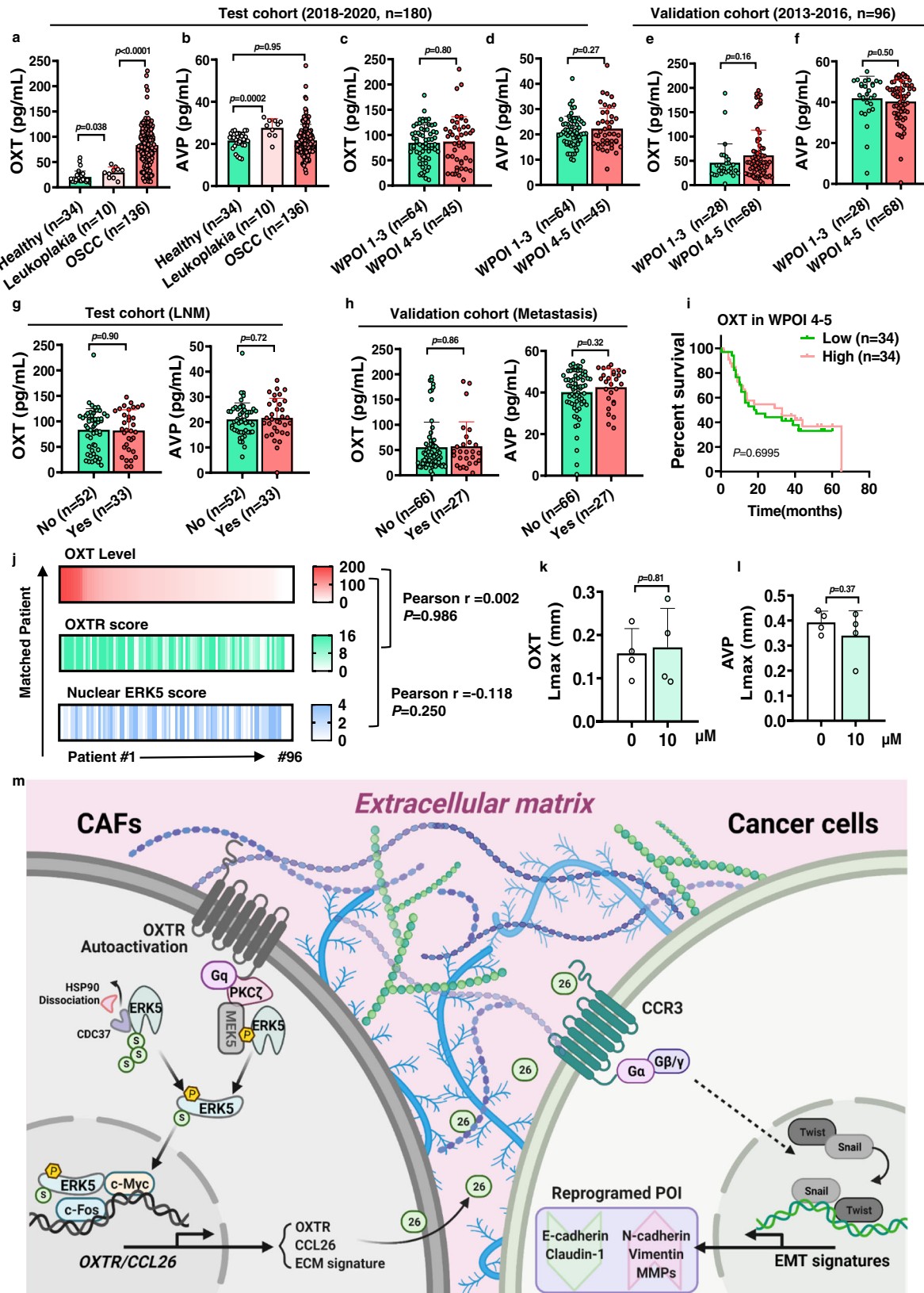

## Immunohistochemistry, immunofluorescence, and fluorescence in situ hybridization

Immunohistochemistry (IHC) and immunofluorescence (IF) assays were performed. All procedures were performed on 5-μm sections of human or mouse tissues. Briefly, for IHC staining of OSCC samples, murine tumor or tumor organoids, sections were deparaffinized, and followed by antigen retrieval, followed by blocking with 3% bovine serum albumin, and incubation with indicated primary antibodies; cytokeratin (CK), α-SMA, CD4, CD8, Foxp3, CD68, CD31, CD146, Collagen I, OXTR, MMP3, MMP9, N-cadherin, E-cadherin, fibronectin, Ki-67, ERK5, or CCL26 overnight at 4 °C. For double IHC on mouse tissue, anti-mouse IgG-HRP (brown), anti-rabbit IgG-AP (Red) were used

**Fig. 8 | WPOI progression is not associated with the serum levels of the OXTR ligand.** Graphical representation of analysis of serum (**a**) OXT and (**b**) AVP levels from healthy, leukoplakia and (**c**, **d**) OSCC patients with WPOI 1–3 ($n = 64$) and 4–5 ($n = 45$) types as indicated and as determined by ELISA ($n = 180$). $P$ = two-tailed $t$ test. **e**, **f** Graphical representation of analysis of serum OXT and AVP levels from patients with WPOI 1–3 ($n = 28$) and 4–5 ($n = 68$) types in a validation group. $P$ = two-tailed $t$ test. **g**, **h** Graphical representation of analysis of OXT/AVP levels from patients with/without LNM ($n = 85$) and postoperative metastasis ($n = 93$). $P$ = two-tailed $t$ test. **i** Graphical representation of survival data from WPOI 4–5 OSCC patients stratified according to OXT level (low: $n = 34$ and high: $n = 34$). **j** Correlative index of OXTR, nuclear ERK5 and serum OXT levels from OSCC patient tissue ($n = 96$) by IHC and ELISA analysis using Pearson's correlation test. **k**, **l** Graphical analysis of Lmax (invasion in 3D collagen matrix) of HN6 in heterotypic spheroids treated with OXT or AVP. $n = 4$/group, $P$ = two-tailed $t$ test. **m** Proposed working model depicting the mechanism of OXTR autoactivation-mediated ERK5 nucleus translocation for maintaining the function and phenotype of OXTR[High] CAFs in WPOI 4–5 type stroma. Results are shown as mean and standard deviation (SD). Source data are provided as a Source data file.

according to the manufacturer's protocol (#DS-0003, Beijing Zhong Shan-Golden Bridge Biological Technology Co., Ltd.). For IF analysis of EMT phenotype, CAFs ratio and MAPK pathway activation, frozen sections or cells were collected and fixed with 4% formaldehyde for 15 min then permeabilized with 100% methanol, followed by incubation with primary antibodies overnight. Following washing three times in PBS, coverslips were incubated in fluorochrome-conjugated secondary antibody (donkey anti-mouse/rabbit Alexa Fluor Plus 488/594, dilution: 1:400) for 1–2 h at room temperature shielded from light followed DAPI staining (Bioword, China). The FISH assay (*ACTA2*-cy3, red and *OXTR*-fam, green) was performed according to the manufacturer's protocol (Shanghai GenePharma Co., Ltd, China), U6 and 18 S were used for cytoplasmic and nuclear positive controls, respectively. Finally, coverslips were mounted onto glass slides with neutral gum and observed using FV10i confocal microscope (OLYMPUS, Japan).

IHC and IF staining was quantified by two independent pathologists who were blinded to the clinical data. The median value was adopted as cut-off value for further analysis. The proportion scores of α-SMA, CD4, CD8, Foxp3, CD68, CD31, CD146, Ki-67, CCL26, nuclear ERK5 were defined as 0 = <5%; 1 = 6–25%; 2 = 26–50%; 3 = 51–75%; and 4 = >75% positive cells. Staining scores of Claudin-1, Vimentin, Collagen I, OXTR, MMP2, MMP3, MMP9, N-cadherin, E-cadherin, fibronectin were defined by multiplying the staining proportion and intensity; intensity was defined as 1 = negative staining, 2 = weak staining; 3 = moderate staining; and 4 = strong staining. Where indicated, low/moderate/high staining were defined as 0–25% percentile, 25% percentile–75% percentile, 75% percentile–maximum.

**OSCC patient-derived CAF cell line generation and identification**
Primary cancer-associated fibroblast (CAF) lines were derived from OSCC tumor tissue[66]. Briefly, sterile fresh OSCC tissues and its corresponding normal tissues were collected from surgery and washed with phosphate-buffered solution (PBS) and antibiotics, and then removed the epithelial and adipose tissues. The specimens were cut into small pieces and digested by enzyme mix, including 0.2% Collagenase II/IV (#BS033A/035A, Biosharp), 0.08% Neutral protease (#J10050, Shanghai Lanji), 0.0032% Hyaluronidase (#H3506, Sigma) for 30 min. The remaining small tissues were plated into cell culture flasks with DMEM/F12 basic medium and incubated at 37 °C. The medium was replaced every 2–3 days. Because epithelial cells and fibroblasts have different tolerance to 0.25% trypsinase (#25200114, Sigma), the epithelial cells were easily removed via trypsinase with careful observation under a phase-contrast microscope, the remaining cells were fibroblasts and were collected by further digestion with trypsinase for 2 min. Suspended epithelial cells after digestion would not adhere to the wall. CAFs were immortalized for continual culturing by infecting with hTERT (pBABE-hygro-hTERT). Primary CAFs and corresponding immortalized CAFs were also used for chromosome karyotype analysis and short tandem-repeat (STR) analysis. The karyotype of immortalized fibroblasts retained the basic characteristics of normal cells without malignant transformation. STR detection and identification results showed that immortalized fibroblasts were primary cells.

Data acquired for the CAF[WPOI 1-3] group was labeled, according to acquisition time, with #11.9 and #8.30 CAFs in further study. In the CAF[WPOI 4-5] group, #7.4 and #8.9 CAFs were used as CAF[WPOI 4-5] in further study. For each independent experiment, the mixing of different CAFs was not permitted. The data relating to #7.4 and #8.30 CAFs were used in this manuscript, with data relating to #11.9 and #8.9 CAFs used to validate the key findings. For example, in Fig. 6h, #11.9 and #8.30 CAFs were used and labeled as CAF1/CAF2. Due to space restrictions, data from CAF1 only is shown in Fig. 6.

**Flow cytometry and cell sorting**
All antibodies used for flow cytometry were purchased as fluorescent dye conjugates, with the exception of the anti-human OXTR antibody. An APC-antibody conjugation Kit (#FMS-ABAPC0002) or PE-antibody conjugation Kit (#FMS-ABPE0001) were used to conjugate human OXTR antibody for flow cytometry detection. To compare CAF phenotypes in different groups, cells were resuspended in PBS containing 1% FBS and stained with fluorescent-conjugated antibodies to detect CD29, FAP, or PDGFR-β. For cell sorting in OSCC tissues, single-cell suspensions were prepared from OSCC tissues via brief trypsinization. This was followed by forward and side scatter flow cytometry gating to exclude cell debris and dead cells by DAPI-positive staining (Thermo Fisher Scientific; D1306). Cells were stained with fluorescent-conjugated antibodies to detect EpCAM, CD45, CD31, PDGFR-β or FAP-positive cell subpopulations. Sorted populations were processed for RNA extraction and qRT-PCR. To facilitate isolation of OXTR high or low-expressing CAFs, live single cells were labeled with APC/PE-conjugated anti-OXTR and sorted by flow cytometry according to the OXTR fluorescence intensity, which was verified at the protein level by western blot.

**ECM-remodeling assay**
To estimate force-mediated matrix remodeling, $1 \times 10^6$ CAFs were embedded in 0.2 ml of collagen I/Matrigel mixture and seeded into the upper chamber of a 24-well Millicell hanging cell culture plate (#PIEP12R48, 8μm). Once the gel was set, cells were maintained in fibroblast medium (#2301, Sciencell, USA). Gel contraction was monitored by daily imaging. The relative diameter of each well and gel were measured and analyzed using ImageJ software. To estimate the gel contraction value, the percentage of contraction was calculated using the formula 100 × (well area – gel area)/well area.

**Organoid culture systems for cell growth and invasion assays**
Several methods and models of 3D invasion and proliferation were used. The method used for Figs. 2h, 4o, and 5k, required five volumes of collagen I/Matrigel (collagen I: Matrigel = 1:1), mixed with five volumes of 1× DMEM/FBS containing conditional fibroblasts ($2 \times 10^6$ allowed to set at 37 °C in the upper chamber of 24-well transwell plate. OSCC cell lines ($2 \times 10^5$) were seeded atop the gel mixture and after 24-h incubation, cancer cells were exposed to the environment by removing surface media. The gel was then supplemented with complete medium from below, which was changed daily. After 14 days, the cultured organoid matrices were fixed by 10% formaldehyde and embedded in paraffin for histochemistry analyses of cancer cell invasion. The relative invasion areas and total areas of organoid matrices were measured to obtain invasive index using ImageJ software as per the formula: Invasive index= invasion area/total area × 100%. The

method used for Figs. 4f, h, 5g, h, and 7b, required $8 \times 10^4$ DiI-labeled CAFs and $2 \times 10^4$ CMFDA-labeled OSCC cells cultured to form homospheroids or heterotypic organoids. Stable heterotypic spheroids formed after 2 days' culture were embedded into Matrigel gels and mixed with collagen I for 5 days. The matrix invasion of CAFs homospheroids or CAF/HN6 heterotypic organoids were then determined. Data were expressed as the maximal distance of invasion from the spheroid border (Lmax), estimated using ImageJ software. To gauge organoid growth of the OSCC cells in Fig. 2h, $2 \times 10^5$ OSCC cell lines were seeded into the gel mixture in transwells and cultured for 7–10 days. The organotypic matrices were then processed for frozen sections and IF analysis. To estimate the growth of CAF/HN6-derived heterotypic spheroids in Fig. 4d, $6 \times 10^4$ HN6 were mixed with $6 \times 10^4$ indicator CAFs in an ultra-low adhesive environment. Stable heterotypic spheroids formed after two days of culture and after 10 days were collected for IF staining and organoid diameter analysis using ImageJ software.

### Orthotopic tongue xenograft model
To generate an orthotopic xenograft tongue tumor model, $2 \times 10^5$ HN6 OSCC cells mixed with $1 \times 10^6$ conditional CAFs or $5 \times 10^4$ SCC7 cells were suspended in 20 μL of PBS/Matrigel (3:1) and injected into the anterior portion of the tongue of 4- to 5-week-old NCG or $Oxtr^{fl/fl}$ $S100a4^{cre}$ mice using a syringe with a 30 gauge needle (BD Biosciences). On days 7–10 after injection, mice were randomly divided into different treatment groups. Mice receiving XMD8-92 were treated twice weekly (50 mg/kg). OXTR^high CAFs with CCL26 knockdown by lentivirus-shRNA were also used in the orthotopic xenograft tongue tumor model to confirm the role of CCL26 in tumor cell/CAF interaction.

### RNA sequencing analysis
RNA-seq was performed on WPOI 1–3 and WPOI 4–5 tissue samples with the help of NovelBio Bio-Pharm Technology Co., Ltd (Shanghai, China).

### CRISPR/Cas9 editing of *OXTR* and validation (T7E1) assays
To knock out *OXTR*, three sgRNA-OXTR were designed and transfected into lenti-Cas9-Blast plasmids (OBiO Technology Co. Ltd., Shanghai, China) and then transfected into 293T cells to produce lentivirus. The virus was concentrated using a Lenti-X Concentrator (Takara Bio Inc.) and CAFs infection was performed. The extent of *OXTR* deficiency was determined by T7E1 CRISPR/Cas9 editing validation assays and DNA sequencing.

### Enzyme-linked immunosorbent assay
Serum from OSCC patients was collected from peripheral blood and stored at −80 °C. Serum levels of OXT and AVP were detected by commercial ELISA kits (Cusabio, #CSB-E09080h, #CSB-E08994h) according to the manufacturer's protocol.

### Human cytokine antibody array
The human Cytokine Antibody Arrays G5 kit (Raybiotech, # AAH-CYT-G5-8) was used to detect serum cytokines according to the manufacturer's instructions. The glass chip was scanned with a laser scanner (such as Innopsys' InnoScan®) using cy3 or "green" channel (excitation frequency = 532 nm).

### Small activating RNAs of *OXTR*
The design of four saRNAs for *OXTR* (−158-−138, −407-−387, −733-−713 and −871-−851) used in this study comply with the general design rules (Shanghai Genechem Co., Ltd., China)[67]. Four parameters were used: (1) targeting gene annotations from UCSC RefSeq database; (2) targeted sequence from antisense RNA; (3) promoter selection of antisense sequences; and (4) identification of candidate short activating RNAs.

### Luciferase reporter assays
Wild-type pOXTR-Luc, pCCL26-Luc and mutant pOXTR-Luc, pCCL26-Luc, renilla luciferase vectors (pRL-TK; Promega) were constructed based on the PGL3 basic vector (OBiO Technology Co., Ltd., Shanghai, China). These were transfected by using Lipofectamine 3000 (Invitrogen, Carlsbad, CA) according to the manufacturer's instructions and followed by the indicated treatments. Luciferase activity was determined after 48–72 h of transfection for using the Dual-Luciferase reporter assay system (Promega). Firefly luciferase activity was normalized to Renilla luciferase activity for each sample.

### Transposase-accessible chromatin (ATAC) assay using sequencing
ATAC–seq was performed using the omni-ATAC protocol, Briefly, 10,000 cells per replicate were pelleted and lysed in 50 μl lysis buffer (10 mM Tris-HCl pH 7.4, 10 mM NaCl, 3 mM MgCl₂) on ice and centrifuged at $500 \times g$ for 10 min at 4 °C. Following centrifugation, pelleted nuclei were resuspended in 50 μl of transposition reaction mix containing Tn5 transposase from Nextera DNA Library Prep Kit (Illumina) and incubated for 30 min at 37 °C with shaking at 1000 rpm. The transposase-associated DNA was purified using the MinElute PCR purification kit (QIAGEN) to enrich the sub- and mono-nucleosome DNA fragments. The library was amplified and the total amplified DNA size selected into fragments less than 800 bp using SPRI beads. ATAC–seq libraries were sequenced on an Illumina NovaSeq with more than 20 million reads per library. The raw reads were processed using the esATAC package. Computational analysis of ATAC–seq data was followed by BedTools and Intervene.

### Western blotting and co-immunoprecipitation
For immunoblotting analysis, conditional CAFs were harvested and protein was extracted using RIPA buffer (20 mM Tris-HCl, 37 mM NaCl2, 2 mM EDTA, 1% Triton-X, 10% glycerol, 0.1% SDS, and 0.5% sodium deoxycholate) with phosphatase and protease inhibitors. Total protein content was measured by bicinchoninic acid assay (Thermo Fisher Scientific). Total lysates were subjected to SDS-PAGE and transferred to a polyvinylidene difluoride (PVDF) membrane, blocked with 5% BSA in TBST (1% Tween 20, tris-buffered saline). The membrane was then immunodetected with the primary antibodies (key resources table), followed by incubation with corresponding HRP-linked secondary antibodies (key resources table) and visualization by enhanced chemiluminescence assay (ECL, Thermo). For Co-IP analysis, IP antibodies or anti-IgG isotype were mixed with protein A/G magnetic beads for 1 h at room temperature and then cultured with cell lysates for a further 1 h at room temperature, followed by separation using DynaMag™-2 magnets (Thermo Fisher Scientific, # 12321D). Beads were then washed five times with lysis buffer and eluted with 20 μL of 1× SDS sample buffer prior to immunoblotting analysis.

### Regulatory motif analysis
The predicted transcriptional factors for *OXTR* and *CCL26* were analyzed by GTRD database analysis (http://gtrd.biouml.org/). JASPAR database (http://jaspar.genereg.net/) was used to obtain the DNA-binding motifs for c-fos and c-myc in the promoter of *OXTR* and *CCL26*, modeled as matrices.

### Statistics and reproducibility
The statistical tests for each figure are indicated in the figure legend. All samples represent biological replicates. Data are presented as the mean ± SD. For in vitro and in vivo analyses, experiments were repeated independently three and two times, respectively, with similar results to demonstrate reproducibility. Statistical analysis was performed using GraphPad Prism 8 and the SPSS software program (SPSS Inc). When numbers permitted, values were tested for Gaussian distribution using Kolmogorov–Smirnov tests. *T* test, Mann–Whitney *U*

test, ANOVA, and Kruskal–Wallis test were used to compare two groups or the differences between more than two groups, respectively. Survival analysis included overall survival (OS) and recurrence-free survival (DFS), evaluated by Kaplan–Meier and log-rank test. The independent prognostic significance was determined by Cox proportional hazard regression model via analysis multivariate hazards ratio (HR) and 95% confidence interval (CI). Where indicated, individual *P* values are shown.

### Reporting summary
Further information on research design is available in the Nature Research Reporting Summary linked to this article.

## Data availability
The mRNA expression profile data and ATAC reported generated in this study have been deposited in the Sequence Read Archive (SRA) database with accession numbers PRJNA741552, PRJNA741553, and PRJNA741554. The publicly available data in Figs. 2c and 4a (TCGA, Firehose Legacy) are available in the cBioPortal database [http://www.cbioportal.org/study/summary?id=hnsc_tcga] and are used to perform GSEA analysis by MSigDB database [http://www.broadinstitute.org/gsea/msigdb/index.jsp]. The publicly available data in Fig. 5o and supplementary Fig. 4g–i are available in the KM-Plotter pan-Cancer database (https://kmplot.com/analysis/). The publicly available data in supplementary Fig. 4e is available in TIMER 2.0 database (http://timer.cistrome.org/). The publicly available data in Figs. 2d, 2i, supplementary Fig. 2g–j is available in the GEO database with accession numbers GSE68241, GSE81850, GSE103322, GSE154778, and GSE129455. The remaining data are available within the Article, Supplementary Information or Source Data file Source data are provided with this paper.

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

## Acknowledgements

We thank Xuan Zhou, a member of Shanghai OE Biotech. Co., Ltd, for his helpful and essential bioinformatics analysis. The research was supported by grants by the National Natural Science Foundation of China (Grant No. 81902754 to L.D., 82173159 to Y.H.N.); Natural Science Foundation of Jiangsu Province (No. BK20190304 to L.D., BE2020628 to Y.H.N.); Nanjing Medical Science and Technology Development Foundation, Nanjing Department of Health (No. YKK21182 to L.D., YKK20151 to Y.X.S.).

## Author contributions

Conceptualization: L.D. and Y.H.N.; data curation: L.D.; formal analysis: L.D.; methodology: L.D., Y.F., N.S.Z., and M.X.Z.; investigation: L.D., Y.F., N.S.Z., Z.D. X.X.Z., and M.X.Z.; human OSCC sections and approvals: X.F.H. and S.C.; cell sequencing analysis: Q.Z., Y.X.S., and Y.J.; supervision: L.D., Y.H.N., and Q.G.H.; validation: Y.H.N.; writing—original draft: L.D.; writing—review & editing: L.D., L.O.R., and J.S. provided intellectual advice, helped interpret data and co-wrote the manuscript.

## Competing interests

The authors declare no competing interests.
