## [Peer Review File · Nature Communications]

OXTRHigh stroma fibroblasts control the Invasion Pattern of Oral Squamous Cell Carcinoma via ERK5 SignalingReviewers' comments:

Reviewer #1 (Remarks to the Author): with expertise in OXT/OXTR signaling in cancer

The paper by Ding et al entitled "Tumor Microenvironment control the Invasion Pattern of Oral Squamous 7 Cell Carcinoma via auto-activating OXTR/ERK5 Signaling" describes a correlation between WPOI and recurrence-free and overall survival in OSCC patients. Overall the study is intriguing, and provides a novel insight into molecular mechanisms of the role of stroma in OSCC invasiveness and progression. Of note is the novel role of OXTR regulation of ERK5 in stromal fibroblasts. The study utilized three-dimensional (3D) organotypic invasion and growth models and xenograft and orthotopic transplantations in mice in their studies, and appears to have used an appropriate number of human samples. OXTR's role was identified through RNA sequencing. This manuscript can be accepted for publication after addressing the following concerns:

1. In many instances OXTR data was just mRNA expression. Where possible, including OXTR protein expression will provide additional important information about OXTR protein signaling.
2. The role of oxytocin (OXT) the physiologic ligand of OXTR needs more explanation in this context.
3. The authors suggest that the mentioned notion of autoactivated OXTR indicate that OXTR functions independent of OXT in this context? This needs to be further established and explained.
4. Given that the role of OXT and OXTR in carcinogenesis is understudied, authors should ensure that prior studies that have reported a role for OXT or OXTR in other solid organ cancers are referenced, for completeness and better context.

Reviewer #2 (Remarks to the Author): with expertise in OSCC

The manuscript by Ding et al report a new molecular mechanism associated with oral squamous cell carcinoma metastasis. They showed that cancer-associated fibroblasts characterized by high oxytocin receptor were the key stroma components contributing to local-regional metastasis. They found that OXTR^{High} CAFs ligand-independently induced desmoplastic stroma and that CCL26 was required for the invasive phenotype of CCR3⁺ tumors. Mechanistically, OXTR cells autonomously activated nuclear ERK5 transcription signaling to maintain high levels of OXTR and CCL26. They propose that activation of ERK5 plays a critical role in Oral squamous cell carcinoma metastasis. While these findings are interesting, these findings are not convincing. There are several major concerns.

- 1) In general, immunostaining quality in Figures 1 and 2 is poor.
- 2) The reviewer was not convinced that macrophage and other inflammatory cells were not associated with tumor metastasis which have been demonstrated in many studies.
- 3) The authors did not have a local metastasis model to test their hypothesis.
- 4) A large amount of studies have shown that IL-6, MCP-1 and other chemokines are associated with tumor metastasis. The reviewer was not convinced that only CCL26 play a critical role in oral cancer metastasis.

Reviewer #3 (Remarks to the Author): with expertise in ERK5 and cancer

This study characterized the role of cancer-associated fibroblasts (CAFs) in the invasion pattern of oral squamous cell carcinoma (OSCC). Specifically, the data indicated that tumors displaying the worst pattern of invasion comprised a high proportion of myofibroblastic CAFs exhibiting high oxytocin receptor (OXTR) expression. This cellular subpopulation contributes to the mesenchymal phenotype

of the tumor and the acceleration of local-regional metastasis through increased ERK5 signaling.

Overall opinion

I have to say that I am very impressed by the amount of data included in this paper. However, the overall lack of accuracy in the scientific interpretation of the results and attention to details give the impression that the findings are not convincing. This is compounded by the fact that there is insufficient evidence to firmly conclude that OXTR is expressed in CAFs, as there is no experimental demonstration that the OXTR antibody used for immunoblot or immunofluorescence analyses is specific (see also comment 6). Moreover, a key question that was not answered and could have made a difference is, if “OXTR^{High} CAFs promote tumor cell invasion in a ligand-independent manner” page 18 line 455, what is the mechanism of OXTR activation in CAFs? The scientific argument is also fairly poor given that many references provided do not support the statements on which the authors built the rationale for their study. Taken together, these major concerns significantly decreased my enthusiasm for supporting the publication of this study.

Major comments

1 - Page 9 line 217: The authors state that “The only component found to be significantly altered between WPOI 1-3 and WPOI 4-5 types were collagen I expressing stromal cells (Figure 2B), which were enriched in the WPOI 4-5 stroma, indicating a potential pro-invasion phenotype of Collagen I rich ECM microenvironments, as previously reported (Bhattacharjee et al., 2021; Chen et al., 2021)”. These two references do not support this statement as both studies proposed that myCAF-expressed type I collagen suppressed tumor growth. Specifically, Chen et al showed that targeted deletion of Col1 in myofibroblasts accelerated the emergence of PanINs and PDAC, while Bhattacharjee et al. proposed that myCAF-expressed type I collagen suppressed tumor growth by mechanically restraining tumor spread. The authors must clarify this issue.

2 - Fig. 3C: The authors must confirm that primary CAFsWPOI 4-5 exhibit lower expression of OXTR than CAFsWPOI 1-3, at protein level. The data should be presented next to panel C.

3 - Pages 12-13 (L306- L311): The rationale for choosing OXTR as opposed to the other candidates identified by transcriptomic analysis (e.g. other neuronal development and function-related genes or inflammation and cancer-related genes) is not explained. Why are RELN, OXTR and CAV3 in both lists?

4 - Page 13 line 317: “To date, the function of peripheral OXTR expression has been elusive, particularly during carcinogenesis (Cuneo et al., 2019)”. The reference provided here does not support this statement.

5 - Figure 3D and S2C: Weren't the authors surprised to find that OXTR mRNA level correlated poorly with neuronal cells (identified as S100B+ cells) given the strong expression of OXTR in the brain? Wouldn't this controversial observation invalidate their analysis?

6 - Page 18 line 466: The authors explained that OXTR is detected at 75 kDa, instead of the expected 45-66 kDa molecular weight, due to glycosylation of the N-terminus. However, they provide no experimental evidence to substantiate their hypothesis. Given the novelty of finding OXTR in CAFs, it is essential that the authors rule out the possibility that the band detected by the OXTR antibody is not specific. For example, they could show that ectopically expressed OXTR in CAFs is expressed as a 75 kDa protein.

7 - Page 23 line 612: The authors state: “Our analysis showed that CCL26 enhanced OXTR^{Low} CAFs-induced E-cadherin suppression via snail and twist, and up-regulated N-cadherin levels, which was significantly inhibited by CCR3 loss (Figure S4F)”. This is clearly a gross over-interpretation of the data. This conclusion must be rephrased to clearly and accurately describe the data presented in Figure S4F.

8 - Fig. 6 is overall very problematic: Initially, the authors searched for the mechanism underpinning OXTR and CCL26 induced expression through analyzing transcription factor (TF) binding sites in the OXTR and CCL26 promoters (panel E). Using this approach they propose that MAPK pathways are involved because the TFs orchestrating OXTR and CCL26 induced expression in CAFs are downstream targets of MAPK pathways. This would logically implied that MAPKs are responsible for controlling OXTR expression in CAFs. However, panels F, G and H showed that overexpression of OXTR via sa-OXTR promote ERK5 nuclear translocation, suggesting that ERK5 is activated by OXTR. This is very confusing. Moreover,

- In contrast to the authors' statement (page 28 line 726), panel G does not show that sa-OXTR targeting up-regulates MEK5 in CAFs.
- The quality of the immunoblot images presented in panels J and I is very poor and the conclusion that "increased levels of OXTR correlated with increased formation of a, ligand independent, Gαq, PKCζ and MEK5 complex, while simultaneously resulting in less association of β-arrestin with OXTR" (page 29 lines 743-746) is not supported by the data. There is no evidence that "OXTR/MEK5 interaction was mainly mediated by Gαq in CAFs (page 29 lines 746).
- The overall quality of the immunoblot presented in panel K is also very poor. siRNAp38 did not decrease p-p38 level and this is completely overlooked by the authors.
- Panel L: XMD8-92 is not a specific inhibitor of ERK5.
- Panel M: Increased MEK5 expression in OXTR^{High} CAF compared with OXTR^{Low} 759 CAFs should be confirmed by immunoblot analysis.
- Panel N: The differences between the intensities of the signals are too small to draw any conclusion.

Minor comments

- Page 6 line 159: The word "genotypically" is not correct in this context
- Page 7 line 183: "The question of whether any of the identified driver mutations actually initiates carcinogenesis asked us to question a primary role for tumor stromal components in multifocal and recurrent OSCC." This sentence must be rephrased.
- Legend of Fig. 2K-P on page 12 line 283: The authors must clarify the meaning of "conditional" referred to CAFWPOI 1-3 and CAFWPOI 4-5. The authors must also clearly indicate which HN6/hCAFs-derived tumor grafts, i.e. CAFWPOI 1-3 or CAFWPOI 4-5, are presented in the photographs (panel M (LMN- and LMN+) and analysed (panel N).
- Fig. 3B: The writing is too small to read the labelling of the genes.
- Page 14 line 366: In the sentence "GSVA results of HNSCC samples revealed that activated OXTR signaling", "of HNSCC samples" should be removed.
- Page 15 line 371: Should be "Figure S2I-J" instead of "Figure S3H-I".
- Page 18 line 461: "ablation of OXTR in cells edited cells, which effectively reduced expression of PDGFR-β and N-cadherin in OXTR^{High} CAFs (Figure 4L).
- Page 28 line 713: "Our Western blot analysis showed that sa-OXTR elevated cytoplasmic expression levels of activated CAF markers including N-cadherin, FAP, PDGFR-β, and CAV-1, CCL26 level, which were also enhanced by sa-OXTR.
- Fig 4: Panel I is too small to visualise efficient oxtr ablation in S100a4 positive cells. Moreover, the legend of panel T-W failed to clearly explain the difference between OXTR^{fl/fl} S100a4^{creW} and OXTR^{fl/fl} S100a4-creT samples.
- Page 28 line 733: What do the authors mean by OXTR can "directly activate MEK1/ERK1/2 in a G protein-independent manner"? Can they provide an example where a GPCR was shown to transmit a

signal independently of a G protein?

Reviewer #4 (Remarks to the Author): with expertise in OSCC

In the current manuscript, Liang Ding and colleagues investigated the pattern of invasion in oral squamous cell carcinoma (OSCC) in order to identify molecular mechanisms underlying the development of different subtypes of invasion that are associated with disease progression and patient survival. Using a broad range of in vitro and in vivo experiments that authors ultimately arrive at a model in which cancer-associated fibroblasts (CAFs) with high expression of the oxytocin receptor (OXTR) support invasion of OSCC cells. These OXTR^{high} fibroblasts belong to the well-known group of myofibroblasts and are able to induce desmoplastic stroma and the secretion of CCL26 in a ligand-independent manner. CCL26 binds to its receptor CCR3 on tumor cells and is required for the invasive phenotype. Expression of OXTR in CAFs is at least partially triggered by a positive feedback loop between OXTR and ERK5, and the authors suggest that inhibition of ERK5 is a promising strategy for treating OSCC patients with the worst pattern of invasion 4-5 (WPOI 4-5). In summary, the authors present a huge amount of data. Some of the data are very convincing, but rather confirmatory in nature. Other data are potentially novel, yet poorly described. And some data are not logical or are missing in order to deliver a fully convincing model. These are my concerns:

1) Novelty and published data.

Several aspects of the manuscript are not really novel. For example, data presented on the pattern of invasion (Figure 1) are confirming what has long been known in the field. Also, the general role and importance of CAFs (and myofibroblasts) in cancer growth, invasion and metastasis is well-known. The link between OXTR and ERK5 has also been established in the past, although the authors did not cite the respective study (Devost et al., *Progress in Brain Research*, 2008).

While OXTR has not been linked with OSCC, there are studies showing a role for oxytocin and its receptor in regulating metastatic features of colorectal, ovarian and melanoma cells. Interestingly, the OXT/OXTR system seems to have tumor suppressive effects in colorectal and ovarian cancer cells (Ma et al., *Front Neurosci.*, 2019; Haoyi et al. *Journal of Cancer*, 2018). In contrast, OXTR was up-regulated in melanoma and promoted migration, invasion and angiogenesis (Haoyi et al., *Carcinogenesis*, 2019). Surprisingly, I could not find a reference to the afore mentioned studies nor did the authors bother to discuss their own results and model in light of previous findings.

2) Data presentation, interpretation and relevance.

The manuscript is packed with data. However, at several instances these data are either poorly described (or not at all), overstated or misinterpreted.

For example, on page 6, lines 157ff that authors state: "Phenotypically, tumor cells in the WPOI 4-5 category, underwent transformations to a mesenchymal phenotype. Genotypically this was associated with significant stromal loss of the junctional adhesion molecules Claudin-1 and E-cadherin, and up-regulation of invasion-related N-cadherin, vimentin and MMP2 & 3 (Figure 1E)." First, the authors did not perform a phenotypic assay that would allow one to conclude a transformation to a mesenchymal phenotype. Second, the authors seem to have a wrong understanding of the term "genotype".

Next, data in Figure 1E should be presented as box plots with individual data points shown.

Furthermore, in Figure 3D the authors performed a correlation analysis and present the correlation matrix. Usually one would expect that each gene shows high correlation with itself resulting in a uniform and intense diagonal row of squares. However, in this figure the correlation of each gene with itself seems to differ. What is plotted in this figure and what is the colour legend telling us?

Next, data shown in Figure 4M, N were not described in the main text whereas Suppl. Figure S4F shows the opposite of what is described in the text or would be expected based on the other data. In detail, treatment of HN6 cells with CCL26 induced E-cadherin expression and decreased N-cadherin levels suggesting a MET instead of an EMT.

Finally, in the discussion on page 38, line 964 the authors state that serum OXT was up-regulated in OSCC patients. While this is true, the data presented in Figure 8 question the relevance of the

OXT/OXTR system in OSCC, especially due to the lack of mechanistic insights into OXTR up-regulation (see below).

3) Description of experimental procedures and details.

Certain data lack a detailed experimental description that would allow one to either repeat the experiment or at least understand better what has been done.

For example, in Figure 2E the authors schematically describe the isolation of CAFs from patients with WPOI 1-3 or 4-5. Subsequent experiments heavily rely on these 3 or 9 lines, respectively. However, it remains a mystery whether these CAF lines have been mixed after their isolation or kept separate. In case of the latter, it is unclear which line (and why) was used for the experiments and whether always the same CAF line was used and/or validation experiments with the other lines have been performed. In addition, the mouse model used in this study is poorly described and it remains unclear how the tissue-specific knockout of OXTR was achieved. The paragraph in the main text is not helpful and makes no sense (page 19, lines 493-497). It seems that the model is based on CreERT2. Hence, tamoxifen-induction would be needed, yet no such treatment was mentioned in the manuscript. Also, the labels creW and creT are uncommon and need to be explained.

The manuscript lacks detailed sequence information about the si/shRNAs and qPCR primers used. Furthermore, the molecular size of OXTR differs between Figure 4 (75 kDa) and Figure 6B, H, I, K, and Q. While the authors discuss different OXTR sizes without any context in the main text on page 18, lines 466-471, this explanation makes no sense, because the same cells (CAFs) have been analyzed in Figure 4 and Figure 6.

Next, CRISPR-mediated OXTR depletion was achieved using three different sgRNAs (Figure 4J-L). However, the T7E1 assay lacks a quantification of the cutting efficiencies. Based on the band intensities shown in Figure 4K, the sgRNAs are not very efficient. In fact, Figure 4L shows that OXTR expression is not fully blunted. Here, the authors could have been used a FACS-based negative enrichment strategy to obtain a pure population of OXTR-negative cells.

Last but not least, the authors chose to overexpress OXTR using a rather unconventional strategy based on small activating RNAs. It is unclear why this strategy was favoured over standard plasmid-based overexpression or CRISPR activation. Please explain and repeat key OXTR overexpression experiments using stable transduction of the transgene. Also, if saRNA

4) Open questions.

While the authors present plenty of data, some key questions remain unanswered.

For example, how is OXTR expression triggered in OSCC? The authors validated a known downstream signalling connection between OXTR and ERK5 (Devost et al., *Progress in Brain Research*, 2008) and suggest a positive feedback loop. Yet, it remains unclear how OXTR expression is initially increased in a subgroup of OSCC patients. Importantly, the experimental strategy and logic behind the data shown in Figure 6 does not really make sense! Why would you artificially overexpress OXTR to begin with, if you want to identify endogenous regulators of this gene.

Moreover, the authors used an ERK5 inhibitor and failed to observe an impact on lymph node metastasis and the mesenchymal phenotype of the tumor in vivo (Figure 7G, H). They argue that this might be due to the paradoxical activation of ERK5 transcriptional activity by the inhibitor XMD8-92. This finding in conjunction with other safety and off-target concerns of this inhibitor suggest that ERK5 inhibition might not be the ideal clinical intervention strategy. Hence, it is surprising that the authors did not test the therapeutic relevance of CCL26, the ultimate effector molecule according to their working model (Figure 8). For example, a CCL26-neutralizing antibody or a CCR3 inhibitor should have been tested in vivo. Alternatively, the authors could have turned directly to OXTR using Atosiban or other antagonists.

5) Readability and Language.

The authors should consider to shorten the manuscript and focus on the most important results in order to improve figure layout and readability. Some data could be fully removed (e.g. Fig.1B, Fig. 3A, Fig.4J) or moved into the supplements. Moreover, some language and typo editing is required, e.g. page 9, lines 224-226 (not 150 cohorts have been analyzed...). Also, please check page 13, line 322 (CAFWPOI1-3 instead of two times CAFWPOI4-5). The manuscript title also requires a minor correction.

Reviewer #5 (Remarks to the Author): with expertise in CAFs

The authors present a very detailed work with the goal of improving diagnosis and prognosis of OSCC. It is by now accepted that cancer-associated fibroblasts are significant factor to take into account when attempting to develop anti-tumor therapies, and as the authors and previous work before show, fibroblasts are important players in OSCC as well. The topic is timely and highly applies to other cancers.

The authors describe a pro-invasive phenotype of specific CAF subtype, WPOI 4-5 CAFs, which depends on ERK5, OXTR and CCL26, and demonstrate their functional relevance in vitro and in vivo. They use orthogonal methods to support their hypothesis, and address many aspects in deciphering the mechanism of action of this CAF subset. The authors also appropriately refer to previous work in the field of CAF heterogeneity, and place their findings within the frame of different CAF subtypes that have been demonstrated in previous research. In general, I think that the manuscript is very thorough and convincing.

General comments:

1. When I tried to figure out how the authors prepared their patient-derived fibroblasts I had to actively search for it in another publication of theirs. Since this is not a standard procedure, and people are using different protocols to produce fibroblasts, this is something that (1) should be clearly detailed in the methods, and (2) should be mentioned in the manuscript.

As far as I understand, the authors used the outgrowth method with differential trypsinization to purify the CAFs from the cancer cells. This is not the best way to purify CAFs, and might leave some cancer cells behind. There are several surface markers (some are used by the authors later on) that can be used to purify your CAF culture. The authors should at least show by FACS that their CAF population is pure by staining for CAF markers and epithelial markers. While searching for the CAF preparation protocol, I was also met with the fact that all the CAF (co)cultures in this manuscript are performed with hTERT immortalized CAFs. This is definitely something that should be highlighted. It is known that immortalization may change gene expression and cell morphology. This is an important detail that should not escape reader's eye. I would want to see that key findings are repeated with primary fibroblasts.

2. The choice of S100a4-Cre mouse is not trivial and requires an explanation: did the author see that OXTR+ CAFs also express s100a4? I say this because single cell analysis of fibroblasts has shown that this marker is not specific nor inclusive enough to cover all CAFs. If the OSCC CAFs are mostly positive for this marker, this is a piece of data I would like to see (probably in supp info)

Minor comments:

- There are many grammar issues in the manuscript, along with some typos and inconsistencies. The authors should fix these.
- The authors should fix the citations so that they match what they are citing.
- Some of the IHC figures are very small and low resolution (example: Fig 2A). It is hard to judge what they show.
- Fig 2J shows that the CAF WPOI 1-3 organoids are higher in density compared to the WPOI 4-5, which is in contrast to the description of the results in the manuscript (rows 248-249).
- Authors should note that Col1 is not an exclusive marker for fibroblasts. It is also expressed by other cell types, such as macrophages. I would suggest double staining to show specificity. New markers for CAFs have been established in the last 2-3 years in other publications and the authors should take advantage of these to refine their staining.
- "HN6" first mention in row 260 with no previous explanation/spelling out
- Fig 4C – this is an unusual way to show gating. I would expect an explanation at least in the legend

- Fig 6H – total ERK5 is elevated mostly in one cell line and not so much in the other. Can the author show another cell line to strengthen their claim?

NCOMMS-21-34717: Tumor Microenvironment control the Invasion Pattern of Oral Squamous Cell Carcinoma via auto-activating OXTR/ERK5 Signaling

Response to reviewer 1: Page 1-3

Response to reviewer 2: Page 4-7

Response to reviewer 3: Page 7-17

Response to reviewer 4: Page 17-27

Response to reviewer 5: Page 27-31

Reviewers' comments:

Reviewer #1 (Remarks to the Author): with expertise in OXTR/ERK5 signaling in cancer

The paper by Ding et al entitled "Tumor Microenvironment control the Invasion Pattern of Oral Squamous Cell Carcinoma via auto-activating OXTR/ERK5 Signaling" describes a correlation between WPOI and recurrence-free and overall survival in OSCC patients. Overall the study is intriguing, and provides a novel insight into molecular mechanisms of the role of stroma in OSCC invasiveness and progression. Of note is the novel role of OXTR regulation of ERK5 in stromal fibroblasts. The study utilized three-dimensional (3D) organotypic invasion and growth models and xenograft and orthotopic transplantations in mice in their studies, and appears to have used an appropriate number of human samples. OXTR's role was identified through RNA sequencing. This manuscript can be accepted for publication after addressing the following concerns:

Response: We thank the review for recognizing of the novelty and critical role of this work. we also thank the reviewer for excellent suggestions to further improve our manuscript.

Comments:

1) In many instances OXTR data was just mRNA expression. Where possible, including OXTR protein expression will provide additional important information about OXTR protein signaling.

Response: We thank the reviewer for raising this concern. We have shown that the OXTR protein and relative signaling expression levels were detected by several approaches including Flow cytometry, Western blot and Immunofluorescence (IF) or Immunohistochemistry (IHC). Successful OXTR knockdown or overexpression by Cas9 and SaRNA were also confirmed the OXTR protein level in CAFs. We now provide additional Western blot results: **Fig. 3b** shows that CAF_{WPOI 4-5} has more enriched OXTR protein than CAF_{WPOI 1-3}, consistent with the RNA-sequence analysis, and OXTR_{high} CAFs also have increased protein level of MEK5 and CDC37 (**Fig. 6t**). These data have been included in our revised manuscript.

2) The role of oxytocin (OXT) the physiologic ligand of OXTR needs more explanation in this context.

Response: We have supplemented the sections explaining the physiologic role of OXT/OXTR in **Results 2 and 8** of revised manuscript (marked in Red).

3) The authors suggest that the mentioned notion of autoactivated OXTR indicate that OXTR functions independent of OXT in this context? This needs to be further established and explained.

Response: The OXTR ligands OXT or AVP were dispensable for the pro-invasion role of OXTR^{High} CAFs. Similarly, in the absence of ligands, high surface concentrations of estrogen receptor, androgen receptor or epidermal growth factor receptor (EGFR) were identified as inducing autonomous receptor signaling activation in several cancer types^{1,2}. Therefore, these ligands and receptors do not always coordinate and work together. We further addressed this question in detail from three perspectives:

1. Clinical OXT level: The OXT levels in different WPOI or TNM stage groups showed no difference (**Supplementary Fig. 6c-d**). Additionally, in patients with WPOI 4-5 group, which harbored high OXTR expression, the OXT level still showed no impact on clinical prognosis. OXTR and ERK5 expression in OSCC tissues were also not correlated to serum OXT levels (**Fig. 8i-j**).

2. OXT *in vitro* and *in vivo* function: OXT and AVP were not detected in any samples of cell culture mediums in this study (**Supplementary Fig. 3e-f**). Considering the endogenous oxytocin in blood, we also evaluated the response of CAFs to the treatment of OXT/AVP. The results in **Supplementary Fig. 6g** showed that OXT/AVP could activated ERK1/2 or AKT as previously reported, however, no enhanced pro-invasion effects of CAFs were found in response to OXT/AVP stimulation (**Fig. 8k-l and Supplementary Fig. 6h**). Additionally, Carbetocin, a long-acting oxytocin analogue, has been reported to elicit interesting and peculiar behavioral effects and selectively activates only the OXTR/Gq pathway³. In this study, the OXTR/Gq/ERK5 pathway was also demonstrated to promote tumor invasion, thus we used Carbetocin (Med Chem Express china, #HY-17573) twice weekly (6 mg/kg/mice, i.p.) in orthotopic model of OSCC in NCG mice. However, treatment o with Carbetocin also showed no effect on OSCC progression (**Supplementary Fig. 3g**).

3. Molecular mechanism of OXTR activation signaling. Based on the new data below, **we have reframed the key logic in the reconstructed Figure 6**. The sequence of data presentation in Figure 6 was also modified in our revised manuscript.

1) ERK5 regulates OXTR/CCL26 transcription: In order to identify and specify the detailed effector of MAPK pathway, we overexpressed OXTR in OXTR^{Low} CAFs to mimic OXTR^{High} CAFs. Our analyses found that more activated nuclear ERK5 (**Fig. 6d-e**), rather than JNK, ERK1/2 or p38, and downstream c-fos/c-myc signaling activated the sa-OXTR group. Moreover, ERK5 was significantly activated in OXTR^{High} CAFs of OSCC patients. Importantly, both knockdown of ERK5 (**Fig. 6g**) and mutation of c-fos/c-myc promoter binding site (**Fig. 6j and I**) significantly reduced OXTR and CCL26 transcription in OXTR^{High} CAFs via dual-luciferase reporter assay, implying that up-

regulated ERK5 signaling in OXTR^{High} CAFs contributed to OXTR and CCL26 expression.

2) OXTR/Gq and CDC37 signaling activates ERK5: Subsequently, we determined why ERK5 activated in OXTR^{High} CAFs. Co-immunoprecipitation (IP) assays showed that OXTR overexpression could also increase the activity of CDC37 pathway (**Fig. 6m-n**) and Gq signaling (**Fig. 6o-p**). Inhibition of Gq/ERK5 activation by Gq knockdown confirmed that OXTR was able to activate Gq/MEK5/ERK5 pathway (**Fig. 6q-r**). Consequently, up-regulated OXTR could in turn further activate ERK5/TFs signaling to form a positive feed-back loop, which was the key mechanism that how OXTR^{High} CAFs maintained its high OXTR and CCL26 phenotype. All supplemented data were included in **Figure 6m-r** and **supplemental Figure 5** in revised manuscript. The pathway diagram in **Fig. 6u** was showed as below:

Reference

1. Baumdick, M., *et al.* A conformational sensor based on genetic code expansion reveals an autocatalytic component in EGFR activation. *Nature communications* **9**, 3847 (2018).
2. Dehm, S.M. & Tindall, D.J. Ligand-independent androgen receptor activity is activation function-2-independent and resistant to antiandrogens in androgen refractory prostate cancer cells. *J Biol Chem* **281**, 27882-27893 (2006).
3. Passoni, I., Leonzino, M., Gigliucci, V., Chini, B. & Busnelli, M. Carbetocin is a Functional Selective Gq Agonist That Does Not Promote Oxytocin Receptor Recycling After Inducing beta-Arrestin-Independent Internalisation. *J Neuroendocrinol* **28**(2016).

- 4) Given that the role of OXT and OXTR in carcinogenesis is understudied, authors should ensure that prior studies that have reported a role for OXT or OXTR in other solid organ cancers are referenced, for completeness and better context.**

Response: We have supplemented and discussed these reported findings in the **Discussion section** in our revised manuscript (marked in Red on **page 33**).

Reviewer #2 (Remarks to the Author): with expertise in OSCC

The manuscript by Ding et al report a new molecular mechanism associated with oral squamous cell carcinoma metastasis. They showed that cancer-associated fibroblasts characterized by high oxytocin receptor were the key stroma components contributing to local-regional metastasis. They found that OXTR^{High} CAFs ligand-independently induced desmoplastic stroma and that CCL26 was required for the invasive phenotype of CCR3+ tumors. Mechanistically, OXTR cells autonomously activated nuclear ERK5 transcription signaling to maintain high levels of OXTR and CCL26. They propose that activation of ERK5 plays a critical role in Oral squamous cell carcinoma metastasis. While these findings are interesting, these findings are not convincing. There are several major concerns.

Response: We would like to thank the reviewer for the time and effort spent in reviewing our manuscript and we have addressed all your key questions below.

1) In general, immunostaining quality in Figures 1 and 2 is poor.

Response: The first submission of the manuscript included the figures as a single PDF, which may have reduced the image quality. We now provided a high-resolution image using a digital microscopy application 3DHISTECH (CaseViewer) and also supplemented the data using enhanced magnification in our revised manuscript (**Fig. 1a and d, Fig. 2a**).

2) The reviewer was not convincing that macrophage and other inflammatory cells were not associated with tumor metastasis which have been demonstrated in many studies.

Response: We agreed with the reviewer's point that immune activated or suppressive-related immunocytes participated in carcinogenesis via regulating dynamic stroma-tumor crosstalk. **Notably, our team has focused previously on inflammation-related disease, such as systemic lupus erythematosus and sepsis¹⁻⁴, and inflammation-driven carcinogenesis⁵⁻⁹ for many years.** However, rescued immune activation by only immunotherapy could partially relieve the tumor progression in clinical disease¹⁰⁻¹², which led us to reconsider the function of heterogeneous stroma cells, especially the non-immune cells. Our team have been focusing on carcinoma-associated fibroblasts in tumor growth and invasion, and immune regulation in OSCC^{9,13-18}.

In this study, the clinical histologic markers WPOI was recognized as one of the invasion-involved factors, although other clinical histopathological characteristics including tumor differentiation, inflammation and invasion depth etc. also reflected the ability of tumor invasion. All these factors are able to cooperatively regulate tumor invasion and metastasis, but, currently, the relationships between WPOI and other invasion-related factor, such as inflammation, are unknown. Our team aimed to identify the potential WPOI-related stroma cells in OSCC, including infiltrated immune cells, blood vessel and CAFs. However, **only CAFs were found to be enriched in OSCC samples with WPOI 4-5. Because WPOI or inflammation**

was just one of the invasion-involved factors, this finding might not be interpreted into which tumor invasion was not associated with immunocytes. Importantly, we also established an orthotopic model of OSCC in NOD^{prkdc^{-/-}IL-2Rg^{-/-}} mice, this tumor model was severely immune-deficient, with lack of T, B and NK cells, which largely excluded the influence of immune cells on tumor lymphatic metastasis. In addition, CAF^{WPOI 4-5} alone was able to dramatically enhance tongue tumorigenesis, lymph node metastasis, with more micro-metastatic foci in the tongue. These findings suggested that CAFs are the key stromal cell type for the development of WPOI 4-5 and tumor local lymph node metastasis. Therefore, although the immunocytes/inflammation are widely recognized as one of the tumor metastasis-related inducers, data from our study indicates that CAFs were the key stromal components responsible for WPOI 4-5-related tumor invasion.

Reference

1. Shi, G., *et al.* mTOR inhibitor INK128 attenuates systemic lupus erythematosus by regulating inflammation-induced CD11b(+)Gr1(+) cells. *Biochim Biophys Acta Mol Basis Dis* **1865**, 1-13 (2019).
2. Dong, G., *et al.* STS-1 promotes IFN-alpha induced autophagy by activating the JAK1-STAT1 signaling pathway in B cells. *Eur J Immunol* **45**, 2377-2388 (2015).
3. Ding, L., Dong, G., Zhang, D., Ni, Y. & Hou, Y. The regional function of cGAS/STING signal in multiple organs: One of culprit behind systemic lupus erythematosus? *Med Hypotheses* **85**, 846-849 (2015).
4. Song, Y., *et al.* Exosomal miR-146a Contributes to the Enhanced Therapeutic Efficacy of Interleukin-1beta-Primed Mesenchymal Stem Cells Against Sepsis. *Stem Cells* **35**, 1208-1221 (2017).
5. Low, J.T., *et al.* Loss of NFKB1 Results in Expression of Tumor Necrosis Factor and Activation of Signal Transducer and Activator of Transcription 1 to Promote Gastric Tumorigenesis in Mice. *Gastroenterology* **159**, 1444-1458 e1415 (2020).
6. Liang, D., *et al.* Activated STING enhances Tregs infiltration in the HPV-related carcinogenesis of tongue squamous cells via the c-jun/CCL22 signal. *Biochim Biophys Acta* **1852**, 2494-2503 (2015).
7. Ni, Y., *et al.* Loss of NF-kB1 and c-Rel accelerates oral carcinogenesis in mice. *Oral Dis* **27**, 168-172 (2021).
8. O'Reilly, L.A., *et al.* Loss of NF-kappaB1 Causes Gastric Cancer with Aberrant Inflammation and Expression of Immune Checkpoint Regulators in a STAT-1-Dependent Manner. *Immunity* **48**, 570-583 e578 (2018).
9. Zhao, X., *et al.* Diminished CD68(+) Cancer-Associated Fibroblast Subset Induces Regulatory T-Cell (Treg) Infiltration and Predicts Poor Prognosis of Oral Squamous Cell Carcinoma Patients. *Am J Pathol* **190**, 886-899 (2020).
10. Butowski, N., *et al.* A North American brain tumor consortium phase II study of poly-ICLC for adult patients with recurrent anaplastic gliomas. *J Neurooncol* **91**, 183-189 (2009).
11. Sun, C., Mezzadra, R. & Schumacher, T.N. Regulation and Function of the PD-L1 Checkpoint. *Immunity* **48**, 434-452 (2018).
12. Lei, Q., Wang, D., Sun, K., Wang, L. & Zhang, Y. Resistance Mechanisms of Anti-PD1/PDL1 Therapy in Solid Tumors. *Front Cell Dev Biol* **8**, 672 (2020).
13. Ding, L., *et al.* A novel stromal lncRNA signature reprograms fibroblasts to promote the growth of oral squamous cell carcinoma via lncRNA-CAF/interleukin-33. *Carcinogenesis* **39**, 397-406 (2018).
14. Ren, J., *et al.* Carcinoma-associated fibroblasts promote the stemness and chemoresistance of colorectal cancer by transferring exosomal lncRNA H19. *Theranostics* **8**, 3932-3948 (2018).
15. Zhang, D., *et al.* Cancer-associated fibroblasts promote tumor progression by lncRNA-mediated RUNX2/GDF10 signaling in oral squamous cell carcinoma. *Mol Oncol* (2021).
16. Wang, Y., *et al.* Epiregulin reprograms cancer-associated fibroblasts and facilitates oral squamous cell carcinoma invasion via JAK2-STAT3 pathway. *Journal of experimental & clinical cancer research : CR* **38**, 274 (2019).
17. Zhang, D., *et al.* Midkine derived from cancer-associated fibroblasts promotes cisplatin-resistance via up-regulation of the expression of lncRNA ANRIL in tumour cells. *Sci Rep* **7**, 16231 (2017).
18. Zhang, X., *et al.* ITGB2-mediated metabolic switch in CAFs promotes OSCC proliferation by oxidation of NADH in mitochondrial oxidative phosphorylation system. *Theranostics* **10**, 12044-12059 (2020).

3) The authors did not have a local metastasis model to test their hypothesis.

Response: In this study, both the tumor xenograft model and the orthotopic model of OSCC had been established to verify the pro-metastatic role of WPOI 4-5-derived CAFs. These tumor models have been utilized by many studies for investigating tumor growth and regional cervical lymph nodes metastasis¹⁻⁵. Two main characteristics were determined in these models: the numbers of **local micrometastatic foci in tongue and regional cervical lymph nodes metastasis**. In xenograft model, we subcutaneously co-injected human CAFs and the human OSCC cell line into nude mice. Tumors arising from the OSCC cell line co-injected with CAF^{WPOI 4-5} showed significantly accelerated growth (**Fig. 2k**), displaying a pattern of marked and widespread cellular dissociation (in small cellular groups, $n < 15$) (**Fig. 2i**), resembling the WPOI 4-5 type with more local micrometastatic foci than those of mice co-injected with CAF^{WPOI 1-3} (**Supplementary Fig. 1h-i**). Related results are shown in Figure 2 and Supl Figure 1 in our revised manuscript. In the orthotopic model, we injected HN6/hCAF_s into the tongue of NOD^{prkdc^{-/-}IL-2Rg^{-/-}} mice and established a model of regional cervical lymph nodes metastasis (**Fig. 2m-n**). As in the previous xenograft model, CAF^{WPOI 4-5} dramatically enhanced tongue tumorigenesis (**Fig. 2o**), regional cervical lymph node metastasis (LNM) (**Fig. 2p**) and more micro-metastatic foci in the tongue (**Fig. 2q**).

Reference

1. Sasabe, E., *et al.* Ephrin-B2 reverse signaling regulates progression and lymph node metastasis of oral squamous cell carcinoma. *PLoS one* **12**, e0188965 (2017).
2. Spenle, C., *et al.* Impact of Tenascin-C on Radiotherapy in a Novel Syngeneic Oral Squamous Cell Carcinoma Model With Spontaneous Dissemination to the Lymph Nodes. *Front Immunol* **12**, 636108 (2021).
3. Wang, Y., *et al.* A Novel Multimodal NIR-II Nanoprobe for the Detection of Metastatic Lymph Nodes and Targeting Chemo-Photothermal Therapy in Oral Squamous Cell Carcinoma. *Theranostics* **9**, 391-404 (2019).
4. Chen, H., *et al.* The antilymphatic metastatic effect of hyaluronic acid in a mouse model of oral squamous cell carcinoma. *Cancer Biol Ther* **21**, 541-548 (2020).
5. Wang, S.H., *et al.* Laminin gamma2-enriched extracellular vesicles of oral squamous cell carcinoma cells enhance in vitro lymphangiogenesis via integrin alpha3-dependent uptake by lymphatic endothelial cells. *Int J Cancer* **144**, 2795-2810 (2019).

4) A large amount of studies has shown that IL-6, MCP-1 and other chemokines are associated with tumor metastasis. The reviewer was not convincing that only CCL26 play a critical role in oral cancer metastasis.

Response: We agree with this valid point that many cytokines/chemokines promoted tumor metastasis. For example, we previously found that CAFs-derived IL-6 up-regulated epiregulin for tumor migration and CAF-derived IL-33 promoted tumor growth^{1,2}. In this study, CXCL6, CCL26, CCL18, IL-1 β , CCL7, CCL5 and TNFSF14/LIGHT were significantly up-regulated in OXTR^{High} CAFs, a result which derived from the comparison between OXTR^{High} and OXTR^{Low} CAFs, while IL-6 and CCL2 were unchanged and not involved in this study.

Notably, the potential ECM remodelling function and pro-invasion effects of each these cytokines or chemokines were estimated and performed on two specific CAF culture

systems: 1. culturing OXTR^{Low} CAFs in Matrigel-collagen gels supplemented with recombinant cytokines, and 2. stable knockdown of each cytokine/chemokine in OXTR^{High} CAFs (Fig. 5d and e). In the co-culture system, **the addition of recombinant human CXCL6, IL-1 β , CCL7, CCL5 or LIGHT into the 3D co-culture system, caused a modest increase in Lmax for tumor invasion as previously reported^{3,4} (Fig. 5f)**. On the other hand, **addition of CCL18 or CCL26 caused a doubling in Lmax**. In order to confirm the specific impacts of CAFs in this co-culture model, we also knocked down seven specific cytokine/chemokine genes in OXTR^{High} CAFs. However, only CCL26 knockdown in OXTR^{High} CAFs significantly attenuated the invasion length of tumor cells, while **knockdown of IL-1 β and CCL7 etc. were not enough to inhibit tumor invasion (Fig. 5g)**. These data demonstrate that CCL26 in OXTR^{High} CAFs has a superior ability to promote tumor invasion.

Reference

1. Wang, Y., *et al.* Epiregulin reprograms cancer-associated fibroblasts and facilitates oral squamous cell carcinoma invasion via JAK2-STAT3 pathway. *Journal of experimental & clinical cancer research : CR* **38**, 274 (2019).
2. Ding, L., *et al.* A novel stromal lncRNA signature reprograms fibroblasts to promote the growth of oral squamous cell carcinoma via lncRNA-CAF/interleukin-33. *Carcinogenesis* **39**, 397-406 (2018).
3. Brunetti, G., *et al.* LIGHT/TNFSF14 Promotes Osteolytic Bone Metastases in Non-small Cell Lung Cancer Patients. *J Bone Miner Res* **35**, 671-680 (2020).
4. Jung, D.W., *et al.* Tumor-stromal crosstalk in invasion of oral squamous cell carcinoma: a pivotal role of CCL7. *Int J Cancer* **127**, 332-344 (2010).

Reviewer #3 (Remarks to the Author): with expertise in ERK5 and cancer

This study characterized the role of cancer-associated fibroblasts (CAFs) in the invasion pattern of oral squamous cell carcinoma (OSCC). Specifically, the data indicated that tumors displaying the worst pattern of invasion comprised a high proportion of myofibroblastic CAFs exhibiting high oxytocin receptor (OXTR) expression. This cellular subpopulation contributes to the mesenchymal phenotype of the tumor and the acceleration of local-regional metastasis through increased ERK5 signaling.

Overall opinion:

I have to say that I am very impressed by the amount of data included in this paper. However, the overall lack of accuracy in the scientific interpretation of the results and attention to details give the impression that the findings are not convincing. This is compounded by the fact that there is insufficient evidence to firmly conclude that OXTR is expressed in CAFs, as there is no experimental demonstration that the OXTR antibody used for immunoblot or immunofluorescence analyses is specific (see also comment 6). Moreover, a key question that was not answered and could have made a difference is, if "OXTR^{High} CAFs promote tumor cell invasion in a ligand-independent manner" page 18 line 455, what is the mechanism of OXTR activation in CAFs? The scientific argument is also fairly poor given that many references provided do not

support the statements on which the authors built the rationale for their study. Taken together, these major concerns significantly decreased my enthusiasm for supporting the publication of this study.

Response: We would like to express our gratitude for the reviewer's time, effort, constructive criticism and feedback. In addition to new data, we have improved the quality of our original data by improving accuracy and interpretation.

Major comments

1) Page 9 line 217: The authors state that “The only component found to be significantly altered between WPOI 1-3 and WPOI 4-5 types were collagen I expressing stromal cells (Figure 2B), which were enriched in the WPOI 4-5 stroma, indicating a potential pro-invasion phenotype of Collagen I rich ECM microenvironments, as previously reported (Bhattacharjee et al., 2021; Chen et al., 2021)”. These two references do not support this statement as both studies proposed that myCAF-expressed type I collagen suppressed tumor growth. Specifically, Chen et al showed that targeted deletion of Col1 in myofibroblasts accelerated the emergence of PanINs and PDAC, while Bhattacharjee et al. proposed that myCAF-expressed type I collagen suppressed tumor growth by mechanically restraining tumor spread. The authors must clarify this issue.

Response: We thanks the reviewer for this insightful comment. Firstly, collagen fibers not only hinder cancer invasion as Chen et al. reported in mouse PDAC; they are well known to promote tumor invasion and support tumor proliferation and invasion in many cancer types, **including in human PDAC** ¹⁻⁶. Consistent with this dual role, invasive cancer cells and cells recruited into tumors produce both increased levels of MMPs and collagen, which interacted to regulate tumor metastasis ^{7,8}. Indeed, MMP2/3 were enriched in WPOI 4-5 tissue in this study (**Fig. 1d**). **Secondly**, fibrillar collagen, particularly collagen type I, is deposited densely in stroma adjacent to tumor ⁹. Our original intention with determining collagen I expression was only to identify the stroma differences between WPOI 1-3 and WPOI 4-5 in OSCC tissues. Since CAFs were the primary source of type I collagen in tumor microenvironment ⁹, our study found that collagen I-expressed stromal cells was the only component significantly increased in WPOI 4-5 OSCC, indicating that stromal CAFs might participate into the development of WPOI 4-5. Since fibronectin (FN) and collagen I are known to interact with each other to regulate cellular behaviour and FN is also an essential component of the CAF-derived matrix ¹⁰⁻¹². The stromal FN, similar to collagen I and α -SMA, was also up-regulated in WPOI 4-5 tumor (**Fig. 2c-d, Supplementary Fig. 1f**).

References

1. Armstrong, T., et al. Type I collagen promotes the malignant phenotype of pancreatic ductal adenocarcinoma. *Clin Cancer Res* **10**, 7427-7437 (2004).
2. Deng, J., et al. DDR1-induced neutrophil extracellular traps drive pancreatic cancer metastasis. *JCI Insight* **6**(2021).
3. Eisinger-Mathason, T.S., et al. Hypoxia-dependent modification of collagen networks promotes sarcoma

- metastasis. *Cancer Discov* **3**, 1190-1205 (2013).
4. Penet, M.F., *et al.* Structure and Function of a Prostate Cancer Dissemination-Permissive Extracellular Matrix. *Clin Cancer Res* **23**, 2245-2254 (2017).
 5. van Kempen, L.C., *et al.* Type I collagen synthesis parallels the conversion of keratinocytic intraepidermal neoplasia to cutaneous squamous cell carcinoma. *J Pathol* **204**, 333-339 (2004).
 6. Barkan, D., *et al.* Metastatic growth from dormant cells induced by a col-I-enriched fibrotic environment. *Cancer Res* **70**, 5706-5716 (2010).
 7. Wang, K., *et al.* Breast cancer cells alter the dynamics of stromal fibronectin-collagen interactions. *Matrix Biol* **60-61**, 86-95 (2017).
 8. Zhuge, Y. & Xu, J. Rac1 mediates type I collagen-dependent MMP-2 activation. role in cell invasion across collagen barrier. *J Biol Chem* **276**, 16248-16256 (2001).
 9. Nissen, N.I., Karsdal, M. & Willumsen, N. Collagens and Cancer associated fibroblasts in the reactive stroma and its relation to Cancer biology. *Journal of experimental & clinical cancer research : CR* **38**, 115 (2019).
 10. Erdogan, B., *et al.* Cancer-associated fibroblasts promote directional cancer cell migration by aligning fibronectin. *J Cell Biol* **216**, 3799-3816 (2017).
 11. Miyazaki, K., *et al.* Collective cancer cell invasion in contact with fibroblasts through integrin- α 5 β 1/fibronectin interaction in collagen matrix. *Cancer Sci* **111**, 4381-4392 (2020).
 12. Ao, M., *et al.* Stretching fibroblasts remodels fibronectin and alters cancer cell migration. *Sci Rep* **5**, 8334 (2015).

2) Fig. 3C: The authors must confirm that primary CAF_{SWPOI 4-5} exhibit lower expression of OXTR than CAF_{SWPOI 1-3}, at protein level. The data should be presented next to panel C.

Response: We thank the reviewer for their insightful comments and as suggested. We have provided the requested Western blots. This additional data shows that CAF^{WPOI 4-5} exhibited elevated OXTR protein levels compared to CAF^{WPOI 1-3} (Fig. 3b), which is consistent with our RNA-sequence analysis.

3) Pages 12-13 (L306- L311): The rationale for choosing OXTR as opposed to the other candidates identified by transcriptomic analysis (e.g. other neuronal development and function-related genes or inflammation and cancer-related genes) is not explained. Why are RELN, OXTR and CAV3 in both lists?

Response: We thank the reviewer for this insightful comment. The reason for choosing OXTR as our candidate, rather than others identified by transcriptomic analysis was based on two key characteristics: KEGG pathway analysis and expression abundance:

Firstly, KEGG pathway annotation and enrichment analysis: KEGG pathway annotation and enrichment analysis for differentially expressed genes revealed several pathways were significantly enriched, including neuroactive ligand-receptor interaction, calcium signaling pathway, cytokine-cytokine receptor interaction, focal adhesion, oxytocin signaling pathway and cell adhesion molecules. **Our extensive literature survey indicated that, among these pathways, oxytocin signaling was most significantly related with the other pathways¹⁻⁴.** The reported function of migration regulation by OXT/OXTR/calcium signaling is also the potential reason for choosing OXTR in this study^{5,6}. Notably, other candidates were also considered in our preliminary study and several of them were under study by our group (RELN, MYH10 and CCL2). Partial RELN-related data in OSCC were published by our group in 2021⁷.

Secondly, expression abundance: The expression level of each candidate might partly reflect its importance grade of function. Therefore, we analyzed the relative expression of candidate genes up-regulated in the transcriptomic analysis in **Supplementary Fig. 2C** and found that OXTR showed the highest relative gene expression abundance. *RELN*, *MYH10* and *CCL2* also showed relatively high gene expression in CAFs, and thus were also investigated by our team in studies outwith the scope of this research.

Reference

1. Arrowsmith, S. & Wray, S. Oxytocin: its mechanism of action and receptor signalling in the myometrium. *J Neuroendocrinol* **26**, 356-369 (2014).
2. Gehrig-Burger, K., Slaninova, J. & Gimpl, G. Depletion of calcium stores contributes to progesterone-induced attenuation of calcium signaling of G protein-coupled receptors. *Cell Mol Life Sci* **67**, 2815-2824 (2010).
3. Martens, H., Kecha, O., Charlet-Renard, C., Defresne, M.P. & Geenen, V. Neurohypophysial peptides stimulate the phosphorylation of pre-T cell focal adhesion kinases. *Neuroendocrinology* **67**, 282-289 (1998).
4. Zatkova, M., Reichova, A., Bacova, Z. & Bakos, J. Activation of the Oxytocin Receptor Modulates the Expression of Synaptic Adhesion Molecules in a Cell-Specific Manner. *J Mol Neurosci* **68**, 171-180 (2019).
5. Cassoni, P., et al. Oxytocin induces proliferation and migration in immortalized human dermal microvascular endothelial cells and human breast tumor-derived endothelial cells. *Mol Cancer Res* **4**, 351-359 (2006).
6. Cattaneo, M.G., Chini, B. & Vicentini, L.M. Oxytocin stimulates migration and invasion in human endothelial cells. *Br J Pharmacol* **153**, 728-736 (2008).
7. Zhang, X., et al. Functional Heterogeneity of Reelin in the Oral Squamous Cell Carcinoma Microenvironment. *Front Oncol* **11**, 692390 (2021).

4) Page 13 line 317: “To date, the function of peripheral OXTR expression has been elusive, particularly during carcinogenesis (Cuneo et al., 2019)”. The reference provided here does not support this statement.

Response: We thank the reviewer for pointing out our error and we have supplemented with additional references in **Results section** of our revised manuscript (marked in Red) as below:

1. Pan, S., et al. Stimulation of hypothalamic oxytocin neurons suppresses colorectal cancer progression in mice. *Elife* 10(2021).
2. Pequeux, C., et al. Oxytocin synthesis and oxytocin receptor expression by cell lines of human small cell carcinoma of the lung stimulate tumor growth through autocrine/paracrine signaling. *Cancer Res* **62**, 4623-4629 (2002).
3. Ji, H., et al. Oxytocin involves in chronic stress-evoked melanoma metastasis via beta-arrestin 2-mediated ERK signaling pathway. *Carcinogenesis* **40**, 1395-1404 (2019).

5) Figure 3D and S2C: Weren't the authors surprised to find that OXTR mRNA level correlated poorly with neuronal cells (identified as S100B+ cells) given the strong expression of OXTR in the brain? Wouldn't this controversial observation invalidate their analysis?

Response: We thank the reviewer for this insightful comment. There are two potential reasons accounting for this irrelevance: **Firstly**, the data set analysed in Figure 3D was derived from the RNA-sequence of head and neck squamous cell carcinoma (HNSCC, n=522, TCGA data set, Firehose Legacy, http://www.cbioportal.org/study/summary?id=hnsk_tcga) (**Fig a. left**). As such, in these human tissue samples, the abundance of peripheral nerves is limited. In our previous nerve-related study¹, we found that 24% OSCC tissues showed no significant

S100b+ peripheral nerves (nerve density ≤ 1 by HE slide analysis) (unpublished data) (**Fig b. right**). Thus, it's not surprising that OXTR mRNA level correlated poorly with S100b. **Secondly**, in addition to OXT and OXTR expression within the brain, expression of OXTR in peripheral tissues and non-neural cells, including Testes, Uterus, Dermal fibroblasts and keratinocytes, Cardiomyocytes, Osteoclasts has been documented². **Such a diverse expression pattern between OXTR and S100b in peripheral tissues, therefore does not invalidate our findings.**

Reference

1. Fu, Y., *et al.* Worst Pattern of Perineural Invasion Redefines the Spatial Localization of Nerves in Oral Squamous Cell Carcinoma. *Front Oncol* **11**, 766902 (2021).
2. Jurek, B. & Neumann, I.D. The Oxytocin Receptor: From Intracellular Signaling to Behavior. *Physiol Rev* **98**, 1805-1908 (2018).

6) Page 18 line 466: The authors explained that OXTR is detected at 75 kDa, instead of the expected 45-66 kDa molecular weight, due to glycosylation of the N-terminus. However, they provide no experimental evidence to substantiate their hypothesis. Given the novelty of finding OXTR in CAFs, it is essential that the authors rule out the possibility that the band detected by the OXTR antibody is not specific. For example, they could show that ectopically expressed OXTR in CAFs is expressed as a 75 kDa protein.

Response: Numerous studies have demonstrated that members of G-protein-coupled receptors (GPCR) proteins are usually glycosylated on extracellular domains, including oxytocin receptor (OXTR)¹⁻³. Consequently, glycosylated peptides migrate with an apparent molecular weight greater than the peptide molecular weight when analyzed by SDS-PAGE¹. Many anti-OXTR antibodies raised against the peptide corresponding to N-terminus amino acids of the human receptor sequence, and glycosylation of the N-terminus of OXTR renders higher molecular weight (~70kd) than native form of OXTR (~47kd) in certain tissues and cells⁴. Consistently, several studies have showed a clear signal was detected for OXTR protein at 70-75 kd⁵⁻⁷.

To document the specificity of the OXTR antibody in this study, we further performed two experiments to validate our findings. OXTR is a seven-membrane spanning receptor. In preliminary study, we ectopically expressed OXTR in human primary CAFs by pSLenti-3flag-plasmid containing full-length OXTR gene via lentivirus and were able to detect an ~75 kDa band in overexpression groups by anti-flag

antibody (see image below).

However, these primary CAFs overexpressing OXTR showed poor vitality and growth condition. Since the exogenous introduction of a plasmid containing a large fragment has a stronger cellular stimulation to primary cells when compared with many established tumor cell lines, leading to significant cell senescence and impaired cell growth during cell culture *in vitro*. **This is a common problem faced during human primary cell culture.** Therefore, exogenous overexpression of OXTR by plasmid containing full-length *OXTR* gene was not used in this study. SaRNAs are short double-stranded oligonucleotides that can selectively and safely enhance gene transcription at the endogenous level and have been developed for clinical use⁸. saRNA was chosen to endogenously overexpress OXTR protein in this study (**Fig. 6a**). Overexpression of OXTR by saRNA showed no impacts on cell vitality and also successfully up-regulated the immunoreactivity for OXTR in ~75 kDa.

Importantly, we also found that sg-OXTR successfully knocked down the OXTR expression at 75 kDa, however, three AVPR family member expression levels remained unchanged (**Fig. 3f and Supplementary Fig. 3h-j**). OXTR knockdown by siRNA also performed in our revised manuscript and showed similar results (**Supplementary Fig. 6k**). In summary, the above data confirm the specific detection of OXTR in CAFs.

Reference

1. Wheatley, M. & Hawtin, S.R. Glycosylation of G-protein-coupled receptors for hormones central to normal reproductive functioning: its occurrence and role. *Hum Reprod Update* **5**, 356-364 (1999).
2. Patwardhan, A., Cheng, N. & Trejo, J. Post-Translational Modifications of G Protein-Coupled Receptors Control Cellular Signaling Dynamics in Space and Time. *Pharmacol Rev* **73**, 120-151 (2021).
3. Kimura, T., *et al.* The role of N-terminal glycosylation in the human oxytocin receptor. *Mol Hum Reprod* **3**, 957-963 (1997).
4. Szeto, A., *et al.* Regulation of the macrophage oxytocin receptor in response to inflammation. *Am J Physiol Endocrinol Metab* **312**, E183-E189 (2017).
5. Kimura, T., *et al.* Expression and immunolocalization of the oxytocin receptor in human lactating and non-lactating mammary glands. *Hum Reprod* **13**, 2645-2653 (1998).
6. Colucci, S., Colaianni, G., Mori, G., Grano, M. & Zallone, A. Human osteoclasts express oxytocin receptor. *Biochemical and biophysical research communications* **297**, 442-445 (2002).
7. Wei, J., *et al.* Involvement of Oxytocin Receptor/Erk/MAPK Signaling in the mPFC in Early Life Stress-Induced Autistic-Like Behaviors. *Front Cell Dev Biol* **8**, 564485 (2020).
8. Sarker, D., *et al.* MTL-CEBPA, a Small Activating RNA Therapeutic Upregulating C/EBP-alpha, in Patients with Advanced Liver Cancer: A First-in-Human, Multicenter, Open-Label, Phase I Trial. *Clin Cancer Res* **26**, 3936-3946 (2020).

7) Page 23 line 612: The authors state: “Our analysis showed that CCL26 enhanced

OXTR^{Low} CAFs-induced E-cadherin suppression via snail and twist, and up-regulated N-cadherin levels, which was significantly inhibited by CCR3 loss (Figure S4F"). This is clearly a gross over-interpretation of the data. This conclusion must be rephrased to clearly and accurately describe the data presented in Figure S4F.

Response: We have rephrased our description of the data in our revised manuscript on page 20 line 503 in accordance with your request.

8) Fig. 6 is overall very problematic: Initially, the authors searched for the mechanism underpinning OXTR and CCL26 induced expression through analyzing transcription factor (TF) binding sites in the OXTR and CCL26 promoters (panel E). Using this approach, they propose that MAPK pathways are involved because the TFs orchestrating OXTR and CCL26 induced expression in CAFs are downstream targets of MAPK pathways. This would logically implied that MAPKs are responsible for controlling OXTR expression in CAFs. However, panels F, G and H showed that overexpression of OXTR via sa-OXTR promote ERK5 nuclear translocation, suggesting that ERK5 is activated by OXTR. This is very confusing.

Response: We thank the reviewer for their insight, however this question partially overlaps with comment 5 from reviewer 1. We have now supplemented the original Figure 6 with additional data as requested. Based on the new data below (Response 9-14), **we have also reframed the key logic with a reconstructed Figure 6:**

1) ERK5 regulates OXTR/CCL26 transcription: To identify and specifically detail the effector of the MAPK pathway, we overexpressed OXTR in OXTR^{Low} CAFs to mimic OXTR^{High} CAFs. Our analyses found more activated nuclear ERK5 (**Fig. 6d-e**) and downstream c-fos/c-myc signaling activation (**Fig. 6h and supplementary Fig. 5e-f**) in the sa-OXTR group. Moreover, ERK5 was significantly activated in OXTR^{High} CAFs of OSCC patients, even without oxytocin stimulation. Importantly, both knockdown of ERK5 (**Fig. 6g**) and mutation of c-fos/c-myc promoter binding site (**Fig. 6j and l**) significantly reduced OXTR and CCL26 transcription in OXTR^{High} CAFs using a dual-luciferase reporter assay. This data indicates that up-regulated ERK5 signaling in OXTR^{High} CAFs is attributed to the level of OXTR and CCL26 expression.

2) OXTR/Gq and CDC37 signaling activates ERK5: Subsequently, we also determined how ERK5 activated in OXTR^{High} CAFs. Co-immunoprecipitation (IP) assays showed that OXTR overexpression increased the activity of CDC37 pathway (**Fig. 6m-n**) and Gq signaling (**Fig. 6o-p**). Inhibition of Gq/ERK5 activation by Gq knockdown confirmed that OXTR was able to activate Gq/MEK5/ERK5 pathway (**Fig. 6q-r**). Consequently, up-regulated OXTR could in turn further activate ERK5/TFs signaling to form a positive feed-back loop, which was the key mechanism by which OXTR^{High} CAFs maintained its high OXTR and CCL26 phenotype. We refer the reviewer to our response to comment 5 of reviewer for additional supportive data.

Diagram of OXTR signalling pathway

9) In contrast to the authors' statement (page 28 line 726), panel G does not show that sa-OXTR targeting up-regulates MEK5 in CAFs.

Response: We thank the reviewer for this comment and we draw the reviewer's attention to **Supplementary Fig. 1I**, where we showed that the activation status of four subfamilies of the MAPK pathway, including ERK1/2, JNK, P38 and ERK5. In accordance with the reviewer's suggestion, we have also supplemented the IF analysis of MEK5 expression here and in our revised manuscript in (**Supplemental Figure.5i**), where we show elevated MEK5 levels in sa-OXTR CAFs, consistent with the results in Figure 6f and t of our revised manuscript.

10) The quality of the immunoblot images presented in panels J and I is very poor and the conclusion that "increased levels of OXTR correlated with increased formation of a, ligand independent, Gαq, PKCζ and MEK5 complex, while simultaneously resulting in less association of β-arrestin with OXTR" (page 29 lines 743-746) is not supported by the data. There is no evidence that "OXTR/MEK5 interaction was mainly mediated by Gαq in CAFs (page 29 lines 746).

Response: We replaced the old western blotting image with a new one with better quality and supplemented the relative quantification of WB results from three biological replications in **Fig. 6m-p** of revised manuscript. We also knocked down the expression of Gαq and β-arrestin 1/2 by siRNAs (**see below**). These data show that Gαq inhibition significantly reduced ERK5 phosphorylation and OXTR expression (**Fig. 6q-r**). However, inhibition of β-arrestin 1/2 showed no significant impact on OXTR expression and ERK5 activation, which was consistent with our previous IP results. Thus, OXTR/ERK5 activation could be mediated by Gαq in CAFs. We also modified our description in the revised manuscript.

11) The overall quality of the immunoblot presented in panel K is also very poor. siRNAp38 did not decrease p-p38 level and this is completely overlooked by the authors.

Response: We thank the reviewer for this comments and we replaced the immunoblot band of p38 with a new one with better quality in revised manuscript (**Supplementary Fig. 5b**), and also showed additional repeated data here. Q-PCR assay was used to determine the siRNA efficiency and western blot was used to show the protein levels of p38, ERK5, OXTR and CCL26 after p38 knockdown.

12) Panel L: XMD8-92 is not a specific inhibitor of ERK5.

Response: We thank the reviewer for this comment and we agree that XMD8-92 is not a specific inhibitor of ERK5. While the first ERK5i XMD8-92, XMD17-26 etc. were selective for ERK5 over other kinases, they have off-target effects on the bromodomain containing protein BRD4. Therefore, we specifically knocked down ERK5 by siRNA to ablate this kinase activity in **Fig. 6g**. The results from the dual-luciferase reporter assay clearly demonstrate that knockdown of ERK5 reduces OXTR and CCL26 transcription, consistent with the findings by using the inhibitors.

13) Panel M: Increased MEK5 expression in OXTR^{High} CAF compared with OXTR^{Low} CAFs should be confirmed by immunoblot analysis.

Response: We supplemented relative quantification of IF staining (**Fig. 6s**) and WB analysis of ERK5 activation-related signaling in OXTR^{Low} CAFs and OXTR^{High} CAFs (**Fig. 6t**). We added these data in revised manuscript in Figure 6s-t.

14) Panel N: The differences between the intensities of the signals are too small to draw any conclusion.

Response: We repeated these experiments and added the data in revised manuscript in **Fig. 6h** and **Supplemental Fig. 5e**. The current image is representative of data from three biological replications.

Minor comments

15) Page 6 line 159: The word “genotypically” is not correct in this context

Response: We modified this inappropriate description in revised manuscript. “genotype” was deleted. More appropriate description was adapted in revised manuscript on **page 6 line 144**.

16) Page 7 line 183: “The question of whether any of the identified driver mutations actually initiates carcinogenesis asked us to question a primary role for tumor stromal components in multifocal and recurrent OSCC.” This sentence must be rephrased.

Response: We deleted this inappropriate description in revised manuscript.

17) Legend of Fig. 2K-P on page 12 line 283: The authors must clarify the meaning of “conditional” referred to CAF^{WPOI 1-3} and CAF^{WPOI 4-5}. The authors must also clearly indicate which HN6/hCAFs-derived tumor grafts, i.e. CAF^{WPOI 1-3} or CAF^{WPOI 4-5}, are presented in the photographs (panel M (LMN- and LMN+) and analysed (panel N).

Response: We thank the reviewer for pointing out issues with the figure legends. We have now modified the legends in our revised manuscript on **page 12 line 261** and also improved clarity of **Fig. 2k-p**.

18) Fig. 3B: The writing is too small to read the labelling of the genes. Page 14 line 366: In the sentence “GSVA results of HNSCC samples revealed that activated OXTR signaling ...”, “of HNSCC samples” should be removed.

Response: We have corrected related figure and removed these words in revised manuscript.

19) Page 15 line 371: Should be “Figure S2I-J” instead of “Figure S3H-I”.

Page 18 line 461: “ablation of OXTR in cells edited cells, which effectively reduced expression of PDGFR-β and N-cadherin in OXTR^{High} CAFs (Figure 4L).

Page 28 line 713: “Our Western blot analysis showed that sa-OXTR elevated cytoplasmic expression levels of activated CAF markers including N-cadherin, FAP, PDGFR-β, and CAV-1, CCL26 level, which were also enhanced by sa-OXTR.

Response: We have corrected these errors in revised manuscript.

20) Fig 4: Panel I is too small to visualise efficient oxtr ablation in S100a4 positive cells. Moreover, the legend of panel T-W failed to clearly explain the difference between Oxftr^{fl/fl} S100a4^{creW} and Oxftr^{fl/fl} S100a4-^{creT} samples.

Response: We thank the reviewer for their insightful comments. In order to improve figure 4I, we have replaced it with a larger format figure. We have also improved the

clarity of panel T-W “Oxtr^{fl/fl} S100a4-cre^w refer to wild type mice (Oxtr^{fl/fl}), Oxtr^{fl/fl} S100a4-cre^T refers to transgenic mice (S100a4-cre; Oxtr^{fl/fl}). We have used the nomenclature Oxtr^{fl/fl} and S100a4-cre; Oxtr^{fl/fl} in the legend of our revised manuscript.

21) Page 28 line 733: What do the authors mean by OXTR can “directly activate MEK1/ERK1/2 in a G protein-independent manner”? Can they provide an example where a GPCR was shown to transmit a signal independently of a G protein?

Response: Although ERK1/2 activation can be caused by activated G protein, the ERK1/2 pathway mediated by β -arrestins has been shown to be segregated from G protein activation. β -arrestins can function as molecular mediators of G protein-independent signaling by acting to scaffold a variety of signaling proteins. For instance, in parathyroid hormone (PTH)/ PTH1R system^{1,2}, Lefkowitz RJ. and Luttrell LM et al. found that MAPK activation can be selectively stimulated through the use of PTH analogs that preferentially induce G protein– or β -arrestin–coupled conformations of the receptor. Such ligands with the ability to preferentially activate one of two distinct signaling pathways through the same receptor are referred to as **biased agonists**. PTH- β arr as a biased agonist has the ability to stimulate β -arrestin–mediated ERK1/2 activation independent of heterotrimeric G protein activation in HEK293 cell. Similarly, the beta adrenergic receptor (beta2AR) can also signal to ERK via a GRK5/6- β -arrestin-dependent pathway, which is independent of G protein coupling³.

Reference

1. Gesty-Palmer, D., et al. A beta-arrestin-biased agonist of the parathyroid hormone receptor (PTH1R) promotes bone formation independent of G protein activation. *Sci Transl Med* **1**, 1ra1 (2009).
2. Gesty-Palmer, D., et al. Distinct beta-arrestin- and G protein-dependent pathways for parathyroid hormone receptor-stimulated ERK1/2 activation. *J Biol Chem* **281**, 10856-10864 (2006).
3. Shenoy, S.K., et al. beta-arrestin-dependent, G protein-independent ERK1/2 activation by the beta2 adrenergic receptor. *J Biol Chem* **281**, 1261-1273 (2006).

Reviewer #4 (Remarks to the Author): with expertise in OSCC

In the current manuscript, Liang Ding and colleagues investigated the pattern of invasion in oral squamous cell carcinoma (OSCC) in order to identify molecular mechanisms underlying the development of different subtypes of invasion that are associated with disease progression and patient survival. These OXTR^{high} fibroblasts belong to the well-known group of myfibroblasts and are able to induce desmoplastic stroma and the secretion of CCL26 in a ligand-independent manner. CCL26 binds to its receptor CCR3 on tumor cells and is required for the invasive phenotype. Expression of OXTR in CAFs is at least partially triggered by a positive feedback loop between OXTR and ERK5, and the authors suggest that inhibition of ERK5 is a promising strategy for treating OSCC patients with the worst pattern of invasion 4-5 (WPOI 4-5). In summary, the authors present a huge amount of data. Some of the data

are very convincing, but rather confirmatory in nature. Other data are potentially novel, yet poorly described. And some data are not logical or are missing in order to deliver a fully convincing model. These are my concerns:

Response: We thank the reviewer for suggestions to improve our manuscript. We underlined all your key questions as below.

1) Novelty and published data.

Several aspects of the manuscript are not really novel. For example, data presented on the **a) pattern of invasion** (Figure 1) are confirming what has long been known in the field. Also, the general role and **b) importance of CAFs** (and myofibroblasts) in cancer growth, invasion and metastasis is well-known. The link between **c) OXTR and ERK5** has also been established in the past, although the authors did not cite the respective study (Devost et al., Progress in Brain Research, 2008). While OXTR has not been linked with OSCC, there are studies showing a role for oxytocin and its receptor in regulating metastatic features of colorectal, ovarian and melanoma cells. Interestingly, the OXT/OXTR system seems to have tumor suppressive effects in colorectal and ovarian cancer cells (Ma et al., Front Neurosc., 2019; Haoyi et al. Journal of Cancer, 2018). In contrast, OXTR was up-regulated in melanoma and promoted migration, invasion and angiogenesis (Haoyi et al., Carcinogenesis, 2019). Surprisingly, **d) I could not find a reference to the afore mentioned studies nor did the authors bothered to discuss their own results and model in light of previous findings.**

Response: We thank the reviewer for their insightful comments and address their two key issues:

1. What's new in this study:

a) We would like clarify that while Worst POI (WPOI) can predict poor recurrence-free survival at early-stage OSCC (T1-2N0M0), it remained unclear whether the WPOI score shifted as OSCC progressed from early stage to advanced stage with high metastasis capability. **Moreover, the molecular mechanisms underlying the disease progressive stages remained unclear.**

b) We agree with the reviewer that the pivotal role of CAFs in tumor development and related therapy has been demonstrated by many studies, including our own ¹⁻⁵. However, in this manuscript, we have focused on the most recent discoveries concerning **fibroblast heterogeneity, plasticity and functionality. Our findings lead us to reconsider the heterogeneous role of CAFs subpopulations in the various WPOI classifications.**

c) The previous report in 2008 which the reviewer refers, only documented that oxytocin (100 nM) could increase phospho-ERK5 in M11 myometrial cells. **However, detailed pathway details** involving ERK5 activation in response to oxytocin and its function in disease progression **were not documented in this report 6.** In the present study, non-classic MAPK family ERK5 signaling was found to be activated by OXTR itself, even without oxytocin stimulation. OXTR overexpression itself was sufficient for induction of activated Gq/MEK5/ERK5 pathway, in turn, phospho-ERK5 further promoted c-fos and c-myc-dependent OXTR transcription. Our finding of a positive

feedback loop also adds to the novelty of this study.

2. The inconsistent role of OXTR in cancer: d) We have extended and revised our discussion within the context of our findings and supplemented with additional references in **Discussion section** of our revised manuscript (marked in Red, **page 32-33**).

Reference

1. Ding, L., *et al.* A novel stromal lncRNA signature reprograms fibroblasts to promote the growth of oral squamous cell carcinoma via lncRNA-CAF/interleukin-33. *Carcinogenesis* **39**, 397-406 (2018).
2. Ren, J., *et al.* Carcinoma-associated fibroblasts promote the stemness and chemoresistance of colorectal cancer by transferring exosomal lncRNA H19. *Theranostics* **8**, 3932-3948 (2018).
3. Zhao, X., *et al.* Diminished CD68(+) Cancer-Associated Fibroblast Subset Induces Regulatory T-Cell (Treg) Infiltration and Predicts Poor Prognosis of Oral Squamous Cell Carcinoma Patients. *Am J Pathol* **190**, 886-899 (2020).
4. Zhang, D., *et al.* Midkine derived from cancer-associated fibroblasts promotes cisplatin-resistance via up-regulation of the expression of lncRNA ANRIL in tumour cells. *Sci Rep* **7**, 16231 (2017).
5. Zhang, X., *et al.* ITGB2-mediated metabolic switch in CAFs promotes OSCC proliferation by oxidation of NADH in mitochondrial oxidative phosphorylation system. *Theranostics* **10**, 12044-12059 (2020).
6. Devost, D., Wrzal, P. & Zingg, H.H. Oxytocin receptor signalling. *Prog Brain Res* **170**, 167-176 (2008).

2) Data presentation, interpretation and relevance.

“Phenotypically, tumor cells in the WPOI 4-5 category, underwent transformations to a mesenchymal phenotype. Genotypically this was associated with significant stromal loss of the junctional adhesion molecules Claudin-1 and E-cadherin, and up-regulation of invasion-related N-cadherin, vimentin and MMP2 & 3 (Figure 1E).” First, the authors did not perform a phenotypic assay that would allow one to conclude a transformation to a mesenchymal phenotype. Second, the authors seem to have a wrong understanding of the term “genotype”.

Response: “genotype” was also deleted. More appropriate description was adapted in revised manuscript and marked in Red on **page 6 line 144** as below:

“Phenotypically, tumor cells in the WPOI 4-5 category showed a significant mesenchymal phenotype when compared with tumors with WPOI 1-3. Significant stromal loss of the junctional adhesion molecules Claudin-1 and E-cadherin, and up-regulation of invasion-related N-cadherin, vimentin and MMP2 & 3 were observed in WPOI 4-5 tissues (**Fig. 1d**).”

3) data in Figure 1E should be presented as box plots with individual data points shown. Furthermore, in Figure 3D the authors performed a correlation analysis and present the correlation matrix. Usually one would expect that each gene shows high correlation with itself resulting in a uniform and intense diagonal row of squares. However, in this figure the correlation of each gene with itself seems to differ. What is plotted in this figure and what is the colour legend telling us?

Response: We thank the reviewer for their insightful comments and to address these comments, we have added as requested individual data points for **Fig. 1d** as in our revised manuscript. Moreover, we apologise for the error in **Fig. 3c**. The R value of correlation analysis on each gene with itself should of course be 1, we have since corrected this issue in our revised **Fig 3c**.

4) data shown in Figure 4M, N were not described in the main text whereas Suppl. Figure S4F shows the opposite of what is described in the text or would be expected based on the other data. In detail, treatment of HN6 cells with CCL26 induced E-cadherin expression and decreased N-cadherin levels suggesting a MET instead of an EMT.

Response: We thank the reviewer for their insightful comments and in order to address these comments we have supplemented the description of Figure 4 in our revised manuscript (marked in Red on page 13). In addition, we have noticed that the labelling of E-cadherin and N-cadherin was actually reversed in **Suppl Fig. 4f**, we apologise and have corrected in a revised version of the figure.

5) Finally, in the discussion on page 38, line 964 the authors state that serum OXT was up-regulated in OSCC patients. While this is true, the data presented in Figure 8 question the relevance of the OXT/OXTR system in OSCC, especially due to the lack of mechanistic insights into OXTR up-regulation (see below).

Response: We would like to thank the reviewer for their insightful comments and the opportunity to provide additional mechanistic insights into OXTR up-regulation.

Firstly, to clarify the production process of OXT *in vivo*. OXT is a highly conserved peptide of nine amino acids synthesized by nerve cells and is released into the blood¹⁻³. OXT plays an important role in regulating digestion, blood pressure, heart rate, and pain etc., which are vulnerable to dysregulation during carcinogenesis^{4,5}. **Exposure to various stressors activates OXT neuronal activity and OXT secretion into blood. Noting that cancer patients are often under great psychological stress^{6,7}.** Therefore, the serum OXT level in OSCC patients was not surprisingly elevated when compared with healthy controls. Our study also found that OXT levels did not necessarily correlate with clinical outcomes, even in OSCC patients with high TNM stage or with WPOI 4-5, which harbored high OXTR expression (**Fig. 8i and Supplementary Fig. 6c-f**). Overall our results suggest that the elevated serum OXT in OSCC patients might be as a consequence of tumor stress.

Secondly, OXT *in vitro* and *in vivo* functions: In **Supplementary Fig. 3e-f**, no OXT and AVP were detected in any samples of cell culture media in this study. Considering the endogenous oxytocin in blood, we therefore also evaluated the response of CAFs to the treatment of OXT/AVP. The results (**Supplementary Fig. 6g**) showed that OXT/AVP could activate ERK1/2 or AKT as previously reported, however, no enhanced pro-invasive effects of CAFs were found (**Fig. 8k-l and Supplementary Fig. 6h**). Since Carbetocin (Med Chem Express china, #HY-17573) a long-acting oxytocin analogue, has been reported to selectively activate only the OXTR/Gq pathway⁸, we used it (twice weekly, 6 mg/kg/mice, i.p.) in an orthotopic model of OSCC in NCG mice. However, the treatment of Carbetocin showed no effects on OSCC progression (**Supplementary Fig. 3g**) our findings therefore further support a ligand-independent manner of action for OXTR^{High} CAFs in tumor invasion.

Lastly, the molecular mechanism of OXTR activation signaling. We have supplemented Figure 6 with additional data at this reviewer's request to improve the logical fluency and to reframe the key logic with a reconstructed Figure 6:

1) ERK5 regulates OXTR/CCL26 transcription: In order to identify and specify the effectors of the MAPK pathway, we overexpressed OXTR in OXTR^{Low} CAFs to mimic OXTR^{High} CAFs. Through several analysis formats, we found more activated nuclear ERK5 (Fig. 6d-e), rather than JNK, ERK1/2 or p38, or downstream c-fos/c-myc signaling activation in the sa-OXTR group. Moreover, ERK5 was found to be significantly activated in OXTR^{High} CAFs of OSCC patients, even without oxytocin stimulation. Importantly, both knockdown of ERK5 (Fig. 6g) and mutation of the c-fos/c-myc promoter binding site (Fig. 6j and I) significantly reduced OXTR and CCL26 transcription in OXTR^{High} CAFs in a dual-luciferase reporter assay.

2) OXTR/Gq and CDC37 signaling activates ERK5: we determined why ERK5 activated in OXTR^{High} CAFs. Using co-immunoprecipitation (IP) assays we showed that OXTR overexpression could also increase the activity of the CDC37 pathway (Fig. 6m-n) and Gq signaling (Fig. 6o-p). Inhibition of Gq/ERK5 activation by Gq knockdown also confirmed that OXTR was able to activate the Gq/MEK5/ERK5 pathway (Fig. 6q-r). Consequently, up-regulated OXTR could in turn further activate ERK5/TFs signaling to form a positive feed-back loop, which was the key mechanism for OXTR^{High} CAFs to maintain it's a high OXTR and CCL26 phenotype. Diagram of pathway shown below:

Reference

1. Ludwig, M. Dendritic release of vasopressin and oxytocin. *J Neuroendocrinol* **10**, 881-895 (1998).
2. Gulliver, D., *et al.* Targeting the Oxytocin System: New Pharmacotherapeutic Approaches. *Trends Pharmacol Sci* **40**, 22-37 (2019).
3. Landgraf, R., Neumann, I. & Schwarzberg, H. Central and peripheral release of vasopressin and oxytocin in the conscious rat after osmotic stimulation. *Brain Res* **457**, 219-225 (1988).
4. Kidoguchi, S., *et al.* New Concept of Onco-Hypertension and Future Perspectives. *Hypertension* **77**, 16-27 (2021).
5. Scarborough, B.M. & Smith, C.B. Optimal pain management for patients with cancer in the modern era. *CA Cancer J Clin* **68**, 182-196 (2018).
6. Douglas, A.J., *et al.* The role of endogenous opioids in neurohypophysial and hypothalamo-pituitary-adrenal axis hormone secretory responses to stress in pregnant rats. *J Endocrinol* **158**, 285-293 (1998).
7. Onaka, T. & Takayanagi, Y. Role of oxytocin in the control of stress and food intake. *J Neuroendocrinol* **31**, e12700 (2019).

8. Passoni, I., Leonzino, M., Gigliucci, V., Chini, B. & Busnelli, M. Carbetocin is a Functional Selective Gq Agonist That Does Not Promote Oxytocin Receptor Recycling After Inducing beta-Arrestin-Independent Internalisation. *J Neuroendocrinol* **28**(2016).

6) Description of experimental procedures and details.

Certain data lack a detailed experimental description that would allow one to either repeat the experiment or at least understand better what has been done. For example, in Figure 2E the authors schematically describe the isolation of CAFs from patients with WPOI 1-3 or 4-5. Subsequent experiments heavily rely on these 3 or 9 lines, respectively. However, it remains a mystery whether these CAF lines have been mixed after their isolation or kept separate. In case of the latter, it is unclear which line (and why) was used for the experiments and whether always the same CAF line was used and/or validation experiments with the other lines have been performed.

Response: We thank the reviewer for their insightful comments and we have endeavoured to improve clarity of our experimental methods. All CAF cell lines were cultured and used individually throughout the study. According to its acquisition time, for the CAF^{WPOI 1-3} group, #11.9 or #8.30 CAFs were used and for the CAF^{WPOI 4-5} group, #7.4 and #8.9 CAFs were used. **Just to be clear for each independent experiment, mixing of independently derived CAFs did not occur.** In addition, all four CAF cell lines showed similar proliferation rates in culture. CAFs #7.4 and #8.30 were used and related data were shown in this manuscript. CAFs #11.9 and #8.9 were used to validate and reduplicate the key findings of #7.4 and #8.30 CAFs in a preliminary study. For example, in the overexpression assay of Figure 6H, #11.9 and #8.30 CAFs were used and labelled as CAF1 or CAF2, separately. Due to space restrictions, data from CAF1 was shown in Figure 6. We have added the cell line information to the **Materials and Methods section** of our revised manuscript on **page 37**.

7) the mouse model used in this study is poorly described and it remains unclear how the tissue-specific knockout of OXTR was achieved. The paragraph in the main text is not helpful and makes no sense (page 19, lines 493-497). It seems that the model is based on CreERT2. Hence, tamoxifen-induction would be needed, yet no such treatment was mentioned in the manuscript. Also, the labels creW and creT are uncommon and need to be explained.

Response: We thank the reviewer for their insightful comments due to lack of clarity with our murine model. We have now included the relevant Information of our mouse model in the **Materials and Methods section** of our revised manuscript on **page 35**. Importantly, this model is not based on CreERT2 and tamoxifen-induction, an error we have since corrected, we apologise for our mistake. Moreover, O_{xtr}^{fl/fl} S100a4-cre^W indicates wild type mice (O_{xtr}^{fl/fl}), while O_{xtr}^{fl/fl} S100a4-cre^T indicates transgenic mice (S100a4-cre; O_{xtr}^{fl/fl}). We have used the nomenclature O_{xtr}^{fl/fl} and S100a4-cre; O_{xtr}^{fl/fl} in our revised manuscript.

8) The manuscript lacks detailed sequence information about the si/shRNAs and

qPCR primers used.

Response: We supplemented the information about si/shRNAs and q-PCR primers in supplemental table 2-3 according to your suggestion.

9) Furthermore, the molecular size of OXTR differs between Figure 4 (75 kDa) and Figure 6B, H, I, K, and Q. While the authors discuss different OXTR sizes without any context in the main text on page 18, lines 466-471, this explanation makes no sense, because the same cells (CAFs) have been analyzed in Figure 4 and 6.

Response: We are so sorry for this wrong mark in Figure 6, the molecular size of OXTR in CAFs was always 75 kDa instead of 55 kDa. We apologize to you again for this unintended mistake and we had corrected this error in Figure 6.

10) Next, CRISPR-mediated OXTR depletion was achieved using three different sgRNAs (Figure 4J-L). However, the T7E1 assay lacks a quantification of the cutting efficiencies. Based on the band intensities shown in Figure 4K, the sgRNAs are not very efficient. In fact, Figure 4L shows that OXTR expression is not fully blunted. Here, the authors could have been used a FACS-based negative enrichment strategy to obtain a pure population of OXTR-negative cells.

Response: We thank the reviewer for these insightful comments, and we ask the reviewer to note that the T7E1 assay was used merely as a qualitative exercise as previously reported ¹ (b in figure below). After cas9 knockdown of OXTR, the DNA sequence was analyzed to confirm efficient editing (a in below figure). Notably, a FACS-based negative enrichment strategy was performed twice, with the protein level of OXTR decreased to < 10%. A pure population of OXTR-negative cells was not acquired in present study (c in figure below). Monoclonal cell culture might be an option in further studies and we thank the reviewer for this suggestion.

Reference

1. Gao, Q., *et al.* Heterotypic CAF-tumor spheroids promote early peritoneal metastasis of ovarian cancer. *J Exp Med* **216**, 688-703 (2019).

11) Last but not least, the authors chose to overexpress OXTR using a rather unconventional strategy based on small activating RNAs. It is unclear why this strategy was favoured over standard plasmid-based overexpression or CRISPR activation. Please explain and repeat key OXTR overexpression experiments using stable transduction of the transgene. Also, if saRNA.

Response: Small activating RNAs (saRNAs) strategy was chosen based on two concerns: On the one hand, saRNAs are short double-stranded oligonucleotides that selectively enhance gene transcription in endogenous level and have been developed for clinical use. The mode of action for saRNA is dependent on the complementarity at the 5'-region of the antisense or guide strand oligo to the intended target DNA near or within gene promoters, **which was more close to the regulatory manner in physiological condition**¹⁻³.

On another hand, isolated human CAFs's cell vitality and growth condition were more vulnerable to external stimuli when compared with many established tumor cell lines, leading to significant cell senescence and impaired growth during cell culture *in vitro*. **This is a common and actual problem we would face during primary cell culture.** Importantly, the exogenous introduction of a plasmid containing a large fragment has a stronger cellular stimulation than saRNAs with small fragment. In preliminary study, we ectopically expressed OXTR in human primary CAFs by a pSLenti-3flag-plasmid containing full-length OXTR gene via lentivirus as below, but these primary CAFs overexpressed with OXTR showed poor vitality and growth condition. Considering stable cell growth is a prerequisite for subsequent gene manipulation experiments, saRNA with high safety was chosen in this study.

Reference

1. Hashimoto, A., *et al.* Upregulation of C/EBPalpha Inhibits Suppressive Activity of Myeloid Cells and Potentiates Antitumor Response in Mice and Patients with Cancer. *Clin Cancer Res* **27**, 5961-5978 (2021).
2. Tan, C.P., Sinigaglia, L., Gomez, V., Nicholls, J. & Habib, N.A. RNA Activation-A Novel Approach to Therapeutically Upregulate Gene Transcription. *Molecules* **26**(2021).
3. Sarker, D., *et al.* MTL-CEBPA, a Small Activating RNA Therapeutic Upregulating C/EBP-alpha, in Patients with Advanced Liver Cancer: A First-in-Human, Multicenter, Open-Label, Phase I Trial. *Clin Cancer Res* **26**, 3936-3946 (2020).

12) Open questions.

While the authors present plenty of data, some key questions remain unanswered. For example, how is OXTR expression triggered in OSCC? The authors validated a known downstream signalling connection between OXTR and ERK5 (Devost et al., Progress in Brain Research, 2008) and suggest a positive feedback loop. Yet, it remains unclear how OXTR expression is initially increased in a subgroup of OSCC patients. Importantly, the experimental strategy and logic behind the data shown in Figure 6 does not really make sense. Why would you artificially overexpress OXTR to begin with, if you want to identify endogenous regulators of this gene.

Response: We thank the reviewer for their insightful comments and indeed we also carefully considered the issue, how is OXTR expression initially triggered and therefore elevated in a subgroup of OSCC patients. In the beginning, we considered that the *OXTR* gene could be mutated in CAF^{WPOI 4-5}, similar to ERFR autoactivation. However, no significant mutational difference was found between CAF^{WPOI 4-5} and CAF^{WPOI 1-3} by DNA sequence. In addition, multiple studies have indicated that OXTR DNA-methylation levels be associated with dysregulated OXTR expression, but again we found no difference in methylation levels between CAF^{WPOI 4-5} and CAF^{WPOI 1-3} by BSP analysis in **Supplementary Fig. 2d**. Therefore, DNA- or epigenetic modifying-related factors do not seem to be involved in OXTR up-regulation.

Subsequently, we considered transcriptional regulation as the key initiator. Our mechanistic insights into OXTR up-regulation revealed that activated ERK5 is the key factor for increased *OXTR* transcription and therefore activated ERK5 in OXTR^{High} CAFs. Indeed, in patients with WPOI 4-5, more activated nuclear ERK5 was found in stromal fibroblasts, which suggests that ERK5 might initially increase OXTR in this subgroup.

13) Moreover, the authors used an ERK5 inhibitor and failed to observe an impact on lymph node metastasis and the mesenchymal phenotype of the tumor in vivo (Figure 7G, H). They argue that this might be due to the paradoxical activation of ERK5 transcriptional activity by the inhibitor XMD8-92. This finding in conjunction with other safety and off-target concerns of this inhibitor suggest that ERK5 inhibition might not be the ideal clinical intervention strategy. Hence, it is surprising that the authors did not test the therapeutic relevance of CCL26, the ultimate effector molecule according to their working model (Figure 8). For example, a CCL26-neutralizing antibody or a CCR3 inhibitor should have been tested in vivo. Alternatively, the authors could have turned directly to OXTR using Atosiban or other antagonists.

Response: Thank you for your attention and suggestion. Since CCR3 have several ligands including CCL5 (RANTES), CCL7 (MCP-2), CCL8 (MCP-3), Eotaxin-1/CCL11, CCL13 (MCP-4), CCL15 (HCC-2), CCL24 (Eotaxin-2), CCL26 (Eotaxin-3) etc., CCR3 inhibitor might be inappropriate to use *in vivo*. Thus we knocked down the genes of CCL26 in OXTR^{High} CAFs and established an orthotopic model of OSCC in NOD^{prkdc^{-/-}IL-2Rg^{-/-}} mice. The results below showed that CCL26 knockdown in OXTR^{High} CAFs could inhibit OXTR^{High} CAFs-supported tumor growth and LNM, which was consistent with the findings in Figure 7F.

In the past decade, a number of clinical trials have been performed in autism and schizophrenia but, unfortunately, no consensus on the real efficacy of oxytocin on these conditions has yet been reached ¹, **The OXTR agonists or antagonists were not considered for target therapeutics in this study based on three aspects. Firstly**, a number of OXT analogues have been developed as OXTR agonists or antagonists, but the pro-invasion function of OXTR^{High} CAFs was independent of its ligand OXT or AVP. **Secondly**, oxytocin could bind to and activates the AVP receptor vasopressin V1a and V1b receptor subtypes, which promote effects different from (and even the opposite to) those of OXTR^{2,3}. **Thirdly**, oxytocin promotes OXTR coupling to a number of different G-protein subtypes and β -arrestins, leading to the activation of multiple signaling pathways with different function, whose precise roles are currently unknown⁴. Thus, **the pharmacological tools may not be sufficiently selective to allow easy identification or manipulations of these receptors, especially *in vivo***⁵. **Carbetocin, a long-acting oxytocin analogue**, has been reported to selectively activate only the **OXTR/Gq pathway**⁶, but Carbetocin in orthotopic model of OSCC in **Supplementary Fig. 3g** showed no effects on OSCC progression. These findings further supported the ligand-independent manner of OXTR^{High} CAFs in tumor invasion.

Reference

1. Sarker, D., *et al.* MTL-CEBPA, a Small Activating RNA Therapeutic Upregulating C/EBP-alpha, in Patients with Advanced Liver Cancer: A First-in-Human, Multicenter, Open-Label, Phase I Trial. *Clin Cancer Res* **26**, 3936-3946 (2020).
2. Huber, D., Veinante, P. & Stoop, R. Vasopressin and oxytocin excite distinct neuronal populations in the central amygdala. *Science* **308**, 245-248 (2005).
3. Viviani, D. & Stoop, R. Opposite effects of oxytocin and vasopressin on the emotional expression of the fear response. *Prog Brain Res* **170**, 207-218 (2008).
4. Busnelli, M., *et al.* Functional selective oxytocin-derived agonists discriminate between individual G protein family subtypes. *J Biol Chem* **287**, 3617-3629 (2012).
5. Duque-Wilckens, N., *et al.* Oxytocin Receptors in the Anteromedial Bed Nucleus of the Stria Terminalis Promote Stress-Induced Social Avoidance in Female California Mice. *Biol Psychiatry* **83**, 203-213 (2018).
6. Passoni, I., Leonzino, M., Gigliucci, V., Chini, B. & Busnelli, M. Carbetocin is a Functional Selective Gq Agonist That Does Not Promote Oxytocin Receptor Recycling After Inducing beta-Arrestin-Independent Internalisation. *J Neuroendocrinol* **28**(2016).

14) Readability and Language.

The authors should consider to shorten the manuscript and focus on the most important results in order to improve figure layout and readability. Some data could be fully removed (e.g. Fig.1B, Fig. 3A, Fig.4J) or moved into the supplements. Moreover, some language and typo editing is required, e.g. page 9, lines 224-226 (not 150

cohorts have been analyzed...). Also, please check page 13, line 322 (CAF^{WPOI1-3} instead of two times CAF^{WPOI4-5}). The manuscript title also requires a minor correction. **Response: Thank you for your attention and suggestion.** We corrected these errors in revised manuscript, and also shortened this manuscript according to your suggestions.

Reviewer #5 (Remarks to the Author): with expertise in CAFs

The authors present a very detailed work with the goal of improving diagnosis and prognosis of OSCC. It is by now accepted that cancer-associated fibroblasts are significant factor to take into account when attempting to develop anti-tumor therapies, and as the authors and previous work before show, fibroblasts are important players in OSCC as well. The topic is timely and highly applies to other cancers. The authors describe a pro-invasive phenotype of specific CAF subtype, WPOI 4-5 CAFs, which depends on ERK5, OXTR and CCL26, and demonstrate their functional relevance in vitro and in vivo. They use orthogonal methods to support their hypothesis, and address many aspects in deciphering the mechanism of action of this CAF subset. The authors also appropriately refer to previous work in the field of CAF heterogeneity, and place their findings within the frame of different CAF subtypes that have been demonstrated in previous research.

In general, I think that the manuscript is very thorough and convincing.

Response: We would like to thank the reviewer for their very positive feedback on our manuscript. We underlined all your key questions as below.

General comments:

1) When I tried to figure out how the authors prepared their patient-derived fibroblasts I had to actively search for it in another publication of theirs. Since this is not a standard procedure, and people are using different protocols to produce fibroblasts, this is something that **(1) should be clearly detailed in the methods, and (2) should be mentioned in the manuscript.** the authors used the outgrowth method with differential trypsinization to purify the CAFs from the cancer cells. This is not the best way to purify CAFs, and might leave some cancer cells behind. There are several surface markers (some are used by the authors later on) that can be used to purify your CAF culture. **(3) The authors should at least show by FACS that their CAF population is pure by staining for CAF markers and epithelial markers.** While searching for the CAF preparation protocol, I was also met with the fact that all the CAF (co)cultures in this manuscript are performed with hTERT immortalized CAFs. This is definitely something that should be highlighted. **(4) It is known that immortalization may change gene expression and cell morphology.** This is an important detail that should not escape reader's eye. I would want to see that key findings are repeated with primary fibroblasts.

Response: We here response to **four points** as you suggested including: **1.** How to

acquire CAFs; **2.** How to validate; **3.** Why to choose immortalization; **4.** Repeated key findings in primary fibroblast.

1. How to acquire CAFs: Primary OSCC-derived CAFs were isolated and validated according to our three previously studies by enzymatic hydrolysis combining with tissue adhesion method¹⁻³, which was specifically optimized by our group. The detailed information was added in **Materials and Methods section** of revised manuscript on **page 37.**

2. How to validate: In our previous study, we used cytokeratin as an epithelial cell marker. The OSCC cells, HSC3 were derived from the epithelium and used as control cells. We found no epithelial cells mixed with isolated primary CAFs (**Figure below part a**). Our results by FACS analysis also indicated that CAFs in OSCC highly expressed other markers, such as FSP-1 (S100a4), FAP etc. but were negative for EPCAM+ (CD326) (**Figure below part b**). In conclusion our analyses indicated that tumor cells were successfully excluded from the CAF population in this study.

3. Why to choose immortalization: Cell senescence and phenotypic loss are a common problem after 5-8 cell passages using primary CAFs in experiments. **Since many of our experiments required gene manipulation and long-time organoid culture**, we were also faced with this issue. It was also necessary to obtain tissues from many patient individuals, each requiring isolation of primary CAFs, resulting in relatively impaired stability of experimental data due to the heterogeneous nature of deriving cells from multiple patients.

To resolve this issue, hTERT immortalized CAFs were developed and used in our previous studies⁴⁻⁶. In the preliminary study, when compared with its corresponding primary CAFs (5th passage), hTERT immortalized CAF also showed similarly high level of FSP-1 (S100a4), α -SMA and PDGFR β (**Figure below part a**), and also showed high ability of pro-migration of OSCC cells. After 10 cell passages, the primary CAFs were significantly senescent and showed no proliferation, but immortalized CAF still remained stable growth rate (**Figure below part b-d**). Chromosome karyotype analysis further confirmed that there were no obvious karyotype differences between

them (Figure below part e).

4. Repeated key findings in primary fibroblast: In our preliminary study, siRNA of the OXTR gene was adapted at inhibiting OXTR function in primary CAFs, with the key phenotype, function and preliminary mechanism determined. Since OXTR^{High} CAFs showed high expression of PDGFR β and N-cadherin and high ability of promoting tumor invasion, siRNA-OXTR was also performed on primary CAF^{WPOI 4-5} (Supplementary Fig. 5k). We found that inhibition of OXTR by siRNA could inhibit tumor invasion using the transwell assay (Supplementary Fig. 11-n). Knockdown of OXTR by siRNA for 48 h was also able to impair ERK5/c-fos/c-jun phosphorylation and down regulated CCL26 expression (Supplementary Fig. 1o). Therefore, these findings performed by siRNA-OXTR are consistent with previous results.

Reference

1. Ding, L., *et al.* A novel stromal lncRNA signature reprograms fibroblasts to promote the growth of oral squamous cell carcinoma via lncRNA-CAF/interleukin-33. *Carcinogenesis* **39**, 397-406 (2018).
2. Zhao, X., *et al.* Diminished CD68(+) Cancer-Associated Fibroblast Subset Induces Regulatory T-Cell (Treg) Infiltration and Predicts Poor Prognosis of Oral Squamous Cell Carcinoma Patients. *Am J Pathol* **190**, 886-899 (2020).
3. Wang, Y., *et al.* Epiregulin reprograms cancer-associated fibroblasts and facilitates oral squamous cell carcinoma invasion via JAK2-STAT3 pathway. *Journal of experimental & clinical cancer research : CR* **38**, 274 (2019).
4. Lecker, L.S.M., *et al.* TGFBI Production by Macrophages Contributes to an Immunosuppressive Microenvironment in Ovarian Cancer. *Cancer Res* **81**, 5706-5719 (2021).
5. Wan, X., *et al.* FOSL2 promotes VEGF-independent angiogenesis by transcriptionally activating Wnt5a in breast cancer-associated fibroblasts. *Theranostics* **11**, 4975-4991 (2021).
6. Ligorio, M., *et al.* Stromal Microenvironment Shapes the Intratumoral Architecture of Pancreatic Cancer. *Cell* **178**, 160-175 e127 (2019).

2) The choice of S100a4-Cre mouse is not trivial and requires an explanation: did the author see that OXTR+ CAFs also express s100a4? I say this because single cell analysis of fibroblasts has shown that this marker is not specific nor inclusive enough to cover all CAFs. If the OSCC CAFs are mostly positive for this marker, this is a piece of data I would like to see (probably in supp info)

Response: We thanks the reviewer for this comment and we agree that the expression gene *FSP-1* (fibroblast specific protein-1; S100A4) is not exclusive to the cancer associated fibroblasts. However, S100a4-Cre mice have been utilised in many studies to investigate the role of stroma fibroblast during carcinogenesis for decades¹⁻³. We have also supplemented our revised manuscript with additional data to show that the majority of CAFs (>97%) were FSP-1 positive (**Figure below part a**). Moreover, FSP-1/ α -SMA immunofluorescence staining also confirmed the extensive stromal expression pattern of FSP-1+ CAFs in OSCC tissues.

Reference

1. Bhowmick, N.A., *et al.* TGF-beta signaling in fibroblasts modulates the oncogenic potential of adjacent epithelia. *Science* **303**, 848-851 (2004).
2. Peuhu, E., *et al.* SHARPIN regulates collagen architecture and ductal outgrowth in the developing mouse mammary gland. *EMBO J* **36**, 165-182 (2017).
3. O'Connell, J.T., *et al.* VEGF-A and Tenascin-C produced by S100A4+ stromal cells are important for metastatic colonization. *Proc Natl Acad Sci U S A* **108**, 16002-16007 (2011).

Minor comments:

3) There are many grammar issues in the manuscript, along with some typos and inconsistencies. The authors should fix these; The authors should fix the citations so that they match what they are citing.

Response: We take note of the reviewer's concerns regarding the grammar issues and have consulted with native English speakers in this regard. All the citations in our revised manuscript have also been thoroughly checked.

4) Some of the IHC figures are very small and low resolution (example: Fig 2A). It is hard to judge what they show.

Fig 2J shows that the CAF WPOI 1-3 organoids are higher in density compared to the WPOI 4-5, which is in contrast to the description of the results in the manuscript (rows 248-249).

Response: We note the reviewer's concerns and have now provided high-resolution images and additional higher magnification specification in our revised manuscript (**Fig. 1a and d, Fig. 2a**). We apologise that the labels in Figure 2J were in the wrong order,

we have now corrected this problem in revised manuscript.

5) Authors should note that Col1 is not an exclusive marker for fibroblasts. I would suggest double staining to show specificity. New markers for CAFs have been established in the last 2-3 years in other publications and the authors should take advantage of these to refine their staining.

Response: We agreed with the reviewer that the expression gene Col1 is not exclusive to fibroblasts, but stromal fibroblasts are the primary source of collagen in the tumor microenvironment. We here also investigated the **Fibronectin⁺ and PDGFR β ⁺ stromal fibroblasts** in WPOI 1-3 and WPOI 4-5 OSCC tissues. Results also showed the similar findings that stroma fibroblasts were increased in the WPOI 4-5 tissues. Related data had been added in revised manuscript in **Supplemental Fig. 1f-g**

6) "HN6" first mention in row 260 with no previous explanation/spelling out Fig 4C – this is an unusual way to show gating. I would expect an explanation at least in the legend

Response: We agree with the reviewer's suggestion and have now added a statement that HN6 is a OSCC cell line with high metastasis potential, in our revised manuscript. The radar chart used in Fig 4C was based on the Q-PCR results and showed the difference in markers between OXTR^{High} CAF and OXTR^{Low} CAF. In accordance with the reviewer's suggestion, we have added this information to the legend for this figure in our revised manuscript.

7) Fig 6H – total ERK5 is elevated mostly in one cell line and not so much in the other. Can the author show another cell line to strengthen their claim?

Response: Heterogeneous CAFs subgroups in different tumor patients render discrepancies in expression gene, such as ERK5. To clarify and allay the reviewer's concerns, we have provided further ERK5 data from additional cell lines. CAF3-5 showed a level of background expression of p-ERK5, overexpression of OXTR but pertinently up-regulated p-ERK5 levels.

REVIEWER COMMENTS

Reviewer #1 (Remarks to the Author):

The authors have addressed the majority of my previous comments and concerns, but they should address the following remaining concerns.

My remaining concerns are:

1. In the concluding sentence of the abstract make clear that you are suggesting targeting of ERK5 as a potential therapeutic strategy in OSCC with OXTR high CAFs with WPOI.

2. In lines 243 and 244 and 247, the suggestion that CAF WPOI 4-5 are a key determinant for tumor invasion and metastasis in WPOI 4-5 type tumors is worded too strongly.

3. There are very serious grammatical errors that impede understanding throughout the manuscript, and these must be corrected before this manuscript is suitable for publication. Even very important acronyms such as GPCR were misspelled.

4. On lines 295 to 297, the assertion that "To date, the function of peripheral OXTR signaling has been elusive, particularly during carcinogenesis" really needs to be tempered, as the authors have also now acknowledged that previous studies have also implicated OXTR signaling in progression of other cancers.

5. In lines 736 to 740, the authors state that "On the other hand, serum OXT levels were significantly much higher in OSCC patients when compared with either healthy people or patients with leukoplakia increased significantly as the tumor progressed (Median: 81.21 pg/ml in OSCC patients vs 16.5 pg/ml in healthy people) (Fig. 8a-b)." This finding cannot be dismissed, and needs to be adequately explained in the light of their other findings, and needs to be considered in their discussion/conclusion.

6. The use of "etc" in line 803 is inappropriate. Instead, authors should reference a more comprehensive list of cancers in which OXTR signaling has been implicated. A notable cancer that was omitted is pancreatic cancer that also has highly desmoplastic stroma. Authors could consider this reference (Hepatobiliary Pancreat Dis Int. 2020;19(2):175-180) or a similar one.

Reviewer #2 (Remarks to the Author):

The reviewer remains unconvinced that the major proinflammatory cytokines and chemokines don't play a major role in oral cancer metastasis. The results are not solid enough for the journal so I cannot recommend for the publication.

Reviewer #3 (Remarks to the Author):

The paper has been improved, but my overall opinion remains that this is a data heavy study that does not really manage to establish the biological significance of the findings. Moreover, I am still very confused about the mechanism by which OXTR stimulates ERK5 signaling through $G\alpha_q$ in CAFs. On one hand, the authors provide evidence that MEK5 and phospho-ERK5 are in a complex with $G\alpha_q$ (Fig. 6O and P), indicating that OXTR activates MEK5/ERK5 via $G\alpha_q$. However, the demonstration that $G\alpha_q$ is required by using siRNA knockdown is not convincing given that siRNA- $G\alpha_q$ caused a significant decrease in OXTR expression in Sa-OXTR CAFs to a level below that detected in Sa-NC CAFs (compare lanes 1 and 2 with lane 3 in panel Q). On the other hand, they imply that OXTR-mediated increased MEK5 expression contributes to ERK5 activation (Fig. 6F and S). How these two

branches (i.e. $G\alpha_q$ induced MEK5 activation versus OXTR induced MEK5 expression) are connected is not clear from the schematic in Fig. 6U. Likewise, what is the evidence that CDC37 significantly contributes to mediating ERK5 phosphorylation downstream of OXTR, apart from showing that CDC37 coimmunoprecipitates with ERK5 (Fig. 6M). In addition, a number of essential controls are missing in the immunoblots presented in Fig 6. Specifically, panels H, M and O must include OXTR in WCL to confirm the higher level of OXTR expression in Sa-OXTR CAFs compared with Sa-NC CAFs. Likewise the level of ERK5 and p-ERK5 must be systemically assessed in WCL. Total ERK5 level should also be included in panel Q to interpret the effect of $G\alpha_q$ silencing on ERK5 phosphorylation. Panel O must show the level of MEK5 in WCL. Finally, I am very puzzled to observe what appears to be such a large amount of p-c-Jun, p-c-Fos and p-c-Myc interacting with ERK5 given that the interaction of MAPKs with their respective transcription factors have always been very difficult to detect by immunoblot following co-immunoprecipitation. Moreover, there is very limited evidence that c-Jun, c-Fos and c-Myc are substrates of ERK5.

The authors must rephrase the new sentence included in response to my initial comment 7 on page 20 line 503 to improve clarity.

Reviewer #4 (Remarks to the Author):

The authors addressed my comments. The new data on the role of CCL26 for the formation of lymph nodes metastasis in xenograft assays using HN6+OXTRhighCAF support their findings and must be included in the manuscript (as well as a proper description in the method section)! Frankly, I don't understand why this was not done during the revision process.

Reviewer #5 (Remarks to the Author):

Regarding my general comment #1:

- Add catalog # of the enzymes used in CAF preparation and their final concentrations.
- Trypsinase is (I guess) a commercial product. Please provide catalog # and concentration used for the differential epithelial/fibroblast dissociation.

Otherwise I am satisfied with the additional data presented.

Regarding my general comment #2:

- I am satisfied with the additional data presented

Upon adding the details above, I support publication of this manuscript.

NCOMMS-21-34717B: OXTR^{High} stroma fibroblasts control the Invasion Pattern of Oral Squamous Cell Carcinoma via ERK5 Signaling

Response to reviewer 1: Page 1-3

Response to reviewer 2: Page 3-5

Response to reviewer 3: Page 6-10

Response to reviewer 4: Page 10

Response to reviewer 5: Page 10

REVIEWER COMMENTS

Reviewer #1 (Remarks to the Author):

The authors have addressed the majority of my previous comments and concerns, but they should address the following remaining concerns.

Response: We would like to thank you for the time and effort spent in reviewing our manuscript.

- 1)** In the concluding sentence of the abstract make clear that you are suggesting targeting of ERK5 as a potential therapeutic strategy in OSCC with OXTR high CAFs with WPOI.

Response: We have modified the revised manuscript according to the reviewer's suggestion (marked in Red in revised manuscript on page 3):

"Therefore, targeting ERK5 signaling of OXTR^{High} CAFs is a potential therapeutic strategy for OSCC patients with WPOI 4-5".

- 2)** In lines 243 and 244 and 247, the suggestion that CAF WPOI 4-5 are a key determinant for tumor invasion and metastasis in WPOI 4-5 type tumors is worded too strongly.

Response: We thank the reviewer for their suggestion and have corrected the descriptions accordingly:

"line 250: Collectively, these data suggested that CAF^{WPOI 4-5} engaged in the tumor invasion and metastasis phenotype of WPOI 4-5 type tumor"

"line 256: Fig 2. CAF^{WPOI 4-5} enhances the invasiveness of OSCC."

- 3)** There are very serious grammatical errors that impede understanding throughout the manuscript, and these must be corrected before this manuscript is suitable for publication. Even very important acronyms such as GPCR were misspelled.

Response: We thank the reviewer for raising this important concern. We have

further polished the manuscript, correcting typos and grammatical errors.

- 4) On lines 295 to 297, the assertion that "To date, the function of peripheral OXTR signaling has been elusive, particularly during carcinogenesis" really needs to be tempered, as the authors have also now acknowledged that previous studies have also implicated OXTR signaling in progression of other cancers.

Response: We thank the reviewer for this suggestion and we have modified the sentence in revised manuscript as below:

".....the function of peripheral OXTR signaling during carcinogenesis of OSCC requires further clarification"

- 5) In lines 736 to 740, the authors state that "On the other hand, serum OXT levels were significantly much higher in OSCC patients when compared with either healthy people or patients with leukoplakia increased significantly as the tumor progressed (Median: 81.21 pg/ml in OSCC patients vs 16.5 pg/ml in healthy people) (Fig. 8a-b)."This finding cannot be dismissed, and needs to be adequately explained in the light of their other findings, and needs to be considered in their discussion/conclusion.

Response: We thank the reviewer for raising this concern. **Firstly**, OXT is synthesized by nerve cells *in vivo* and is released into the blood to regulate digestion, blood pressure, heart rate, and pain etc.¹⁻³, which are vulnerable to be dysregulated during carcinogenesis⁴⁻⁵. **Secondly**, cancer patients are often under great psychological stress⁶⁻⁷, and exposure to various stressors activates OXT neuronal activity and OXT secretion into blood. Therefore, the serum OXT level in the whole OSCC patients was not surprisingly elevated when compared with healthy control. Our study also found that OXT level showed no effects on the clinical outcomes, even in OSCC patients with high TNM stage or with WPOI 4-5, which harbored high OXTR expression. Therefore, these findings imply that elevated serum OXT might be due to tumor stress seen in all OSCC patients.

We have addressed this issue in our revised manuscript (marked in red) in the discussion section (line 809). Thank you again for this comment.

Reference

1. Ludwig, M. Dendritic release of vasopressin and oxytocin. *J Neuroendocrinol* 10, 881-895 (1998).
2. Gulliver, D., et al. Targeting the Oxytocin System: New Pharmacotherapeutic Approaches. *Trends Pharmacol Sci* 40, 22-37 (2019).
3. Landgraf, R., Neumann, I. & Schwarzberg, H. Central and peripheral release of vasopressin and oxytocin in the conscious rat after osmotic stimulation. *Brain Res* 457, 219-225 (1988).
4. Kidoguchi, S., et al. New Concept of Onco-Hypertension and Future Perspectives. *Hypertension* 77, 16-27 (2021).
5. Scarborough, B.M. & Smith, C.B. Optimal pain management for patients with cancer in the modern era. *CA Cancer J Clin* 68, 182-196 (2018).

6. Douglas, A.J., et al. The role of endogenous opioids in neurohypophysial and hypothalamo-pituitary-adrenal axis hormone secretory responses to stress in pregnant rats. *J Endocrinol* 158, 285-293 (1998).
7. Onaka, T. & Takayanagi, Y. Role of oxytocin in the control of stress and food intake. *J Neuroendocrinol* 31, e12700 (2019).

6) The use of "etc" in line 803 is inappropriate. Instead, authors should reference a more comprehensive list of cancers in which OXTR signaling has been implicated. A notable cancer that was omitted is pancreatic cancer that also has highly desmoplastic stroma. Authors could consider this reference (*Hepatobiliary Pancreat Dis Int.* 2020;19(2):175-180) or a similar one.

Response: We thank the reviewer for this suggestion and we have cited the indicated reference in our revised manuscript as **reference #64**.

Reviewer #2 (Remarks to the Author):

The reviewer remains unconvinced that the major pro-inflammatory cytokines and chemokines don't play a major role in oral cancer metastasis. The results are not solid enough for the journal so I cannot recommend for the publication.

Response: We would like to thank reviewer #2 for the time and effort spent in reviewing the manuscript.

We would like to make it clear that we agree with reviewer #2 that major proinflammatory cytokines and chemokines do play a major role in carcinogenesis and oral cancer metastasis as we (IL-6, TNF- α , IL-33, CCL17, CCL22)¹⁻⁵ and others have previously reported⁶, and immunosuppressive tumor microenvironment with large amounts of anti-inflammation cytokines (e.g. IL-10, TGF- β) also promote tumor progression of patients with advanced stage. Currently, whether major pro-inflammatory cytokines are primary influencing factors in different WPOI subtypes still remain unclear. Limited studies have focused mainly on the intrinsic genetic changes in WPOI 4-5 tumor cells^{7,8}.

In our revised manuscript, we here provided new data to clarify this issue. **Firstly**, the expressions of five classical inflammation mediators (IL-1 β , IL-6, IL-8, SDF-1, TNF- α) were simultaneously detected *in situ* in OSCC tissues from 14 patients (**shown below**). Our analysis showed that, in addition to SDF-1, fibroblast-derived IL-1 β /IL-6, tumor cell derived-IL-8 and tumor infiltrating lymphocyte derived-TNF- α were confirmed to be up-regulated in OSCC patients with high TNM stage. Data from this analysis has been added to the revised manuscript (**Supplementary Fig. 1f-j**). However, fibroblast-derived IL-1 β was the only cytokine measured to be enriched in WPOI 4-5 tissues (**Supplementary Fig. 1f**). Notably, our previous findings showed that OXTR^{Hgh} CAFs were enriched in WPOI 4-5 tissues and IL-1 β was up-regulated in OXTR^{Hgh} CAFs, which was consistent with IL-1 β expression *in situ* in WPOI 4-5 OSCC

tissues. Importantly, the addition of recombinant human IL-1 β into the 3D co-culture system resulted in a modest increase in Lmax for tumor invasion as previously reported^{9,10}, but addition of CCL26 caused a doubling of Lmax.

IHC scores = The grade of staining intensity (0-3) x The grade of staining area (1-4); Staining intensity: 0=normal, 1=mild, 2= moderate, 3=intense, staining area: 1=0-25%, 2=25-50%, 3=50-75%, 4=75-100%; N=14, Scale bars, 100 μ m. TC: Tumor cell; FC: Fibroblasts; TIL: Tumor-infiltrated lymphocytes.

Secondly, using our biobank and clinical laboratory samples, as previously reported^{11,12}, we retrospectively analyzed the ratio and absolute number of lymphocyte subsets in the peripheral blood of OSCC patients (n=328), categorized according to WPOI subtype (**Table1 below**). Human CD3⁺ T cells, CD3⁺CD4⁺ helper/inducer T cells, CD3⁺CD8⁺ cytotoxic T cells, CD3⁺CD19⁺ B cells, and CD3⁺CD16⁺ and/or CD56⁺ NK cells were analyzed in WPOI 1-3 and 4-5 groups by BD Multitest™ reagent (Cat No.340503) (**a in below figure**). We found no significant difference in the immune cell composition in peripheral blood from the various WPOI subgroups (**Supplementary Fig. 1d-e**), which was consistent with the OSCC tissues immune analysis *in situ* (**Figure 2a**). This additional analysis has been incorporated into revised manuscript in **Supplementary Fig. 1d-e**.

TABLE1. Clinicopathological features of 328 OSCC patients in this study

Features	n(%)
Age(years): median(range)	56 (26-85)
Sex	
Female	125 (38.1)
Male	203 (61.9)
Tumor site	
Buccal mucosa	67 (20.4)
Tongue	137 (41.8)
Gingiva	45 (13.7)
Others	79 (24.1)
Tumor TNM stage	
I	77 (23.5)
II	112 (34.1)
III	56 (17.1)
IV	83 (25.3)
Tumor differentiation	
Well	73 (22.3)
Moderately/Poor	255 (77.7)
Worst pattern of invasion	
1-3	214 (65.2)
4-5	114 (34.8)

Finally, our analysis of clinical samples and those of others have confirmed that WPOI is an independent prognostic factor for early-stage OSCC patients^{13,14}. In this

study, we have expanded the analysis of patient cohorts to also incorporate advanced stage OSCC patients, where WPOI was also found to be the independent prognostic factor for OSCC patients.

Therefore, major proinflammatory cytokines and chemokines are important contributors to tumor growth and migration, however, it may not be the primary factor for inducing different WPOI subtype as our work suggests. We have supplemented and discussed these data in our revised manuscript (**page 9 and 19**, marked in red).

References

1. Low, J.T., *et al.* Loss of NFKB1 Results in Expression of Tumor Necrosis Factor and Activation of Signal Transducer and Activator of Transcription 1 to Promote Gastric Tumorigenesis in Mice. *Gastroenterology* **159**, 1444-1458 e1415 (2020).
2. Liang, D., *et al.* Activated STING enhances Tregs infiltration in the HPV-related carcinogenesis of tongue squamous cells via the c-jun/CCL22 signal. *Biochim Biophys Acta* **1852**, 2494-2503 (2015).
3. Zhao, X., *et al.* Diminished CD68(+) Cancer-Associated Fibroblast Subset Induces Regulatory T-Cell (Treg) Infiltration and Predicts Poor Prognosis of Oral Squamous Cell Carcinoma Patients. *Am J Pathol* **190**, 886-899 (2020).
4. Wang, Y., *et al.* Epiregulin reprograms cancer-associated fibroblasts and facilitates oral squamous cell carcinoma invasion via JAK2-STAT3 pathway. *Journal of experimental & clinical cancer research : CR* **38**, 274 (2019).
5. Ding, L., *et al.* A novel stromal lncRNA signature reprograms fibroblasts to promote the growth of oral squamous cell carcinoma via lncRNA-CAF/interleukin-33. *Carcinogenesis* **39**, 397-406 (2018).
6. Todoric, J. & Karin, M. The Fire within: Cell-Autonomous Mechanisms in Inflammation-Driven Cancer. *Cancer Cell* **35**, 714-720 (2019).
7. Jayakar, S.K., *et al.* Apolipoprotein E Promotes Invasion in Oral Squamous Cell Carcinoma. *Am J Pathol* **187**, 2259-2272 (2017).
8. Loudig, O., *et al.* Illumina whole-genome complementary DNA-mediated annealing, selection, extension and ligation platform: assessing its performance in formalin-fixed, paraffin-embedded samples and identifying invasion pattern-related genes in oral squamous cell carcinoma. *Hum Pathol* **42**, 1911-1922 (2011).
9. Chen, X., *et al.* IL-1beta maintains the redox balance by regulating glutaredoxin 1 expression during oral carcinogenesis. *J Oral Pathol Med* **46**, 332-339 (2017).
10. Finegan, K.G., *et al.* ERK5 is a critical mediator of inflammation-driven cancer. *Cancer Res* **75**, 742-753 (2015).
11. Ding, Z., *et al.* CD38 Multi-Functionality in Oral Squamous Cell Carcinoma: Prognostic Implications, Immune Balance, and Immune Checkpoint. *Front Oncol* **11**, 687430 (2021).
12. Fu, Y., *et al.* Worst Pattern of Perineural Invasion Redefines the Spatial Localization of Nerves in Oral Squamous Cell Carcinoma. *Front Oncol* **11**, 766902 (2021).
13. Pu, Y., *et al.* Biopsy pattern of invasion type to determine the surgical approach in early-stage oral squamous cell carcinoma. *Virchows Arch* **479**, 109-119 (2021).
14. Almagush, A., *et al.* Depth of invasion, tumor budding, and worst pattern of invasion: prognostic indicators in early-stage oral tongue cancer. *Head Neck* **36**, 811-818 (2014).

Reviewer #3 (Remarks to the Author):

The paper has been improved, but my overall opinion remains that this is a data heavy study that does not really manage to establish the biological significance of the findings. Moreover, I am still very confused about the mechanism by which OXTR stimulates ERK5 signaling through $G\alpha_q$ in CAFs.

Response: We would like to thank reviewer #3 for the time and effort spent in reviewing and providing constructive suggestions helped to improve the quality of this manuscript.

1. **Question #1:** On one hand, the authors provide evidence that MEK5 and phospho-ERK5 are in a complex with $G\alpha_q$ (Fig. 6O and P), indicating that OXTR activates MEK5/ERK5 via $G\alpha_q$. However, the demonstration that $G\alpha_q$ is required by using siRNA knockdown is not convincing given that siRNA- $G\alpha_q$ caused a significant decrease in OXTR expression in Sa-OXTR CAFs to a level below that detected in Sa-NC CAFs (compare lanes 1 and 2 with lane 3 in panel Q).

Question #2: On the other hand, they imply that OXTR-mediated increased MEK5 expression contributes to ERK5 activation (Fig. 6F and S). How these two branches (i.e. $G\alpha_q$ induced MEK5 activation versus OXTR induced MEK5 expression) are connected is not clear from the schematic in Fig. 6U.

Question #3: Likewise, what is the evidence that CDC37 significantly contributes to mediating ERK5 phosphorylation downstream of OXTR, apart from showing that CDC37 coimmunoprecipitates with ERK5 (Fig. 6M).

Response to Question #1: In present study, we found that ERK5 signaling attributed to the expression of OXTR and CCL26 in CAFs. Additionally, up-regulated OXTR could in turn further activate ERK5 signaling via $G\alpha_q$ and CDC37 to form a positive feed-back loop. $G\alpha_q$ inhibition possibly acting as a brake. SaRNAs are short double-stranded oligonucleotides that can bind to gene promoters in a seed-dependent manner to activate the associated gene transcription in endogenous level. This approach may be more sensitive in terms of assessing the role of $G\alpha_q$ knockdown than forced exogenous overexpression by plasmid containing full-length gene. Inhibition of $G\alpha_q$ efficiently suppressed OXTR levels in this study using this approach. More importantly, the heterogeneous nature of primary CAFs from different tumor patients may render additional gene expression and regulation discrepancies. According to CAF acquisition time, 11.9 (CAF1), 8.30 (CAF2), 11.22 (CAF3) were used in this manuscript and in the preliminary study, and the knockdown efficiency of $G\alpha_q$ protein in three CAFs are different (**Figure below (a)**). Likewise, the change of OXTR signaling induced by $G\alpha_q$ inhibition was also variable.

In order to address these inconsistencies, we have both supplemented and repeated experiments using all three CAFs. The data presented in **Figure 6q**

derived from CAF1, and Gαq knockdown in CAF2 and CAF3 also inhibited OXTR expression to some extent when compared with CAF1 (**Figure below (b)**). The discrepancy between CAF1 and CAF2/3 might be due to: 1. Higher GNAQ gene level in CAF1; 2. CAF1 patients tend to be younger and also have lymph node metastasis (c-d in below figure). Therefore, these variabilities may account for the observed results in Gαq/OXTR signaling intensity in the three primary CAFs.

Question #2: In order to provide additional evidence for Gαq/MEK5/OXTR signaling, we also knocked down Gαq and analyzed the phosphorylated MEK5 and total MEK5 in sa-OXTR groups. Our data show that Gαq knockdown suppressed sa-OXTR-up-regulated MEK5 signaling (**b in below figure**). We have added this data into revised manuscript as **supplementary Fig. 5k**.

Question #3: Similarly, we also knocked down CDC37 by siRNA in all three CAFs (**a in below figure**) and analyzed the ERK5/OXTR signaling in sa-OXTR groups. Our data show that knockdown of CDC37 could partially reduce sa-OXTR-activated ERK5 expression (**b in below figure**) and inhibit ERK5 nuclear localization (**c in below figure**). Moreover, to further confirm the activation of OXTR/CDC37 pathway, we overexpressed CDC37 to rescue siRNA-OXTR-induced ERK5 inactivation. Our results show that siRNA-OXTR down-regulated ERK5 activation, which could be restored by CDC37 overexpression (**d in below figure**). We have added this data into our revised manuscript as **supplementary Fig. 5i-j**.

2. In addition, a number of essential controls are missing in the immunoblots presented in Fig 6. Specifically, panels H, M and O must include OXTR in WCL to confirm the higher level of OXTR expression in Sa-OXTR CAFs compared with Sa-NC CAFs. Likewise, the level of ERK5 and p-ERK5 must be systemically assessed in WCL. Total ERK5 level should also be included in panel Q to interpret the effect of Gαq silencing on ERK5 phosphorylation. Panel O must show the level of MEK5 in WCL.

Response: We thank the reviewer for their insightful comments. To address this comment, we supplemented Figure 6 with all the suggested immunoblot controls (Figure 6 H, M, O and Q) using previously stored protein samples at -80°C. Thank your again for this comment.

3. Finally, I am very puzzled to observe what appears to be such a large amount of p-c-Jun, p-c-Fos and p-c-Myc interacting with ERK5 given that the interaction of MAPKs with their respective transcription factors have always been very difficult to detect by immunoblot following co-immunoprecipitation. Moreover, there is very limited evidence that c-Jun, c-Fos and c-Myc are substrates of ERK5.

Response: We thank the reviewer for this comment and we agree with your point of view. Indeed, few physiological substrates of ERK5 have been identified so far. The best characterized are transcription factor¹. MEF2C in this study was confirmed to interact with ERK5 in all three CAFs as previously reported². Activated ERK5 also acts as an activator of other transcription factors, such as AP-1 transcriptional complex (c-jun/c-fos)^{2,3}. It has been shown that ERK5 directly phosphorylates and activates c-Myc (in 293 cells⁴, pancreatic cancer cells⁵ and glioma cells⁶), c-Fos (in fibroblast-like cell line COS-7^{7,8} and 293T cells⁹). For

example, Der C.J et al. first performed co-immunoprecipitation experiments to demonstrate that *KRAS* knockdown treatment stimulated endogenous ERK5 association with endogenous MYC.

In this revision, additional two primary CAFs from different tumor patients were used to confirm the interaction between ERK5 and transcription factors. All three heterogeneous primary CAFs showed varying degrees of protein binding between ERK5 and AP-1, c-Myc and MEF2C, although CAF2 showed relative weak binding ability (**a and b in below figure**). Thank you again for this comment.

Reference

- Pereira, D.M. & Rodrigues, C.M.P. Targeted Avenues for Cancer Treatment: The MEK5-ERK5 Signaling Pathway. *Trends Mol Med* **26**, 394-407 (2020).
- Kato, Y., et al. BMK1/ERK5 regulates serum-induced early gene expression through transcription factor MEF2C. *EMBO J* **16**, 7054-7066 (1997).
- Morimoto, H., Kondoh, K., Nishimoto, S., Terasawa, K. & Nishida, E. Activation of a C-terminal transcriptional activation domain of ERK5 by autophosphorylation. *J Biol Chem* **282**, 35449-35456 (2007).
- English, J.M., Pearson, G., Baer, R. & Cobb, M.H. Identification of substrates and regulators of the mitogen-activated protein kinase ERK5 using chimeric protein kinases. *J Biol Chem* **273**, 3854-3860 (1998).
- Vaseva, A.V., et al. KRAS Suppression-Induced Degradation of MYC Is Antagonized by a MEK5-ERK5 Compensatory Mechanism. *Cancer Cell* **34**, 807-822 e807 (2018).
- Koncar, R.F., et al. Identification of Novel RAS Signaling Therapeutic Vulnerabilities in Diffuse Intrinsic Pontine Gliomas. *Cancer Res* **79**, 4026-4041 (2019).
- Terasawa, K., Okazaki, K. & Nishida, E. Regulation of c-Fos and Fra-1 by the MEK5-ERK5 pathway. *Genes Cells* **8**, 263-273 (2003).
- Sasaki, T., et al. Spatiotemporal regulation of c-Fos by ERK5 and the E3 ubiquitin ligase UBR1, and its biological role. *Mol Cell* **24**, 63-75 (2006).
- Yamamoto, H., et al. TRAF1 Is Critical for DMBA/Solar UVR-Induced Skin Carcinogenesis. *J Invest Dermatol* **137**, 1322-1332 (2017).

4. The authors must rephrase the new sentence included in response to my initial comment 7 on page 20 line 503 to improve clarity.

Response: We have rephrased the data description in our revised manuscript (page 20) as requested and we hope this version could be more clear and appropriate.

“Moreover, when HN6 was treated with OXTR^{Low} CAFs-derived culture medium, added extra CCL26 caused up-regulation of snail, twist and N-cadherin but down-regulation of E-cadherin expression in HN6 cells. This process was impaired in HN6 cells with CCR3 knockdown (Supplementary Fig. 4f).”

Reviewer #4 (Remarks to the Author):

The authors addressed my comments. The new data on the role of CCL26 for the formation of lymph nodes metastasis in xenograft assays using HN6+OXTR^{high}CAF support their findings and must be included in the manuscript (as well as a proper description in the method section)! Frankly, I don't understand why this was not done during the revision process.

Response: We would like to thank reviewer #4 for this additional comment. As suggested, the data has now been incorporated into the revised manuscript as **supplementary Fig. 5c** in Results section on **page 27**. Relative description was added in the method section on **page 40**.

Reviewer #5 (Remarks to the Author):

Regarding my general comment #1:

- Add catalog # of the enzymes used in CAF preparation and their final concentrations.
- Trypsinase is (I guess) a commercial product. Please provide catalog # and concentration used for the differential epithelial/fibroblast dissociation.

Otherwise I am satisfied with the additional data presented.

Response: We would like to thank reviewer #4 for the time and effort spent in reviewing the manuscript and at your suggestion, the additional information have been added to the revised manuscript on **page 37**.

Regarding my general comment #2:

- I am satisfied with the additional data presented

Upon adding the details above, I support publication of this manuscript.

Response: We would like to thank the reviewer for the time and effort spent in reviewing the manuscript and their support for publication in **Nature Communications**

REVIEWERS' COMMENTS

Reviewer #3 (Remarks to the Author):

The authors have addressed my concerns by performing additional experiments. But, it is still debatable whether or not ERK5 signaling is a relevant pathway downstream of OXTR/Gαq, mostly because the amplitude of the effects associated with "knockdowns" is very small. For example, decreased MEK5 phosphorylation upon the downregulation of CDC37 (new data in panel b provided to answer question 2) is not convincing. Nonetheless, there is huge amount of information in this paper which might be of interest to scientists working in the field of oral squamous cell carcinoma.

NCOMMS-21-34717B: OXTR^{High} stroma fibroblasts control the Invasion Pattern of Oral Squamous Cell Carcinoma via ERK5 Signaling

REVIEWER COMMENTS

Reviewer #3 (Remarks to the Author):

The authors have addressed my concerns by performing additional experiments. But, it is still debatable whether or not ERK5 signaling is a relevant pathway downstream of OXTR/Gαq, mostly because the amplitude of the effects associated with "knockdowns" is very small. For example, decreased MEK5 phosphorylation upon the downregulation of CDC37 (new data in panel b provided to answer question 2) is not convincing. Nonetheless, there is huge amount of information in this paper which might be of interest to scientists working in the field of oral squamous cell carcinoma.

Response: We would like to thank you for the time and effort spent in reviewing our manuscript. In this study, Gαq or ERK5 knockdown by shRNA was used to identify the role of Gαq/PKCζ/ERK5 pathway in the invasive tumor phenotype of OXTR^{High} CAFs. Similarly, stimulation of Gαq-coupled GPCR by angiotensin II in cardiac fibroblasts also found to promote ERK5 activation via PKCζ, and activation of MEK5 signaling was inhibited in PKCζ-deficient mice¹. Further work in Gαq-deficient mice are warranted to clarify the role of Gαq/PKCζ/MEK5 complexes in OXTR/ERK5 activation. We have discussed this issue in revised manuscript on **page 19-20**.

Thank you again for this comment.

Reference

1. Garcia-Hoz, C., *et al.* Protein kinase C (PKC)zeta-mediated Galphaq stimulation of ERK5 protein pathway in cardiomyocytes and cardiac fibroblasts. *J Biol Chem* **287**, 7792-7802 (2012).